# In-Context Learning for Full Bayesian Inference

## Abstract

Transformers have emerged as the dominant architecture in the field of deep learning, with a broad range of applications and remarkable in-context learning (ICL) capabilities. While not yet fully understood, ICL has already proved to be an intriguing phenomenon, allowing transformers to learn in-context—without requiring further training. In this paper, we further advance the understanding of ICL by demonstrating that transformers can perform full Bayesian inference for commonly used statistical models in-context. More specifically, we introduce a general framework that builds on ideas from prior fitted networks and continuous normalizing flows and enables us to infer complex posterior distributions for models such as generalized linear models and latent factor models. Extensive experiments on real-world datasets demonstrate that our ICL approach yields posterior samples that are similar in quality to state-of-the-art MCMC or variational inference methods that do not operate in-context. The source code for this paper is available at `https://anonymous.4open.science/r/ICL_For_Full_Bayesian_Inference-3F53`

## 1 Introduction

Can we leverage in-context learning (ICL) to perform full Bayesian inference? In this paper, we investigate this question. The core principle of ICL is that a system adapts to a given task based on information provided in its context, enabling it to solve complex problems, such as question-answering or text summarization, using a fixed model and without requiring any gradient-based fine-tuning, simply by referencing the context. This way, ICL enables the generation of real-time solutions via local understanding of data without explicit re-training.

ICL has become not only a central paradigm in natural language processing, with LLMs as ubiquitous in-context learners (Brown et al., 2020; Touvron et al., 2023), but led to a paradigm shift in machine learning in general (Dong et al., 2022). In the domain of tabular data, tabular prior-data fitted networks (TabPFNs) are in-context learners that achieve state-of-the-art classification accuracy on small datasets in combination with minimal prediction time (Hollmann et al., 2022). The central idea of prior fitted networks (PFNs) is to train a transformer model that takes as input a small tabular dataset and directly outputs the class labels for test samples. The training data for such a model is purely synthetic, and sampled from a distribution referred to as the "prior" in the context of PFNs.

### 1.1 Full Bayesian inference in-context

Performing full Bayesian inference can be challenging, even for relatively simple models such as generalized linear models (GLMs; Nelder & Wedderburn, 1972). We use the notion of *full Bayesian inference* for methods yielding potentially complex and high-dimensional posterior distributions—in contrast to, for instance, methods that yield only the posterior predictive, as e.g. in Müller et al. (2021). Full Bayesian inference is thus fundamental for a wide range of applications in Bayesian statistics and probabilistic machine learning, even though it cannot always be achieved. Methods of central importance for full Bayesian inference are Markov chain Monte Carlo (MCMC) and variational inference (VI).

While previous studies of ICL are mostly concerned with learning to simulate univariate predictions of models in-context, and only discuss synthetic scenarios (Garg et al., 2022; Ahuja et al., 2023; Bai et al., 2024), our work attempts to perform full Bayesian inference via ICL on real-world data.

The idea underlying the proposed approach is founded on two observations relating to full Bayesian inference and the working principle of PFNs: First, many Bayesian models have a generative formulation that allows the simulation of arbitrarily large amounts of training samples from the joint distribution $P^{\boldsymbol{x},\boldsymbol{z}}$. We assume that samples from $P^{\boldsymbol{x},\boldsymbol{z}}$ comprise a dataset $\boldsymbol{x} = \{\boldsymbol{x}_j\}_{j=1}^{K}$ containing $K$ samples $\boldsymbol{x}_j \in \mathcal{X}$ and a corresponding (latent) variable $\boldsymbol{z} \in \mathcal{Z}$.[1] This joint distribution $P^{\boldsymbol{x},\boldsymbol{z}}$ corresponds to the "prior" in PFNs and allows the training of a large neural network that implicitly learns to perform Bayesian inference. Second, Bayesian inference is especially useful for smaller datasets $\boldsymbol{x}$ that can be processed in a single forward pass. This makes an entire dataset a viable context for Bayesian ICL.

To summarize, our main contributions are as follows:

1. Using the aforementioned idea, we develop, train, and examine a model that yields samples from the posterior distribution $P^{\boldsymbol{z}|\boldsymbol{x}}$ given data $\boldsymbol{x}$ as context without any (explicit) parameter updates or parametric assumption about the posterior.
2. To achieve this, we propose to use synthetic samples from the joint distribution $P^{\boldsymbol{x},\boldsymbol{z}}$ in order to train a large transformer model that performs ICL regarding the posterior $P^{\boldsymbol{z}|\boldsymbol{x}}$, and provide a general framework to analyze the circumstances that enable learning $P^{\boldsymbol{z}|\boldsymbol{x}}$ purely through samples from $P^{\boldsymbol{x},\boldsymbol{z}}$.
3. We then analyze the efficacy of our approach for GLMs and latent factor models. For these applications, we show that including the "prior" used for TabPFNs results in reliably inferring posterior distributions on real-world data.
4. In a variety of experiments, we demonstrate that this approach yields posterior samples that are very similar to those from a Hamiltonian Monte Carlo sampler. Furthermore, we find that the quality of the samples, when compared to various popular VI techniques that do not operate in-context, is preferable.

## 2 RELATED WORK

Beyond the perspective of prior fitted networks, the contribution of this work can be summarized from two additional viewpoints: First, from the perspective of in-context learning, we show that (large) transformer models can not only implement statistical models in terms of their predictions, but also explicitly and in a full probabilistic setting. From the simulation-based inference viewpoint, we demonstrate that sample-based posterior estimation (Dax et al., 2021) can be used for full Bayesian inference in complex scenarios arising in commonly used latent variable models and demonstrate the effectiveness of this approach on real-world datasets.

**In-context learning** ICL is a special case of meta-learning (Hospedales et al., 2021) characterized by using a large pre-trained model in order to learn from a context dataset without explicitly updating task-specific parameters. Several recent lines of work investigate the in-context learning capabilities of transformers (Garg et al., 2022; Ahuja et al., 2023; Wang et al., 2024; Chan et al., 2022). For instance, Garg et al. (2022) show that a model similar to GPT-2 can implicitly implement various interesting function classes in-context. More specifically, the model learns to reproduce the predictions of different statistical models, in particular linear functions, as well as sparse linear functions, decision trees, and even two-layer neural networks. This approach can be extended to multiple families of functions and even mixtures of tasks (Ahuja et al., 2023). However, the results by Garg et al. (2022) and Ahuja et al. (2023) are restricted to rather small problem scales and scalar-valued predictions instead of multivariate posterior distributions. Additionally, the experiments are conducted exclusively on simulated data.

Furthermore, while Xie et al. (2021) explain ICL from a Bayesian perspective, several lines employ ICL as a tool for fundamentally Bayesian tasks such as Bayesian optimization (Müller et al., 2023;

---

[1]We do not assume any specific form of $\boldsymbol{z}$. That is, there can be a single $\boldsymbol{z}_j$ associated with each data point $\boldsymbol{x}_j$ in $\boldsymbol{x}$, but the case where a single "global" $\boldsymbol{z}$ governs the behavior of each $\boldsymbol{x}_j$ in $\boldsymbol{x}$ is equally included in this notation.

Ramos et al., 2023) or regression with probabilistic predictions based on natural language (Requeima et al., 2024).

**Amortized Inference** Amortized inference is a central paradigm in the field of variational inference (Kingma, 2013; Zhai et al., 2018; Kim et al., 2018; Margossian & Blei, 2023). The central idea here is to model the posterior distribution $P^{\boldsymbol{z}|\boldsymbol{x}}$ of latent variables $\boldsymbol{z}$ given a dataset $\boldsymbol{x}$ via $p(\boldsymbol{z}|\boldsymbol{x}) \approx \prod_{j=1}^{K} q_\theta(\boldsymbol{z}_j | h_\phi(\boldsymbol{x}_j))$. Here, in contrast to our more general assumption, each datapoint $\boldsymbol{x}_j$ in $\boldsymbol{x}$ has a corresponding latent variable $\boldsymbol{z}_j$. While the parameter $\theta$ determines global aspects of the variational distribution, the function $h_\phi$ is shared for all $\boldsymbol{x}_j$ and thus amortized across a dataset $\boldsymbol{x}$. For example, variational autoencoders (Kingma, 2013; Rezende et al., 2014) and neural processes (Garnelo et al., 2018a;b; Rudner et al., 2018) are important model classes based on amortized inference.

In contrast, our ICL approach amortizes its parameters on the level of datasets, such that a single functional relationship is learned for a set $\mathcal{D} \subset (\mathcal{X} \times \mathcal{Z})^N$ of datasets. From this point of view, $\mathcal{D} = \left\{ \left( \boldsymbol{x}^{(i)}, \boldsymbol{z}^{(i)} \right) \right\}_{i=1}^{N}$ comprising $N$ datasets $\boldsymbol{x}^{(i)} \in \mathcal{X}$ and the corresponding latent variables $\boldsymbol{z}^{(i)} \in \mathcal{Z}$ can be seen as a "meta-dataset" for which we perform amortized inference. Furthermore, unlike amortized variational inference, we do not use the notion of an evidence lower bound (Blei et al., 2017) or even the Kullback-Leibler divergence to learn the posterior distribution, but rather utilize ideas that also appear in the context of simulation-based inference.

**Simulation-based inference** Analogously to latent variable models, some scientific simulations, for instance in neuroscience or astrophysics (Fan & Markram, 2019; Schmit & Pritchard, 2018), allow to draw samples from the joint distribution $P^{\boldsymbol{x},\boldsymbol{z}}$ of data and latent variable of interest. Amortized posterior inference in this context is referred to as simulation-based inference (SBI; Cranmer et al., 2020). Several recent approaches focus on using neural networks to directly infer aspects of the likelihood $p(\boldsymbol{x}|\boldsymbol{z})$, the posterior $P^{\boldsymbol{z}|\boldsymbol{x}}$ or the joint distribution $P^{\boldsymbol{x},\boldsymbol{z}}$ in the aforementioned simulation cases. More specifically, techniques based on discrete normalizing flows (Dax et al., 2021) or flow-matching (Wildberger et al., 2024) are used to approximate the posterior $P^{\boldsymbol{z}|\boldsymbol{x}}$, while Gloeckler et al. (2024) propose to use a transformer-based diffusion model in order to approximate the joint distribution $P^{\boldsymbol{x},\boldsymbol{z}}$.

## 3 IN-CONTEXT LEARNING FOR FULL BAYESIAN INFERENCE

Bayesian inference is a tool of central importance for countless applications. However, exact posterior inference can become computationally expensive when using sampling-based methods (Hastings, 1970; Hoffman et al., 2014; Betancourt, 2017) and even impossible when relying on fully factorized VI methods, which can incur substantial approximation errors (Bishop et al., 2002; Blei, 2012; Margossian & Blei, 2023). Amortized variational inference can alleviate those issues but typically requires the development of specialized and complex modeling frameworks (Kingma, 2013; Srivastava & Sutton, 2017; Garnelo et al., 2018b; Lin et al., 2021). Another issue with variational inference arises from having to choose a variational distribution. While insufficient flexibility in this respect can lead to overly simplistic posteriors, a too flexible variational distribution might overfit the given data (Cremer et al., 2018).

We propose a simple and effective solution based on ideas from ICL, which can be seen as conducting amortized inference on a dataset level. Training a model on a potentially unlimited amount of synthetic datasets yields an in-context learner that can not only approximate a vast, almost arbitrarily large, class of distributions, but is also highly efficient when used for sampling. Furthermore, this does not incur any issues with overly or insufficiently flexible distribution assumptions as in VI.

More specifically, our central goal is to develop a method allowing to infer the posterior distribution $P^{\boldsymbol{z}|\boldsymbol{x}}$ of latent variables $\boldsymbol{z} \in \mathcal{Z}$, given observations $\boldsymbol{x} \in \mathcal{X}$ using ICL. From a supervised-learning perspective, we thus aim to directly learn the mapping $f_0 : \mathcal{X} \to \mathcal{M}(\mathcal{Z}), \boldsymbol{x} \mapsto P^{\boldsymbol{z}|\boldsymbol{x}}$, where $\mathcal{M}(\mathcal{Z})$ is the space of all probability measures. Therefore, we want a model $f_\theta(\boldsymbol{x}) = Q_\theta^{\boldsymbol{z}|\boldsymbol{x}}$ for the posterior to be as close as possible to the true posterior $P^{\boldsymbol{z}|\boldsymbol{x}} = f_0(\boldsymbol{x})$. We measure "closeness" w.r.t. some divergence $d : \mathcal{M}(\mathcal{Z}) \times \mathcal{M}(\mathcal{Z}) \to [0, \infty)$. When considering the expected divergence over data

samples $\boldsymbol{x} \sim P^{\boldsymbol{x}}$, this gives rise to the following objective:

$$\mathcal{R}_\theta := \mathbb{E}_{\boldsymbol{x} \sim p(\boldsymbol{x})} \left[ d \left( f_\theta(\boldsymbol{x}), f_0(\boldsymbol{x}) \right) \right] = \mathbb{E}_{\boldsymbol{x} \sim p(\boldsymbol{x})} \left[ d \left( Q_\theta^{\boldsymbol{z}|\boldsymbol{x}}, P^{\boldsymbol{z}|\boldsymbol{x}} \right) \right]. \tag{1}$$

Note that we use the notion of a divergence $d$ loosely to refer to any measure of similarity of two distributions. Although $\mathcal{R}_\theta$ itself is usually intractable, specific choices of $d$ and the use of the joint distribution $P^{\boldsymbol{x},\boldsymbol{z}}$ make Eq. (1) accessible via

$$\widetilde{\mathcal{R}}_\theta := \mathbb{E}_{\boldsymbol{x},\boldsymbol{z} \sim p(\boldsymbol{x},\boldsymbol{z})} \left[ \mathcal{L}_d(\boldsymbol{x}, \boldsymbol{z}, \theta) \right], \tag{2}$$

where the loss function $\mathcal{L}_d$ depends on $d$ and the structure of $Q_\theta^{\boldsymbol{z}|\boldsymbol{x}}$ (discussed in detail later). Performing empirical risk minimization for $\widetilde{\mathcal{R}}_\theta$ with samples from the joint distribution $P^{\boldsymbol{x},\boldsymbol{z}}$ then corresponds to learning to approximate $P^{\boldsymbol{z}|\boldsymbol{x}}$. The model for the posterior $P^{\boldsymbol{z}|\boldsymbol{x}}$ is thereby only implicitly defined by the joint distribution $P^{\boldsymbol{x},\boldsymbol{z}}$. While this requires the ability to sample from $P^{\boldsymbol{x},\boldsymbol{z}}$, drawing samples from the joint distribution is often a weak requirement in terms of model specification that immediately follows from specifying the generative process of a model. Furthermore, a simple sufficient condition that follows directly from the law of total expectation implies the equivalence of $\mathcal{R}_\theta$ and $\widetilde{\mathcal{R}}_\theta$:

**Proposition 1.** *Let* $d(Q_\theta^{\boldsymbol{z}|\boldsymbol{x}}, P^{\boldsymbol{z}|\boldsymbol{x}}) = \int \gamma \left( Q_\theta^{\boldsymbol{z}|\boldsymbol{x}} \right) dP^{\boldsymbol{z}|\boldsymbol{x}}$ *for some measurable functional* $\gamma :$ $\mathcal{M}(\mathcal{Z}) \to \mathbb{R}$. *Then* $\mathcal{R}_\theta = \widetilde{\mathcal{R}}_\theta$ *with* $\mathcal{L}_d(\boldsymbol{x}, \boldsymbol{z}, \theta) = \gamma \left( Q_\theta^{\boldsymbol{z}|\boldsymbol{x}} \right)$.

For instance, choosing $d$ to be the forward Kullback-Leibler divergence $d_{KLD}(Q_\theta^{\boldsymbol{z}|\boldsymbol{x}}, P^{\boldsymbol{z}|\boldsymbol{x}}) = D_{KL} \left[ p(\cdot|\boldsymbol{x}) || q_\theta(\cdot|\boldsymbol{x}) \right]$ implies that $\mathcal{L}_{d_{KLD}}(\boldsymbol{x}, \boldsymbol{z}, \theta) = -\log q_\theta(\boldsymbol{z}|\boldsymbol{x}) + const.$ (Müller et al., 2021). In this case, minimizing $\widetilde{\mathcal{R}}_\theta$ thus directly corresponds to performing maximum likelihood inference on samples from $P^{\boldsymbol{x},\boldsymbol{z}}$.

### 3.1 DEFINING THE FORM OF THE POSTERIOR

To learn the posterior distribution $P^{\boldsymbol{z}|\boldsymbol{x}}$ in-context, we use the framework of flow matching (Lipman et al., 2022). More specifically, we utilize continuous normalizing flows (CNFs) to specify and ultimately sample from $P^{\boldsymbol{z}|\boldsymbol{x}}$. CNFs, currently excelling in the field of image synthesis (Esser et al., 2024), do not only allow to flexibly learn almost arbitrary distributions, but are also found to be more sample-efficient in training than for instance diffusion objectives (Lipman et al., 2022; Wildberger et al., 2024). Furthermore, unlike discrete normalizing flows (Papamakarios et al., 2021a), CNF objectives do not limit the architecture of the used neural network, allowing to incorporate complex conditioning on the data $\boldsymbol{x}$ in addition to flexibly modeling the posterior, which is a crucial aspect of our ICL framework. Refer to Appendix K for more information on CNFs.

#### 3.1.1 NORMALIZING FLOWS

The key idea of modeling a distribution $P^{\boldsymbol{z}|\boldsymbol{x}}$ with normalizing flows (see, e.g., Papamakarios et al., 2021b), which are the basis of CNFs, is to assume that $P^{\boldsymbol{z}|\boldsymbol{x}}$ is the result of "pushing forward" a simple base distribution $P_\mathcal{B}$ into $P^{\boldsymbol{z}|\boldsymbol{x}}$ using a conditional flow $\psi_\theta(\cdot|\boldsymbol{x})$:

$$P^{\boldsymbol{z}|\boldsymbol{x}} \approx [\psi_\theta(\cdot|\boldsymbol{x})]_\sharp P_\mathcal{B}. \tag{3}$$

Therefore, one assumes that samples from $P^{\boldsymbol{z}|\boldsymbol{x}}$ are generated by first drawing $\boldsymbol{z}_0 \sim P_\mathcal{B}$, and then applying $\psi_\theta(\cdot|\boldsymbol{x})$, such that $\psi_\theta(\boldsymbol{z}_0|\boldsymbol{x}) \sim P^{\boldsymbol{z}|\boldsymbol{x}}$. The base distribution $P_\mathcal{B}$ is commonly set to be a standard normal distribution, i.e., $P_\mathcal{B} = \mathcal{N}(0, I)$. The conditional flow $\psi_\theta(\cdot|\boldsymbol{x})$ is the object to be learned, such that our model of $P^{\boldsymbol{z}|\boldsymbol{x}}$ is defined as $Q_\theta^{\boldsymbol{z}|\boldsymbol{x}} := [\psi_\theta(\cdot|\boldsymbol{x})]_\sharp P_\mathcal{B}$.

#### 3.1.2 CONTINUOUS NORMALIZING FLOWS

In flow matching (Lipman et al., 2022), which we will use to obtain an in-context learner for full Bayesian inference, the normalizing flow $\psi_\theta(\cdot|\boldsymbol{x})$ is implicitly defined via a (conditional) vector field $v_{t,\boldsymbol{x}}^\theta$ of an ordinary differential equation (ODE):

$$\frac{d}{dt} \psi_{\theta,t}(\boldsymbol{z}|\boldsymbol{x}) = v_{t,\boldsymbol{x}}^\theta(\psi_{\theta,t}(\boldsymbol{z}|\boldsymbol{x})), \qquad \psi_{\theta,0}(\boldsymbol{z}|\boldsymbol{x}) = \boldsymbol{z}, \tag{4}$$

where $0 \leq t \leq 1$. The first condition $\frac{d}{dt}\psi_{\theta,t}(\boldsymbol{z}|\boldsymbol{x}) = v_{t,\boldsymbol{x}}^{\theta}(\psi_{\theta,t}(\boldsymbol{z}|\boldsymbol{x}))$ means that $v_{t,\boldsymbol{x}}^{\theta}$ describes the change in $\psi_{\theta,t}(\boldsymbol{z}|\boldsymbol{x})$ at time $t$, and the second condition $\psi_{\theta,0}(\boldsymbol{z}|\boldsymbol{x}) = \boldsymbol{z}$ implies that initially the flow is just the identity. The family of vector fields $v_{t,\boldsymbol{x}}^{\theta}$ is parameterized by a neural network whose parameters $\theta$ will be learned. In order to ultimately compute the flow $v_{1,\boldsymbol{x}}^{\theta}$, that yields $Q_{\theta}^{\boldsymbol{z}|\boldsymbol{x}} = [\psi_{\theta,1}(\cdot|\boldsymbol{x})]_{\sharp}P_{\mathcal{B}}$, a numerical ODE solver can be used to forward-solve the ODE, which ultimately corresponds to evaluating $\psi_{1,\boldsymbol{x}}$ at a datapoint $\boldsymbol{z}_0 \sim P_{\mathcal{B}}$.

Assuming Gaussian conditional probability paths with an optimal-transport mean- and variance-function (Lipman et al., 2022), one obtains the following discrepancy measure $d_{CFM}$ between $Q_{\theta}^{\boldsymbol{z}|\boldsymbol{x}} := [\psi_{\theta,1}(\cdot|\boldsymbol{x})]_{\sharp}P_{\mathcal{B}}$ and $P^{\boldsymbol{z}|\boldsymbol{x}}$:

$$d_{CFM}\left(Q_{\theta}^{\boldsymbol{z}|\boldsymbol{x}}, P^{\boldsymbol{z}|\boldsymbol{x}}\right) := \mathbb{E}\left[\left\|v_{t,\boldsymbol{x}}^{\theta}((1-(1-\sigma_{min})t)\boldsymbol{z}_0 + t\boldsymbol{z}_1) - \frac{\boldsymbol{z}_1 - (1-\sigma_{min})\boldsymbol{z}_0}{1-(1-\sigma_{min})t}\right\|_2^2\right], \quad (5)$$

where the expectation is taken w.r.t. to three random variables:

1. a uniform time-step $t \sim \mathcal{U}([0, 1])$;

2. samples from the base distribution $\boldsymbol{z}_0 \sim P_{\mathcal{B}}$;

3. samples from the ground-truth conditional distribution $\boldsymbol{z}_1 \sim P^{\boldsymbol{z}|\boldsymbol{x}}$.

We refer to Wildberger et al. (2024) for mathematical results on the relationship of $d_{CFM}$ and the (forward) Kullback-Leibler divergence. The hyperparameter $\sigma_{min}$, which is the variance at time $t = 1$ in the Gaussian conditional probability paths, appears to have negligible influence when set to a sufficiently small value (Lipman et al., 2022).[2]

In order to make optimizing $\mathbb{E}_{\boldsymbol{x}\sim p(\boldsymbol{x})}\left[d_{CFM}\left(Q_{\theta}^{\boldsymbol{z}|\boldsymbol{x}}, P^{\boldsymbol{z}|\boldsymbol{x}}\right)\right]$ tractable, and thus train our in-context learner, we make use of the sufficient condition in Proposition 1. Thus, the divergence $d_{CFM}$ admits the re-formulation as an objective $\widetilde{\mathcal{R}}_{\theta}$ using samples from the joint distribution $P^{\boldsymbol{x},\boldsymbol{z}}$. We can therefore optimize $\widetilde{\mathcal{R}}_{\theta}$ using $N$ independent and identically distributed (i.i.d.) samples $t^{(i)} \sim \mathcal{U}([0, 1])$ from the time-distribution, $\boldsymbol{z}_0^{(i)} \sim P_{\mathcal{B}}$ from the base distribution, and $(\boldsymbol{z}_1^{(i)}, \boldsymbol{x}^{(i)}) \sim P^{\boldsymbol{x},\boldsymbol{z}}$ from the joint distribution. With this, we obtain the following empirical risk used for the training of all ICL models:

$$\hat{\mathcal{R}}_{\theta} = \sum_{i=1}^{N}\left\|v_{t^{(i)},\boldsymbol{x}^{(i)}}^{\theta}((1-(1-\sigma_{min})t^{(i)})\boldsymbol{z}_0^{(i)} + t^{(i)}\boldsymbol{z}_1^{(i)}) - \frac{\boldsymbol{z}_1^{(i)} - (1-\sigma_{min})\boldsymbol{z}_0^{(i)}}{1-(1-\sigma_{min})t^{(i)}}\right\|_2^2. \quad (6)$$

## 3.2 SAMPLING FROM THE JOINT DISTRIBUTION

In order to learn a model that can perform posterior inference according to Section 3.1, we require to sample $(\boldsymbol{x}, \boldsymbol{z}) \sim P^{\boldsymbol{x},\boldsymbol{z}}$. Given $p(\boldsymbol{x}, \boldsymbol{z}) = p(\boldsymbol{x}|\boldsymbol{z})p(\boldsymbol{z})$, this is always possible as long as one can draw samples from $P^{\boldsymbol{z}}$ and then from $P^{\boldsymbol{x}|\boldsymbol{z}}$. Hence, this is a relatively weak requirement allowing for a broad variety of priors and observation models. More specifically, for ICL, we generate a training dataset $\mathcal{D}$ which comprises i.i.d. samples $\left\{\left(\boldsymbol{x}^{(i)}, \boldsymbol{z}^{(i)}\right)\right\}_{i=1}^{N}$ resulting from sampling $\boldsymbol{z}^{(i)} \sim P^{\boldsymbol{z}}$ and then $\boldsymbol{x}^{(i)} \sim P^{\boldsymbol{x}|\boldsymbol{z}^{(i)}}$. We use this simple yet fundamental and very general template to generate samples from the joint $P^{\boldsymbol{x},\boldsymbol{z}}$ for GLMs, factor analysis (FA), and Gaussian mixture models (GMMs) in our later applications.

**GLM example** For example, assume that $\boldsymbol{x} := (\boldsymbol{u}, y)$ is partitioned into covariates $\boldsymbol{u}$ and a response $y$ related via a conditional distribution $P^{y|\boldsymbol{x}}$ depending on a linear predictor $\boldsymbol{u}^{\top}\boldsymbol{\beta}$. This allows to define various GLM structures. In case of a fully Bayesian GLM, one further assumes a prior $P^{\boldsymbol{\beta}}$ on the regression coefficients and additionally on the variance $\sigma^2$ of the responses, which takes the role of a separate dispersion parameter, as well as a link function $g : \mathbb{R} \to \mathbb{R}$. Algorithm 1 specifies how a dataset $\mathcal{D}$ with i.i.d samples from $P^{\boldsymbol{x},\boldsymbol{z}}$ can be sampled in this case.

---

[2]In our experiments, we follow Wildberger et al. (2024) and set $\sigma_{min} := 10^{-4}$ for all experiments.

---

**Algorithm 1:** Generation of synthetic data for GLMs

1   Initialize $\mathcal{D} \leftarrow \varnothing$;
2   **for** $i = 1, \ldots N$ **do**
3      draw $\boldsymbol{\beta}_i \sim P^{\boldsymbol{\beta}}$ ;
4      draw $\sigma_i^2 \sim P^{\sigma^2}$ ;
5      **for** $j = 1, \ldots, K$ **do**
6          draw $\boldsymbol{u}_{i,j} \sim P^{\boldsymbol{u}}$;
7          draw $y_{i,j} \sim p(y|g^{-1}\left(\boldsymbol{u}_{i,j}^{\top}\boldsymbol{\beta}_i\right), \sigma_i^2)$;
8      **end**
9      set $\boldsymbol{x}^{(i)} := ((\boldsymbol{u}_{i,j}, y_{i,j}))_{j=1}^{K}$;
10     set $\boldsymbol{z}^{(i)} := \boldsymbol{\beta}_i$;
11     $\mathcal{D} \leftarrow \mathcal{D} \cup \left\{(\boldsymbol{x}^{(i)}, \boldsymbol{z}^{(i)})\right\}$;
12   **end**

---

Variations in the structure of the distributions $P^{\boldsymbol{\beta}}$, $P^{\sigma^2}$, $P^{y|\boldsymbol{x}}$, as well as $g$ give rise to different models. Examples include Bayesian ridge, Bayesian lasso and logistic regression (Box & Tiao, 2011; Murphy, 2023), which we all consider in our later experiments (see Appendix A.1 for details on the distributional setups and GLMs in general). Analogously, albeit with different data generating mechanisms, one can obtain samples for FA and GMMs, which we detail in Appendix A.2 and Appendix A.3.

### 3.3 GENERATING REALISTIC DATA

While we assume a data-generating process such as the one in Algorithm 1, this is not necessarily the data-generating process that produces the data in the model's application as an in-context learner. Even when the generative process $P^{\boldsymbol{x},\boldsymbol{z}}$ underlying a statistical model is sophisticated and complex in nature, model misspecification is inevitable in almost every practical application. While mismatches between the real data-generating processes and model assumptions can lead to various problems in traditional Bayesian modeling (Grünwald & van Ommen, 2017), the question of model misspecification plays a somewhat different and yet an especially central role for our ICL approach.

More specifically, the ICL model learns the relationship between $P^{\boldsymbol{z}|\boldsymbol{x}}$ and a datapoint $\boldsymbol{x}$ exclusively based on synthetic samples from the marginal $P^{\boldsymbol{x}}$ implied by the statistical model with generative process $P^{\boldsymbol{x},\boldsymbol{z}}$. Given a real-world dataset $\boldsymbol{x}^* \sim P^{\boldsymbol{x}^*}$, model misspecification in terms of $P^{\boldsymbol{x}^*}$ implies that the in-context learner needs to infer the posterior based on out-of-distribution data, where the problem is aggravated the more unrealistic $P^{\boldsymbol{x}}$ is.

To be able to access a reference or ground truth distribution, the data generating processes in our experiments need to match the structure of the GLM, FA and GMM approaches. While the generative processes of FA and GMMs directly prescribe how all parts of the data are generated, this can potentially cause a discrepancy between synthetically generated and real-world datasets. However, our empirical results (Section 4.1) demonstrate that the in-context learner can generalize to real-world data despite the discrepancy to the simulated datasets.

**GLM example continued**   In the aforementioned GLM case, the distribution of the covariates $P^{\boldsymbol{u}}$ does not affect the structure of $P^{\boldsymbol{z}|\boldsymbol{x}}$ in the data generating process (cf. Algorithm 1). We can therefore use a flexible prior $P^{\boldsymbol{u}}$ such as the TabPFN-"prior" (Hollmann et al., 2022) to generate covariates $\boldsymbol{u}$ and thereby effectively tackle the issue of model specification. More specifically, by generating a plethora of highly realistic samples of tabular covariates with different ranges, domains, and correlations, the in-context learner will learn the GLM structure on a broad mixture of distributions regarding the covariates $\boldsymbol{u}$.

### 3.4 THE ARCHITECTURE

In order to implement the idea of learning full Bayesian inference in-context, we extend ideas of diffusion transformers (Peebles & Xie, 2023), where the conditioning on the time $t$ is implemented

via adaptive layer norm (adaLN) blocks initialized as the identity function. As we potentially require complex conditioning on the data $x$, an additional transformer encoder is added. The input to the decoder is a vector in the form $1 - (1 - \sigma_{min})t)z_0 + tz_1$, which is treated as a sequence with length one and processed by a transformer decoder without self-attention, but the adaLN blocks. Therefore, the decoder has an equivalent interpretation as a multi-layer perceptron with skip-connections, cross-attention, and adaptive layer normalization. For the final processing in the decoder, only conditional feedforward layers with adaptive layer normalization are used, which corresponds exactly to the architecture of the decoder before, albeit without cross attention. We call this part an "MLP with Conditioning". Samples for the time $t \in [0, 1]$ are mapped onto a conditioning vector using several fully connected layers, which yields a richer representation of $t$ that is well-suited as an input to the adaLN blocks. Fig. 1 depicts of the resulting architecture.

### 3.5 IMPLEMENTING FLOW MATCHING

During the training phase, a tuple $(z_1, x)$ is drawn from the distribution $P^{z,x}$. Additionally, a time step $t \sim \mathcal{U}[0, 1]$ and a sample $z_0$ is drawn from the base distribution $P_{\mathcal{B}}$, which is a standard Gaussian for all our applications. Subsequently, the ground-truth conditional flow $\psi(z_0|x) = 1 - (1 - \sigma_{min})t)z_0 + tz_1$ is computed, pushing forward $P_{\mathcal{B}}$ into $P^{z|x}$ up to time-point $t$. The transformer encoder processes $x$ and the decoder takes the representation of the encoder into account in order to output $v_{t,x}^{\theta}(\psi(z_0|x))$. This output should match the vector field that describes how the ground-truth flow $\psi(z_0|x)$ continues at time $t$. The discrepancy to the ground-truth vector field is measured with the MSE-loss in Eq. (6).

In the sampling phase, we are given $x$ and the goal is to sample from $P^{z|x}$. To do so, first a

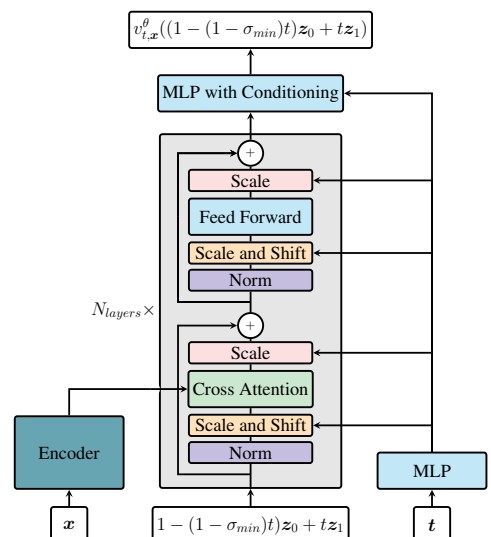

Figure 1: Architecture to perform ICL for full Bayesian inference.

vector $z_0 \sim P_{\mathcal{B}}$ is drawn. The data $x$ is passed through the encoder. The decoder defines a function that maps a time-point $t$ and a vector $\nu$ onto a vector field: $(t, \nu) \mapsto v_{t,x}^{\theta}(\nu)$ taking $x$ into account. This function is given to an ODE-solver in order to forward-solve the corresponding ODE with boundary conditions $0 \le t \le 1$.

## 4 EXPERIMENTS

To show that the proposed methodology is not just an abstract concept, we derive exemplary use cases that demonstrate how well ICL is able to keep up with MCMC and VI approaches in practice.

For this, we will use two prominent statistical modeling classes, namely generalized linear models (GLMs) and latent factor models. For the latent factor models, we consider factor analysis (FA) and Gaussian mixture models (GMMs).

**Modeling scenarios**  We use seven different scenarios for the GLMs, where we vary the prior distribution on the parameters, the conditional distribution of the response, and whether an intercept is included. For FA, we vary the form of the priors and dimensionalities of variables, and for the GMMs investigate different dimensionalities as well as prior configurations. We refer to Appendix A for details on the model structure and scenarios.

**Datasets**  We evaluate the methods on 50 synthetic datasets and 17 real-world datasets from a benchmark suite proposed by Grinsztajn et al. (2022). We refer to Appendix J for more details on the preprocessing of the datasets.

**Methods**   Apart from a comparison with a gold standard, we compare our ICL approach to a Laplace approximation (Daxberger et al., 2021) and different established VI methods based on automatic differentiation VI (Kucukelbir et al., 2017). For the variational distribution, we incorporate a normal distribution with 1) a diagonal and 2) a full covariance matrix, as well as 3) a structured normal distribution with linear dependencies between the latent variables, and 4) an approach based on inverse auto-regressive flows (IAF; Kingma et al., 2016). Appendix D contains a discussion about the hyperparameters of all considered methods.

**Evaluation process**   For every synthetic and real-world dataset, 1000 posterior samples from each method are compared against samples from the analytical solution, if available, or from a Hamiltonian Monte Carlo (HMC) sampler with a NUTS kernel (Hoffman et al., 2014) as the gold standard. For unimodal problems, we run a single chain. For posteriors with multiple modes, we use three times the number of modes as the number of chains to capture multimodality.

**Evaluation metrics**   Three metrics are employed to compare samples from different approximations of the posterior distribution. The first metric is a classifier 2-sample test (C2ST; Lueckmann et al., 2021; Lopez-Paz & Oquab, 2016), where the ROC-AUC score of a random forest classifier, trained to distinguish between samples from the gold standard and the method in question, is utilized. For random forest, we use default hyperparameters, as defined in Scikit-learn (Pedregosa et al., 2011) and 10-fold cross-validation. The second metric is the maximum mean discrepancy (MMD) between the two distributions (gold-standard and each tested method) with an exponential kernel (Gretton et al., 2012). The third metric is the empirical Wasserstein-2 distance ($\mathcal{W}_2$; Givens & Shortt, 1984) of the two distributions, using a quadratic solver implemented in the POT library (Flamary et al., 2021).

## 4.1   Results

### 4.1.1   Generalized linear models

Across seven different variants of GLMs, we find that ICL yields samples that have overall the highest agreement with the gold-standard (see Table 1). Specifically on the synthetic datasets, the C2ST, MMD and $\mathcal{W}_2$ metrics indicate that the posterior distribution can be approximated more accurately with ICL than via variational inference. Particularly in cases where the posterior has a shape deviating from a normal distribution, ICL and HMC agree more closely than VI. For instance in the case where a gamma prior, i.e. a skewed distribution, is used on the coefficients of a regression model, we find that ICL substantially outperforms VI both on synthetic and real-world data (see Table 2). On the real-world data, ICL still matches the performance of VI methods and has the best (or not significantly worse than the best) performance in terms of C2ST in four out of seven cases (see Appendix B.1).

Table 1: Results for GLMs. Average performance of VI methods and our ICL approach on 50 synthetic and 17 real-world datasets across 7 different GLM scenarios. Comparison to the analytical solution when available and HMC otherwise. The best average result is marked in **bold**.

| Model | Synthetic Evaluation | | | Real-World Evaluation | | |
|---|---|---|---|---|---|---|
| | C2ST ($\downarrow$) | MMD ($\downarrow$) | $\mathcal{W}_2$ ($\downarrow$) | C2ST ($\downarrow$) | MMD ($\downarrow$) | $\mathcal{W}_2$ ($\downarrow$) |
| Laplace Approximation | 1.000 | 2.770 | 2.049 | 1.000 | 2.091 | 0.849 |
| VI: DiagonalNormal | 0.869 | 1.586 | 1.742 | 0.819 | 0.583 | 0.529 |
| VI: MultivariateNormal | 0.714 | 1.016 | 1.601 | 0.668 | 0.116 | 0.374 |
| VI: Structured Normal | 0.711 | 0.929 | 1.580 | 0.664 | 0.109 | **0.370** |
| VI: IAF | 0.784 | 1.648 | 2.349 | 0.732 | 0.516 | 0.680 |
| ICL | **0.657** | **0.183** | **0.556** | **0.648** | **0.090** | 0.387 |

### 4.1.2   Factor analysis

On the factor analysis tasks, ICL has notably lower dissimilarity scores compared to the gold standard than all other considered methods in the synthetic evaluation (Table 3). Notably, an average C2ST score of 0.568 is remarkably close to the theoretical lower bound of 0.5. Regarding the real world datasets, C2ST and MMD indicate that our ICL approach yields samples most similar to the reference, while the average $\mathcal{W}_2$ score is substantially higher. We hypothesize that this discrepancy

Table 2: Results for GLMs. Real-world Evaluation on 17 datasets: Linear regression with a gamma prior on the coefficients $\beta$, and an inverse gamma prior on the variance $\sigma^2$ of the responses (scenario 5). Comparison to HMC samples. All results within two standard errors of the best average result are marked in **bold**.

| Model | C2ST ($\downarrow$) | MMD ($\downarrow$) | $\mathcal{W}_2$ ($\downarrow$) |
|---|---|---|---|
| Laplace Approximation | 1.000 ($\pm$ 0.000) | 1.982 ($\pm$ 0.126) | 0.623 ($\pm$ 0.084) |
| VI: DiagonalNormal | 0.810 ($\pm$ 0.036) | 0.441 ($\pm$ 0.252) | 0.384 ($\pm$ 0.089) |
| VI: MultivariateNormal | 0.711 ($\pm$ 0.038) | 0.148 ($\pm$ 0.093) | **0.279** ($\pm$ 0.056) |
| VI: Structured Normal | 0.705 ($\pm$ 0.032) | 0.140 ($\pm$ 0.081) | **0.269** ($\pm$ 0.045) |
| VI: IAF | 0.777 ($\pm$ 0.106) | 0.684 ($\pm$ 0.939) | 0.625 ($\pm$ 0.525) |
| ICL | **0.610** ($\pm$ 0.045) | **0.046** ($\pm$ 0.020) | **0.242** ($\pm$ 0.038) |

in the metrics might be caused by numerical issues when computing the empirical $\mathcal{W}_2$ distance. Furthermore, the relatively high number of latent variables in comparison to the limited number of data-points can yield overly flexible assumptions on the variational posterior causing the VI methods to overfit. While the ICL approach is well suited for cases with little data, the small number of data points is likely the cause for the poor performance of the VI methods on the FA tasks.

Table 3: Results for FA: Average performance of VI methods and our ICL approach on 50 synthetic and 17 real-world datasets across 6 different FA scenarios. Comparison to HMC samples. The best average result is marked in **bold**.

| Model | Synthetic Evaluation | | | Real-World Evaluation | | |
|---|---|---|---|---|---|---|
| | C2ST ($\downarrow$) | MMD ($\downarrow$) | $\mathcal{W}_2$ ($\downarrow$) | C2ST ($\downarrow$) | MMD ($\downarrow$) | $\mathcal{W}_2$ ($\downarrow$) |
| Laplace Approximation | 1.000 | 4.115 | 2.543 | 1.000 | 4.127 | 0.597 |
| VI: DiagonalNormal | 0.999 | 3.321 | 1.998 | 0.960 | 1.220 | 0.288 |
| VI: MultivariateNormal | 0.993 | 3.222 | 1.955 | 0.950 | 1.173 | 0.281 |
| VI: Structured Normal | 0.995 | 3.404 | 2.079 | 0.955 | 1.189 | 0.283 |
| VI: IAF | 0.987 | 3.226 | 1.973 | 0.902 | 0.969 | **0.251** |
| ICL | **0.568** | **0.057** | **0.409** | **0.751** | **0.673** | 0.583 |

### 4.1.3 GAUSSIAN MIXTURE MODELS

Full Bayesian inference for GMMs is arguably much more challenging than for GLMs or FA. First, the generative process of GMMs involves discrete assignments to clusters, which poses a challenge not only for NUTS, but especially for VI methods. Second, the dimensionality of the posterior samples can be relatively large since for diagonal normal distributions, each component of the mixture has a mean and a variance parameter per dimension. Finally, the considered GMMs are not identifiable leading to multi-modal posterior distributions, which are impossible to perfectly approximate with the most commonly used VI methods based on normal approximations.

Due to this inherent difficulty of the GMM scenarios, we find the overall performances of all models to be worse than in the GLM and FA cases. In particular, the C2ST metric is almost saturated for the VI approaches and has a value of around 83 percent for ICL (Table 4). The MMD and $\mathcal{W}_2$ metrics also indicate that ICL yields samples with higher agreement with the reference than the other approaches on synthetic data. A plot of the marginals of the posterior shows high agreement between the posterior distributions of both HMC and ICL while VI is incapable of perfectly approximating a bimodal distribution and exhibits typical mode-seeking behavior (Figure 2). Note that also the VI approach based on inverse autoregressive flows, which in theory allows flexible modeling of a wide range of posterior shapes, fails to learn the bi-modality accurately from the limited number of 50 data points in this GMM scenario. This demonstrates the strength of our ICL approach in flexibly learning distributions agnostic of the provided sample size. On the real-world evaluation, the differences are similar in nature, albeit slightly less pronounced. While C2ST and MMD are better for ICL than for VI, the $\mathcal{W}_2$ metric is not substantially different.

## 5 DISCUSSION

This paper explores in-context learning for the purpose of full Bayesian inference in latent variable models. We propose to use conditional flow matching as a generic and flexible framework to approximate posterior distributions and an architecture that utilizes a transformer encoder for potentially complex conditioning on the data. We find that our ICL approach yields, on average, a closer

Table 4: Average performance of VI methods and our ICL approach on 50 synthetic and 17 real-world datasets across 4 different GMM scenarios. Comparison to HMC samples. The best average result is marked in **bold**.

| Model | Synthetic Evaluation | | | Real-World Evaluation | | |
|---|---|---|---|---|---|---|
| | C2ST ($\downarrow$) | MMD ($\downarrow$) | $\mathcal{W}_2$ ($\downarrow$) | C2ST ($\downarrow$) | MMD ($\downarrow$) | $\mathcal{W}_2$ ($\downarrow$) |
| Laplace Approximation | 1.000 | 3.916 | 8.324 | 1.000 | 3.385 | 12.740 |
| VI: DiagonalNormal | 0.994 | 2.676 | 7.938 | 0.992 | 2.182 | 11.633 |
| VI: MultivariateNormal | 0.995 | 2.556 | 7.947 | 0.987 | 2.143 | 11.696 |
| VI: Structured Normal | 0.994 | 2.595 | 7.929 | 0.988 | 2.129 | 11.521 |
| VI: IAF | 0.985 | 2.308 | 7.489 | 0.957 | 1.845 | 11.541 |
| ICL | **0.825** | **0.706** | **4.348** | **0.881** | **1.051** | **10.691** |

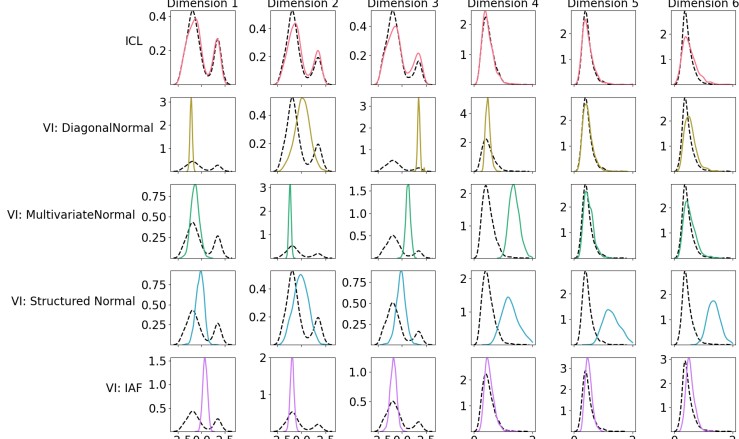

Figure 2: Density plots for the marginals of the posterior for GMM scenario 1. Comparison to HMC samples a on a synthetic dataset. Only the marginals of the first three components of the mean and the variance are shown.

approximation of the posterior than several state-of-the-art variational inference methods across different datasets and model setups. This does not only hold for synthetic data, but also real-world tabular datasets, emphasizing the flexibility of ICL and its applicability for full Bayesian inference.

**Limitations** While our experiments indicate the effectiveness of ICL as a Bayesian inference method, it requires an extensive up-front training routine on modern GPU hardware. Despite ICL being consistently faster at inference time than the considered HMC methods, the overall computational burden to train our approach is much higher. As with many other ICL approaches, large datasets as a context can further become computationally very expensive.

**Outlook and future work** Despite its vast up-front computational cost, ICL has not only proven fundamentally transformative in the field of natural language processing (Brown et al., 2020; Touvron et al., 2023), but recently also appears to be very promising for tabular classification (Hollmann et al., 2022). Exploring the frontiers of ICL in terms of full Bayesian inference, starting from the feasibility results of this work, might therefore yield a path into similarly fertile territories.

Even though our experiments show that ICL works well despite being trained on data that is potentially very different from real-world data, the approach will only be as flexible as the data and model structures it was trained on. As a result, ICL might fail if the model, which implies the synthetic data generation, is severely misspecified. However, this is the same limitation as when misspecifying the hypothesis space of, e.g., a deep neural network or other machine learning approaches, effectively providing the model with the wrong inductive bias.

While flexible state-of-the-art sampling-based methods, such as HMC, are an efficient and highly effective reference in terms of inference for standard and simple statistical methods discussed in this paper, the proposed ICL approach is fundamentally more general in nature. In particular, any probabilistic model for which a generative process is conceivable can be fitted using our ICL approach—the potential for fitting models beyond the horizon of standard Bayesian methods is therefore manifold.

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

APPENDIX

## A    DATA-GENERATING PROCESSSES

This section contains more details on the data generating processes of the latent variable models we fit via ICL.

### A.1    GENERALIZED LINEAR MODELS

In this section we expand the description and explanation regarding GLMs from section 3.2. GLMs are among the most commonly used statistical models with myriads of applications (Nelder & Wedderburn, 1972; Fahrmeir et al., 2013). In the context of GLMs, we assume that the response $y$ follows a distribution $P^{y|u}$ depending on the linear predictor $\eta := u^\top \beta$ and an additional parameter $\sigma^2$. We denote the covariates as $u$, the regression coefficients as $\beta$, and use $\sigma^2$ for the variance of the response. The mean of $P^{y|u}$ depends on the linear predictor via a link function $g$, such that $g\left(\mathbb{E}[y|u]\right) = u^\top \beta$. Ultimately, the density of distribution of the response $y$ depending on the linear predictor and the additional parameter is denoted by $p(y|g\left(u^\top \beta\right), \sigma^2)$. To showcase the flexibility of our framework, we experiment with different priors $P^\beta$ on the regression coefficients, $P^{\sigma^2}$ on the parameter $\sigma^2$, and also different parametric distributions of the response. Additionally, to include covariates $u$ that resemble practically relevant tabular data in the generative process, allowing for meaningful inference on real-world datasets, we utilize samples from the Tab-PFN "prior" for $P^u$.

GLMs belong to the framework of latent variable models defined by data $x$ and (latent) variables $z$, where the data comprises covariates and response $x := (u, y)$. The variables of interest are the coefficients $z := \beta$. This yields the following generative process for a set of synthetic samples $\mathcal{D} := \left\{(x^{(i)}, z^{(i)})\right\}_{i=1}^N$ from $P^{x,z}$:

---

**Algorithm 2:** Generation of synthetic data for GLMs

1  Initialize $\mathcal{D} \leftarrow \emptyset$;
2  **for** $i = 1, \ldots N$ **do**
3  $\quad$ draw $\beta_i \sim P^\beta$ ;
4  $\quad$ draw $\sigma_i^2 \sim P^{\sigma^2}$ ;
5  $\quad$ **for** $j = 1, \ldots, K$ **do**
6  $\quad\quad$ draw $u_{i,j} \sim P^u$;
7  $\quad\quad$ draw $y_{i,j} \sim p(y|g^{-1}\left(u_{i,j}^\top \beta_i\right), \sigma_i^2)$;
8  $\quad$ **end**
9  $\quad$ set $x^{(i)} := ((u_{i,j}, y_{i,j}))_{j=1}^K$;
10 $\quad$ set $z^{(i)} := \beta_i$;
11 $\quad$ $\mathcal{D} \leftarrow \mathcal{D} \cup \left\{(x^{(i)}, z^{(i)})\right\}$;
12 **end**

---

We consider seven different GLM scenarios by varying the structure of the prior distributions and the conditional distribution of the response (Table 5). In particular, we consider a normal $\mathcal{N}(0, 1)$ prior, a Laplace$(0, 1)$ and a gamma Ga$(1, 1)$ prior that factorizes over the coefficients $\beta_j$ contained in $\beta = (\beta_1, \ldots, \beta_p)$. In two cases we include an intercept in the model using a normal prior $\mathcal{N}(0, 9)$ with a relatively large variance. We consider regression cases with a normally distributed response $\mathcal{N}(u^\top \beta, \sigma^2)$, a Bernoulli distributed response Bin$(1, \text{sigmoid}(u^\top \beta))$, i.e. logistic regression, and a response following a gamma distribution Ga$(\sigma^{-2} \exp\left(u^\top \beta\right), \sigma^{-2} \exp(2u^\top \beta))$. In the last case, we set $\exp(u^\top \beta)$ to be the mean and $\sigma^2$ to be the conditional variance of the response. An inverse gamma prior IG$(5, 2)$ is used on the variance $\sigma^2$ for each scenario except the logistic regression. We fix the number of covariates and thus also the dimensionality of $\beta$ at $p = 5$ and set the number of data points per dataset to $K = 50$.

Table 5: Distribution of variables for the considered GLM scenarios.

| Scenario | $\beta_{i,j}$ | $\beta_{i,0}$ | $\sigma_i^2$ | $y_{i,j}\|(\boldsymbol{u}_{i,j}, \boldsymbol{\beta}_i, \beta_{0,i}, \sigma_i^2)$ |
|---|---|---|---|---|
| Scenario 1 | $\mathcal{N}(0,1)$ | - | IG(5,2) | $\mathcal{N}(\boldsymbol{u}_{i,j}^\top \boldsymbol{\beta}_i, \sigma_i^2)$ |
| Scenario 2 | $\mathcal{N}(0,1)$ | $\mathcal{N}(0,9)$ | IG(5,2) | $\mathcal{N}(\boldsymbol{u}_{i,j}^\top \boldsymbol{\beta}_i, \sigma_i^2)$ |
| Scenario 3 | Laplace$(0,1)$ | - | IG(5,2) | $\mathcal{N}(\boldsymbol{u}_{i,j}^\top \boldsymbol{\beta}_i, \sigma_i^2)$ |
| Scenario 4 | Laplace$(0,1)$ | $\mathcal{N}(0,9)$ | IG(5,2) | $\mathcal{N}(\boldsymbol{u}_{i,j}^\top \boldsymbol{\beta}_i, \sigma_i^2)$ |
| Scenario 5 | Ga$(1,1)$ | - | IG(5,2) | $\mathcal{N}(\boldsymbol{u}_{i,j}^\top \boldsymbol{\beta}_i, \sigma_i^2)$ |
| Scenario 6 | $\mathcal{N}(0,1)$ | - | - | Bin$(1, \text{sigmoid}(\boldsymbol{u}_{i,j}^\top \boldsymbol{\beta}_i))$ |
| Scenario 7 | $\mathcal{N}(0,1)$ | - | IG(5,2) | Ga$(\sigma_i^{-2} \exp(\boldsymbol{u}_{i,j}^\top \boldsymbol{\beta}_i), \sigma_i^{-2} \exp(2\boldsymbol{u}_{i,j}^\top \boldsymbol{\beta}_i))$ |

## A.2 FACTOR ANALYSIS

The goal of factor analysis is to explain data $\boldsymbol{x}$ in terms of latent, typically lower-dimensional, factors $\boldsymbol{z}$ (Lawley & Maxwell, 1962; Rummel, 1988). In the Bayesian setting, one assumes a prior $P^{\boldsymbol{z}}$ on the latent variable $\boldsymbol{z}$, a prior $P^{\boldsymbol{W}}$ on the factor loading matrix $\boldsymbol{W}$ and additional priors $P^{\boldsymbol{\Psi}}$ and $P^{\boldsymbol{\mu}}$ on the covariance matrix and the mean vector. The conditional distribution $P^{\boldsymbol{z}|\boldsymbol{x}}$ of the data given $\boldsymbol{z}$ has mean $\mathbb{E}[\boldsymbol{z}|\boldsymbol{x}] = \boldsymbol{W}\boldsymbol{z} + \boldsymbol{\mu}$ and covariance matrix $\text{Cov}[\boldsymbol{z}|\boldsymbol{x}] = \boldsymbol{\Psi}$. In the case where $P^{\boldsymbol{z}}$ and $P^{\boldsymbol{z}|\boldsymbol{x}}$ are Gaussian, one can set $P^{\boldsymbol{z}} = \mathcal{N}(\boldsymbol{0}, I)$ and assume a diagonal covariance matrix $\boldsymbol{\Psi}$ without loosing expressiveness of the model (Murphy, 2023). We make the assumption that $\boldsymbol{W}$ is lower triangular with positive entries on the diagonal in order to ensure identifiability of the model (Lopes & West, 2004). Additionally, we assume that the distributions $\boldsymbol{\mu}$, $\boldsymbol{\Psi}$ and $P^{\boldsymbol{W}}$ fully factorize. In order to ensure that the diagonal of $\boldsymbol{W}$ is positive, we consider absolute values in the generative process. Algorithm 3 details the data generating process.

Table 6 summarizes the different configurations for FA. We assume a Gaussian prior on the mean components, and an inverse gamma prior on the elements of the diagonal covariance matrix $\boldsymbol{\Psi}$. For the factor loading matrix $\boldsymbol{W}$, independent normal and Laplace priors are investigated. Furthermore, we use a normal prior on the latent factors $\boldsymbol{z}^{(i)}$ in five cases and a Laplace prior in one case. We vary the number of samples $K$ per dataset $\boldsymbol{x}$, the dimensionality $P$ of each data point, as well as the dimensionality $\boldsymbol{z}_{dim}$.

---

**Algorithm 3:** Generation of synthetic data for FA

1 Initialize $\mathcal{D} \leftarrow \varnothing$;
2 **for** $i = 1, \ldots N$ **do**
3 $\quad$ draw $\boldsymbol{\mu}_i \sim P^{\boldsymbol{\mu}}$ ;
4 $\quad$ draw $\boldsymbol{\Psi}_i \sim P^{\boldsymbol{\Psi}}$ ;
5 $\quad$ draw $\boldsymbol{W}_i \sim P^{\boldsymbol{W}}$ ;
6 $\quad$ draw $\boldsymbol{z}^{(i)} \sim P^{\boldsymbol{z}}$ ;
7 $\quad$ **for** $j = 1, \ldots, K$ **do**
8 $\quad\quad$ draw $\boldsymbol{x}_{i,j} \sim \mathcal{N}(\boldsymbol{W}_i \boldsymbol{z}^{(i)} + \boldsymbol{\mu}_i, \boldsymbol{\Psi}_i)$;
9 $\quad$ **end**
10 $\quad$ $\mathcal{D} \leftarrow \mathcal{D} \cup \{(\boldsymbol{x}^{(i)}, \boldsymbol{z}^{(i)})\}$;
11 **end**

---

Table 6: Distribution and dimensionalitites of variables for the considered FA scenarios.

| Scenario | $K$ | $P$ | $\mu_{i,j}$ | $\Psi_{i,j,j}$ | $W_{i,j,k}$ | $z_{i,j}$ | $\boldsymbol{z}_{dim}$ |
|---|---|---|---|---|---|---|---|
| Scenario 1 | 50 | 3 | $\mathcal{N}(0,1)$ | IG(5,1) | $\mathcal{N}(0,1)$ | $\mathcal{N}(0,1)$ | 3 |
| Scenario 2 | 50 | 3 | $\mathcal{N}(0,0.1)$ | IG(5,1) | Laplace$(0,10)$ | $\mathcal{N}(0,1)$ | 3 |
| Scenario 3 | 25 | 5 | $\mathcal{N}(0,0.1)$ | IG(5,2) | $\mathcal{N}(0,3)$ | $\mathcal{N}(0,1)$ | 3 |
| Scenario 4 | 25 | 15 | $\mathcal{N}(0,0.1)$ | IG(5,2) | $\mathcal{N}(0,3)$ | $\mathcal{N}(0,1)$ | 5 |
| Scenario 5 | 25 | 5 | $\mathcal{N}(0,0.1)$ | IG(5,2) | Laplace$(0,3)$ | $\mathcal{N}(0,1)$ | 3 |
| Scenario 6 | 25 | 5 | $\mathcal{N}(0,0.1)$ | IG(5,2) | $\mathcal{N}(0,3)$ | Laplace$(0,1)$ | 3 |

### A.3 GAUSSIAN MIXTURE MODELS

In GMMs one assumes that the data of interest is generated by a convex combination of $M$ (multivariate) normal distributions, such that $p(\boldsymbol{x}|\boldsymbol{z}) = \sum_{m=1}^{M} \phi_m p_m(\boldsymbol{x})$, where the probability vector $\boldsymbol{\phi} = (\phi_1, \ldots, \phi_M)$ comprises the mixture weights and $p_m$ denotes the $m$-th mixture component. We consider $p_m$ to take the form of a diagonal Gaussian with mean vector $\boldsymbol{\mu}_m$ and covariance matrix with diagonal elements $\boldsymbol{\sigma}_m^2$. We assume a prior $P^{\boldsymbol{\phi}}$ on $\boldsymbol{\phi}$, a prior $P^{\boldsymbol{\sigma}^2}$ on the variances of each component and a prior $P^{\boldsymbol{\mu}|\boldsymbol{\sigma}^2}$ for the means that depends on the variance of the respective component. More specifically, we assume a symmetric Dirichlet prior on $\boldsymbol{\phi}$ such that $P^{\boldsymbol{\phi}} = \mathrm{Dir}(\alpha_{Dir})$ and an independent inverse gamma distribution as prior on each component $\sigma_m^2$ of $\boldsymbol{\sigma}_m^2$. The prior on each component of $\boldsymbol{\mu}_{i,m} \in \mathbb{R}^L$ is then given by an independent normal distribution $P^{\boldsymbol{\mu}|\sigma_{i,m,l}^2} = \mathcal{N}(0, \lambda\sigma_{i,m,l}^2)$. We use $\omega_{i,j}$ to denote the assignment of datapoint $j$ a component. Algorithm 4 details the data generating process and Table 22 summarizes the different setups regarding the prior distributions.

---

**Algorithm 4:** Generation of synthetic data for a GMM.

1   Initialize $\mathcal{D} \leftarrow \emptyset$;
2   **for** $i = 1, \ldots N$ **do**
3      draw $\boldsymbol{\phi}_i \sim P^{\boldsymbol{\phi}}$ ;
4      **for** $m = 1, \ldots, M$ **do**
5          **for** $l = 1, \ldots, L$ **do**
6              draw $\sigma_{i,m,l}^2 \sim P^{\boldsymbol{\sigma}^2}$ ;
7              draw $\mu_{i,m,l} \sim P^{\boldsymbol{\mu}|\sigma_{i,m,l}^2}$ ;
8          **end**
9      **end**
10     **for** $j = 1, \ldots, K$ **do**
11        draw $\omega_{i,j} \sim \mathrm{Cat}(\boldsymbol{\phi}_i)$;
12        draw $\boldsymbol{x}_{i,j} \sim \mathcal{N}(\boldsymbol{\mu}_{i,\boldsymbol{\omega}_{i,j}}, \boldsymbol{\sigma}_{i,\boldsymbol{\omega}_{i,j}}^2)$;
13     **end**
14     set $\boldsymbol{z}^{(i)} := \left( (\sigma_{i,m,l}^2, \mu_{i,m,l}) \right)_{m,l=1}^{M,L}$ ;
15     $\mathcal{D} \leftarrow \mathcal{D} \cup \left\{ (\boldsymbol{x}^{(i)}, \boldsymbol{z}^{(i)}) \right\}$;
16 **end**

---

Table 7: Distribution and dimensionalitites of variables for the considered GMM scenarios.

| Scenario | $K$ | $M$ | $L$ | $\phi_i$ | $\sigma_{i,m,l}^2$ | $\mu_{i,m,l}|\sigma_{i,m,l}^2$ |
|---|---|---|---|---|---|---|
| Scenario 1 | 50 | 5 | 1 | $\mathrm{Dir}(1)$ | $\mathrm{IG}(5,2)$ | $\mathcal{N}(0, 3\sigma_{i,m,l}^2)$ |
| Scenario 2 | 25 | 3 | 3 | $\mathrm{Dir}(1)$ | $\mathrm{IG}(5,2)$ | $\mathcal{N}(0, 3\sigma_{i,m,l}^2)$ |
| Scenario 3 | 50 | 3 | 5 | $\mathrm{Dir}(0.5)$ | $\mathrm{IG}(5,2)$ | $\mathcal{N}(0, 5\sigma_{i,m,l}^2)$ |
| Scenario 4 | 50 | 3 | 3 | $\mathrm{Dir}(1)$ | $\mathrm{IG}(5,2)$ | $\mathcal{N}(0, 3\sigma_{i,m,l}^2)$ |

## B   DETAILED EXPERIMENTAL RESULTS

### B.1   GENERALIZED LINEAR MODELS

Table 8 contains detailed results regarding the performance of the proposed ICL and the reference VI approaches. In summary, we find that on the synthetic data, our ICL method has the overall best performance, or a performance not significantly[3] worse than that of the best model, with respect to the C2ST metric. More specifically, ICL significantly outperforms all other models in 5 out of seven

---

[3]We refer to a difference that is larger than two standard deviations as "significant".

cases w.r.t. the C2ST and also the MMD metric. While the $\mathcal{W}_2$ metric exhibits a larger variance, it also indicates that on the synthetic data, ICL yields the significantly best result in those 5 cases.

On the real-world data, the differences between ICL and VI are less pronounced, and ICL attains the best average result without any other model within two standard errors in three scenarios in terms of the C2ST metric. ICL is among those models not significantly worse than the best in four cases with respect to the C2ST metric, in six cases in terms of the MMD metric, and also in six cases in terms of $\mathcal{W}_2$.

In scenario 1, which is a linear regression scenario with a normal prior on the coefficients $\beta$ and an inverse gamma prior on the variance $\sigma^2$, ICL and HMC show a similarly large agreement with the analytical solution. Furthermore, the VI approaches with an ordinary multivariate normal distribution, a structured normal distribution as well as the approach based on inverse autoregressive flows also show a large agreement with the analytical solution, which is to be expected since scenario 1 is has a conjugate prior structure yielding a multivariate t-distribution for the posterior of the coefficients (Murphy, 2023).

Scenario 2 and scenario 4 are those where an intercept is included in the generative structure of the GLM. The notably superior performance of the ICL approach in those two cases might be explained by its ability to model distributions with substantially different variances in different dimensions better than VI. Similarly, the posterior in scenario 5 is determined by the gamma prior on the coefficients leading to a (slightly) skewed posterior distribution, which might explain the good relative performance of ICL. See Fig. 3 for a plot of the marginals of the posterior in this scenario on the Miami housing 2016 dataset.

Finally, scenarios 6 and 7 demonstrate the versatility of the ICL method in terms of posterior inference for logistic regression and regression with a gamma response.

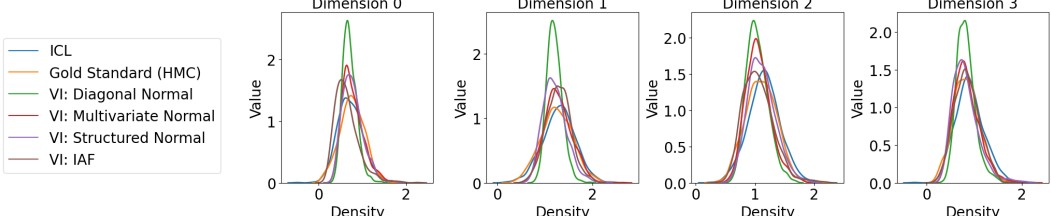

Figure 3: Density plots for first three the marginals of the posterior in a GLM with a gamma prior on the coefficients $\beta$, and an inverse gamma prior on the variance $\sigma^2$ of the responses. The data is part of the Miami housing 2016 dataset.

Table 8: Generalized Linear Models: Evaluation on 50 synthetic and 17 real-world datasets for seven different scenarios. All results within two standard errors of the best average result for each scenario are marked in **bold**.

| Scenario | Model | Synthetic Evaluation | | | Real-World Evaluation | | |
|---|---|---|---|---|---|---|---|
| | | C2ST ($\downarrow$) | MMD ($\downarrow$) | $\mathcal{W}_2$ ($\downarrow$) | C2ST ($\downarrow$) | MMD ($\downarrow$) | $\mathcal{W}_2$ ($\downarrow$) |
| Scenario 1 | Laplace Approximation | 1.000 ($\pm$ 0.000) | 2.738 ($\pm$ 0.721) | **0.825** ($\pm$ 0.279) | 1.000 ($\pm$ 0.000) | 2.150 ($\pm$ 0.323) | **0.642** ($\pm$ 0.124) |
| | VI: DiagonalNormal | 0.904 ($\pm$ 0.076) | 1.452 ($\pm$ 0.984) | **0.669** ($\pm$ 0.301) | 0.797 ($\pm$ 0.083) | 0.612 ($\pm$ 0.511) | **0.414** ($\pm$ 0.152) |
| | VI: MultivariateNormal | **0.750** ($\pm$ 0.128) | **0.735** ($\pm$ 0.733) | **0.565** ($\pm$ 0.292) | **0.607** ($\pm$ 0.070) | **0.167** ($\pm$ 0.196) | **0.301** ($\pm$ 0.123) |
| | VI: Structured Normal | **0.753** ($\pm$ 0.126) | **0.736** ($\pm$ 0.737) | **0.570** ($\pm$ 0.310) | **0.600** ($\pm$ 0.070) | **0.169** ($\pm$ 0.214) | **0.306** ($\pm$ 0.131) |
| | VI: IAF | **0.777** ($\pm$ 0.122) | **0.864** ($\pm$ 0.844) | 0.725 ($\pm$ 0.523) | 0.683 ($\pm$ 0.132) | 0.440 ($\pm$ 0.559) | 0.503 ($\pm$ 0.383) |
| | HMC | **0.745** ($\pm$ 0.130) | **0.722** ($\pm$ 0.732) | **0.569** ($\pm$ 0.301) | **0.595** ($\pm$ 0.075) | **0.173** ($\pm$ 0.213) | **0.321** ($\pm$ 0.140) |
| | **ICL** | **0.765** ($\pm$ 0.123) | **0.767** ($\pm$ 0.727) | **0.585** ($\pm$ 0.301) | **0.614** ($\pm$ 0.074) | **0.175** ($\pm$ 0.219) | **0.310** ($\pm$ 0.138) |
| Scenario 2 | Laplace Approximation | 1.000 ($\pm$ 0.000) | 4.853 ($\pm$ 2.333) | 5.770 ($\pm$ 5.946) | 1.000 ($\pm$ 0.000) | 2.572 ($\pm$ 0.206) | 0.809 ($\pm$ 0.149) |
| | VI: DiagonalNormal | 0.957 ($\pm$ 0.091) | 3.906 ($\pm$ 2.679) | 5.628 ($\pm$ 6.092) | 0.892 ($\pm$ 0.044) | 0.847 ($\pm$ 0.389) | **0.530** ($\pm$ 0.175) |
| | VI: MultivariateNormal | 0.910 ($\pm$ 0.131) | 3.407 ($\pm$ 2.781) | 5.584 ($\pm$ 6.104) | 0.820 ($\pm$ 0.031) | 0.243 ($\pm$ 0.148) | **0.408** ($\pm$ 0.118) |
| | VI: Structured Normal | 0.908 ($\pm$ 0.119) | 3.139 ($\pm$ 2.763) | 5.480 ($\pm$ 6.164) | 0.824 ($\pm$ 0.023) | 0.215 ($\pm$ 0.110) | **0.392** ($\pm$ 0.109) |
| | VI: IAF | 0.968 ($\pm$ 0.063) | 4.416 ($\pm$ 2.473) | 7.474 ($\pm$ 6.235) | 0.888 ($\pm$ 0.067) | 0.921 ($\pm$ 0.860) | 0.942 ($\pm$ 0.733) |
| | **ICL** | **0.839** ($\pm$ 0.072) | **0.707** ($\pm$ 0.658) | **1.111** ($\pm$ 0.300) | **0.768** ($\pm$ 0.033) | **0.143** ($\pm$ 0.089) | **0.411** ($\pm$ 0.094) |
| Scenario 3 | Laplace Approximation | 1.000 ($\pm$ 0.000) | 2.203 ($\pm$ 0.997) | 1.170 ($\pm$ 0.949) | 1.000 ($\pm$ 0.000) | 1.841 ($\pm$ 0.185) | 0.729 ($\pm$ 0.175) |
| | VI: DiagonalNormal | 0.866 ($\pm$ 0.101) | 1.069 ($\pm$ 1.150) | 0.846 ($\pm$ 0.747) | 0.797 ($\pm$ 0.083) | 0.526 ($\pm$ 0.361) | 0.480 ($\pm$ 0.207) |
| | VI: MultivariateNormal | 0.656 ($\pm$ 0.131) | 0.445 ($\pm$ 1.061) | 0.660 ($\pm$ 0.737) | **0.560** ($\pm$ 0.035) | **0.032** ($\pm$ 0.028) | **0.249** ($\pm$ 0.069) |
| | VI: Structured Normal | 0.653 ($\pm$ 0.125) | 0.421 ($\pm$ 0.993) | 0.659 ($\pm$ 0.736) | **0.552** ($\pm$ 0.028) | **0.027** ($\pm$ 0.015) | **0.239** ($\pm$ 0.055) |
| | VI: IAF | 0.751 ($\pm$ 0.148) | 0.939 ($\pm$ 1.349) | 0.964 ($\pm$ 0.924) | 0.673 ($\pm$ 0.141) | 0.399 ($\pm$ 0.543) | 0.563 ($\pm$ 0.433) |
| | **ICL** | **0.611** ($\pm$ 0.070) | **0.089** ($\pm$ 0.114) | **0.423** ($\pm$ 0.348) | 0.576 ($\pm$ 0.027) | **0.037** ($\pm$ 0.026) | **0.257** ($\pm$ 0.044) |
| Scenario 4 | Laplace Approximation | 1.000 ($\pm$ 0.000) | 3.511 ($\pm$ 2.025) | 2.166 ($\pm$ 1.722) | 1.000 ($\pm$ 0.000) | 2.011 ($\pm$ 0.058) | 0.993 ($\pm$ 0.144) |
| | VI: DiagonalNormal | 0.968 ($\pm$ 0.036) | 2.798 ($\pm$ 2.255) | 2.065 ($\pm$ 1.745) | 0.916 ($\pm$ 0.040) | 0.928 ($\pm$ 0.339) | 0.732 ($\pm$ 0.181) |
| | VI: MultivariateNormal | 0.855 ($\pm$ 0.123) | 1.648 ($\pm$ 2.052) | 1.853 ($\pm$ 1.745) | 0.771 ($\pm$ 0.017) | **0.087** ($\pm$ 0.030) | **0.539** ($\pm$ 0.070) |
| | VI: Structured Normal | 0.847 ($\pm$ 0.116) | 1.505 ($\pm$ 1.978) | 1.889 ($\pm$ 1.883) | 0.769 ($\pm$ 0.012) | **0.083** ($\pm$ 0.018) | **0.543** ($\pm$ 0.070) |
| | VI: IAF | 0.942 ($\pm$ 0.077) | 3.029 ($\pm$ 2.210) | 3.554 ($\pm$ 2.715) | 0.833 ($\pm$ 0.069) | 0.636 ($\pm$ 0.756) | 0.978 ($\pm$ 0.600) |
| | **ICL** | **0.753** ($\pm$ 0.049) | **0.171** ($\pm$ 0.153) | **0.631** ($\pm$ 0.294) | **0.762** ($\pm$ 0.015) | **0.105** ($\pm$ 0.046) | **0.597** ($\pm$ 0.104) |
| Scenario 5 | Laplace Approximation | 1.000 ($\pm$ 0.000) | 2.060 ($\pm$ 0.472) | 0.797 ($\pm$ 0.577) | 1.000 ($\pm$ 0.000) | 1.982 ($\pm$ 0.126) | 0.623 ($\pm$ 0.084) |
| | VI: DiagonalNormal | 0.866 ($\pm$ 0.085) | 0.954 ($\pm$ 1.022) | 0.651 ($\pm$ 0.549) | 0.810 ($\pm$ 0.036) | 0.441 ($\pm$ 0.252) | 0.384 ($\pm$ 0.089) |
| | VI: MultivariateNormal | 0.765 ($\pm$ 0.100) | 0.537 ($\pm$ 1.019) | 0.633 ($\pm$ 1.067) | 0.711 ($\pm$ 0.038) | 0.148 ($\pm$ 0.093) | **0.279** ($\pm$ 0.056) |
| | VI: Structured Normal | 0.758 ($\pm$ 0.098) | 0.447 ($\pm$ 0.818) | 0.572 ($\pm$ 0.816) | 0.705 ($\pm$ 0.032) | 0.140 ($\pm$ 0.081) | **0.269** ($\pm$ 0.045) |
| | VI: IAF | 0.814 ($\pm$ 0.105) | 0.953 ($\pm$ 1.165) | 0.881 ($\pm$ 1.067) | 0.777 ($\pm$ 0.106) | 0.684 ($\pm$ 0.939) | 0.625 ($\pm$ 0.525) |
| | **ICL** | **0.621** ($\pm$ 0.063) | **0.067** ($\pm$ 0.080) | **0.299** ($\pm$ 0.195) | **0.610** ($\pm$ 0.045) | **0.046** ($\pm$ 0.020) | **0.242** ($\pm$ 0.038) |
| Scenario 6 | Laplace Approximation | 1.000 ($\pm$ 0.000) | 2.026 ($\pm$ 0.027) | 1.612 ($\pm$ 0.162) | 1.000 ($\pm$ 0.000) | 1.993 ($\pm$ 0.032) | 1.299 ($\pm$ 0.106) |
| | VI: DiagonalNormal | 0.724 ($\pm$ 0.060) | 0.185 ($\pm$ 0.082) | **0.787** ($\pm$ 0.078) | 0.703 ($\pm$ 0.039) | 0.147 ($\pm$ 0.063) | 0.637 ($\pm$ 0.089) |
| | VI: MultivariateNormal | **0.534** ($\pm$ 0.018) | **0.014** ($\pm$ 0.006) | **0.581** ($\pm$ 0.074) | **0.538** ($\pm$ 0.019) | **0.016** ($\pm$ 0.007) | **0.466** ($\pm$ 0.029) |
| | VI: Structured Normal | **0.536** ($\pm$ 0.016) | **0.014** ($\pm$ 0.005) | **0.583** ($\pm$ 0.071) | **0.536** ($\pm$ 0.019) | **0.017** ($\pm$ 0.009) | **0.469** ($\pm$ 0.033) |
| | VI: IAF | 0.542 ($\pm$ 0.026) | 0.031 ($\pm$ 0.031) | 0.613 ($\pm$ 0.092) | **0.535** ($\pm$ 0.015) | **0.015** ($\pm$ 0.006) | **0.467** ($\pm$ 0.031) |
| | **ICL** | **0.532** ($\pm$ 0.019) | 0.016 ($\pm$ 0.008) | **0.590** ($\pm$ 0.066) | 0.556 ($\pm$ 0.017) | 0.035 ($\pm$ 0.015) | **0.504** ($\pm$ 0.038) |
| Scenario 7 | Laplace Approximation | 1.000 ($\pm$ 0.000) | 3.559 ($\pm$ 1.933) | 1.347 ($\pm$ 1.067) | 1.000 ($\pm$ 0.000) | 2.016 ($\pm$ 0.080) | 0.763 ($\pm$ 0.174) |
| | VI: DiagonalNormal | 0.938 ($\pm$ 0.074) | 2.536 ($\pm$ 2.097) | 1.142 ($\pm$ 0.993) | 0.936 ($\pm$ 0.024) | 1.029 ($\pm$ 0.255) | 0.579 ($\pm$ 0.181) |
| | VI: MultivariateNormal | 0.814 ($\pm$ 0.181) | 1.999 ($\pm$ 2.283) | 1.033 ($\pm$ 0.969) | **0.741** ($\pm$ 0.020) | 0.093 ($\pm$ 0.025) | **0.391** ($\pm$ 0.074) |
| | VI: Structured Normal | 0.824 ($\pm$ 0.177) | 1.891 ($\pm$ 2.127) | 1.041 ($\pm$ 0.934) | **0.734** ($\pm$ 0.025) | **0.072** ($\pm$ 0.019) | **0.385** ($\pm$ 0.065) |
| | VI: IAF | 0.939 ($\pm$ 0.091) | 2.707 ($\pm$ 1.712) | 1.590 ($\pm$ 0.820) | 0.864 ($\pm$ 0.093) | 0.830 ($\pm$ 0.697) | 1.064 ($\pm$ 0.616) |
| | **ICL** | **0.700** ($\pm$ 0.116) | **0.317** ($\pm$ 0.355) | **0.400** ($\pm$ 0.286) | 0.773 ($\pm$ 0.048) | 0.294 ($\pm$ 0.457) | 0.559 ($\pm$ 0.256) |

## B.2 FACTOR ANALYSIS

Table 16 contains detailed results regarding FA for 50 synthetic and 17 real-world datasets across 6 different scenarios. We find that overall the ICL method has a very high agreement with the gold standard HMC reference with scores of more than than 56 percent in five scenarios on the synthetic data. In comparison, the C2ST metric is almost saturated for all considered VI methods. For MMD and $\mathcal{W}_2$ the ICL method is again the best.

The real-world datasets show a similar picture except for scenario 4 where C2ST and MMD indicate that VI with inverse autoregressive flows performs best. The $\mathcal{W}_2$ metric, however exhibits a relatively large variance in those cases and does not yield significant results regarding the best performance.

Table 9: Factor Analysis: Evaluation on 50 synthetic and 17 real-world datasets for six different scenarios. All results within two standard errors of the best average result for each scenario are marked in **bold**.

| Scenario | Model | Synthetic Evaluation | | | Real-World Evaluation | | |
|---|---|---|---|---|---|---|---|
| | | C2ST ($\downarrow$) | MMD ($\downarrow$) | $\mathcal{W}_2$ ($\downarrow$) | C2ST ($\downarrow$) | MMD ($\downarrow$) | $\mathcal{W}_2$ ($\downarrow$) |
| Scenario 1 | Laplace Approximation | 1.000 ($\pm$ 0.000) | 3.459 ($\pm$ 1.553) | 1.987 ($\pm$ 1.363) | 1.000 ($\pm$ 0.000) | 2.487 ($\pm$ 0.454) | **0.875** ($\pm$ 0.036) |
| | VI: DiagonalNormal | 1.000 ($\pm$ 0.001) | 4.695 ($\pm$ 1.488) | 2.865 ($\pm$ 1.681) | 0.979 ($\pm$ 0.008) | 1.283 ($\pm$ 0.225) | **0.625** ($\pm$ 0.058) |
| | VI: MultivariateNormal | 0.998 ($\pm$ 0.003) | 4.163 ($\pm$ 1.473) | 2.603 ($\pm$ 1.959) | 0.966 ($\pm$ 0.010) | 1.213 ($\pm$ 0.260) | **0.608** ($\pm$ 0.047) |
| | VI: Structured Normal | 0.997 ($\pm$ 0.004) | 4.655 ($\pm$ 1.189) | 2.700 ($\pm$ 1.333) | 0.979 ($\pm$ 0.010) | 1.231 ($\pm$ 0.132) | **0.611** ($\pm$ 0.041) |
| | VI: IAF | 0.953 ($\pm$ 0.104) | 3.992 ($\pm$ 2.089) | 2.750 ($\pm$ 1.838) | 0.849 ($\pm$ 0.075) | 0.772 ($\pm$ 0.335) | **0.503** ($\pm$ 0.063) |
| | ICL | **0.552** ($\pm$ 0.028) | **0.034** ($\pm$ 0.034) | **0.289** ($\pm$ 0.083) | **0.606** ($\pm$ 0.038) | **0.068** ($\pm$ 0.069) | 0.265 ($\pm$ 0.078) |
| Scenario 2 | Laplace Approximation | 1.000 ($\pm$ 0.000) | 3.687 ($\pm$ 1.661) | 1.954 ($\pm$ 1.129) | 1.000 ($\pm$ 0.000) | 1.690 ($\pm$ 0.182) | **0.598** ($\pm$ 0.058) |
| | VI: DiagonalNormal | 0.998 ($\pm$ 0.002) | 3.135 ($\pm$ 1.482) | 1.629 ($\pm$ 0.938) | 0.975 ($\pm$ 0.010) | 1.156 ($\pm$ 0.068) | **0.496** ($\pm$ 0.052) |
| | VI: MultivariateNormal | 0.989 ($\pm$ 0.009) | 2.945 ($\pm$ 1.019) | 1.482 ($\pm$ 0.683) | 0.951 ($\pm$ 0.025) | 0.764 ($\pm$ 0.053) | **0.421** ($\pm$ 0.052) |
| | VI: Structured Normal | 0.984 ($\pm$ 0.031) | 3.790 ($\pm$ 1.572) | 2.106 ($\pm$ 1.429) | 0.958 ($\pm$ 0.025) | 1.001 ($\pm$ 0.126) | **0.465** ($\pm$ 0.056) |
| | VI: IAF | 0.966 ($\pm$ 0.066) | 3.523 ($\pm$ 1.340) | 2.153 ($\pm$ 0.968) | 0.799 ($\pm$ 0.058) | 0.462 ($\pm$ 0.226) | **0.342** ($\pm$ 0.070) |
| | ICL | **0.542** ($\pm$ 0.006) | **0.017** ($\pm$ 0.006) | **0.244** ($\pm$ 0.033) | **0.622** ($\pm$ 0.032) | **0.098** ($\pm$ 0.039) | **0.287** ($\pm$ 0.046) |
| Scenario 3 | Laplace Approximation | 1.000 ($\pm$ 0.000) | 4.137 ($\pm$ 0.932) | 2.188 ($\pm$ 1.011) | 1.000 ($\pm$ 0.000) | 3.653 ($\pm$ 0.183) | **0.473** ($\pm$ 0.026) |
| | VI: DiagonalNormal | 0.999 ($\pm$ 0.002) | 3.339 ($\pm$ 0.985) | 1.722 ($\pm$ 0.870) | 0.951 ($\pm$ 0.007) | 1.114 ($\pm$ 0.080) | **0.245** ($\pm$ 0.016) |
| | VI: MultivariateNormal | 0.994 ($\pm$ 0.007) | 3.189 ($\pm$ 0.960) | 1.644 ($\pm$ 0.859) | 0.945 ($\pm$ 0.007) | 1.085 ($\pm$ 0.082) | **0.242** ($\pm$ 0.015) |
| | VI: Structured Normal | 0.997 ($\pm$ 0.003) | 3.159 ($\pm$ 0.968) | 1.614 ($\pm$ 0.793) | 0.942 ($\pm$ 0.009) | 1.084 ($\pm$ 0.071) | **0.242** ($\pm$ 0.018) |
| | VI: IAF | 0.990 ($\pm$ 0.011) | 3.145 ($\pm$ 1.203) | 1.705 ($\pm$ 0.990) | 0.928 ($\pm$ 0.015) | 1.022 ($\pm$ 0.093) | **0.235** ($\pm$ 0.018) |
| | ICL | **0.537** ($\pm$ 0.023) | **0.024** ($\pm$ 0.021) | **0.259** ($\pm$ 0.088) | **0.609** ($\pm$ 0.019) | **0.124** ($\pm$ 0.037) | **0.179** ($\pm$ 0.018) |
| Scenario 4 | Laplace Approximation | 1.000 ($\pm$ 0.000) | 4.354 ($\pm$ 0.572) | 3.339 ($\pm$ 0.932) | 1.000 ($\pm$ 0.000) | 6.617 ($\pm$ 0.259) | 0.598 ($\pm$ 0.135) |
| | VI: DiagonalNormal | 1.000 ($\pm$ 0.000) | 3.396 ($\pm$ 0.591) | 2.420 ($\pm$ 0.720) | 0.977 ($\pm$ 0.003) | 1.499 ($\pm$ 0.066) | **0.096** ($\pm$ 0.003) |
| | VI: MultivariateNormal | 0.999 ($\pm$ 0.001) | 3.447 ($\pm$ 0.567) | 2.479 ($\pm$ 0.848) | 0.973 ($\pm$ 0.008) | 1.484 ($\pm$ 0.097) | **0.096** ($\pm$ 0.005) |
| | VI: Structured Normal | 1.000 ($\pm$ 0.000) | 3.421 ($\pm$ 0.610) | 2.481 ($\pm$ 0.884) | 0.973 ($\pm$ 0.007) | 1.474 ($\pm$ 0.078) | **0.095** ($\pm$ 0.004) |
| | VI: IAF | 0.999 ($\pm$ 0.001) | 3.269 ($\pm$ 0.552) | 2.307 ($\pm$ 0.779) | **0.961** ($\pm$ 0.018) | **1.337** ($\pm$ 0.142) | **0.092** ($\pm$ 0.005) |
| | ICL | **0.684** ($\pm$ 0.060) | **0.198** ($\pm$ 0.141) | **0.918** ($\pm$ 0.246) | 0.988 ($\pm$ 0.003) | 1.764 ($\pm$ 0.026) | 1.248 ($\pm$ 0.008) |
| Scenario 5 | Laplace Approximation | 1.000 ($\pm$ 0.000) | 4.456 ($\pm$ 0.785) | 2.608 ($\pm$ 0.946) | 1.000 ($\pm$ 0.000) | 4.559 ($\pm$ 0.494) | 0.663 ($\pm$ 0.127) |
| | VI: DiagonalNormal | 0.999 ($\pm$ 0.002) | 3.520 ($\pm$ 1.073) | 2.012 ($\pm$ 0.886) | 0.944 ($\pm$ 0.010) | 1.007 ($\pm$ 0.129) | **0.261** ($\pm$ 0.036) |
| | VI: MultivariateNormal | 0.995 ($\pm$ 0.007) | 3.472 ($\pm$ 1.021) | 1.982 ($\pm$ 0.814) | 0.930 ($\pm$ 0.017) | 0.964 ($\pm$ 0.111) | **0.255** ($\pm$ 0.038) |
| | VI: Structured Normal | 0.998 ($\pm$ 0.005) | 3.369 ($\pm$ 1.044) | 1.916 ($\pm$ 0.852) | 0.934 ($\pm$ 0.011) | 0.996 ($\pm$ 0.133) | **0.259** ($\pm$ 0.035) |
| | VI: IAF | 0.992 ($\pm$ 0.012) | 3.166 ($\pm$ 0.967) | 1.761 ($\pm$ 0.671) | 0.910 ($\pm$ 0.011) | 0.892 ($\pm$ 0.094) | **0.247** ($\pm$ 0.037) |
| | ICL | **0.535** ($\pm$ 0.016) | **0.021** ($\pm$ 0.011) | **0.279** ($\pm$ 0.060) | **0.886** ($\pm$ 0.017) | 1.207 ($\pm$ 0.101) | **1.002** ($\pm$ 0.042) |
| Scenario 6 | Laplace Approximation | 1.000 ($\pm$ 0.000) | 3.942 ($\pm$ 0.971) | 2.624 ($\pm$ 1.682) | 1.000 ($\pm$ 0.000) | 3.319 ($\pm$ 0.196) | **0.377** ($\pm$ 0.020) |
| | VI: DiagonalNormal | 0.998 ($\pm$ 0.002) | 3.214 ($\pm$ 1.072) | 2.209 ($\pm$ 1.543) | 0.949 ($\pm$ 0.008) | 1.196 ($\pm$ 0.093) | **0.210** ($\pm$ 0.011) |
| | VI: MultivariateNormal | 0.991 ($\pm$ 0.013) | 3.056 ($\pm$ 1.237) | 2.189 ($\pm$ 1.698) | 0.938 ($\pm$ 0.009) | 1.121 ($\pm$ 0.075) | **0.205** ($\pm$ 0.012) |
| | VI: Structured Normal | 0.997 ($\pm$ 0.005) | 3.279 ($\pm$ 1.071) | 2.276 ($\pm$ 1.787) | 0.944 ($\pm$ 0.006) | 1.161 ($\pm$ 0.066) | **0.208** ($\pm$ 0.012) |
| | VI: IAF | 0.989 ($\pm$ 0.029) | 3.027 ($\pm$ 0.910) | 1.936 ($\pm$ 1.060) | 0.865 ($\pm$ 0.027) | 0.822 ($\pm$ 0.106) | **0.179** ($\pm$ 0.015) |
| | ICL | **0.543** ($\pm$ 0.021) | **0.023** ($\pm$ 0.015) | **0.345** ($\pm$ 0.173) | **0.666** ($\pm$ 0.020) | **0.200** ($\pm$ 0.034) | **0.224** ($\pm$ 0.014) |

## B.3 GAUSSIAN MIXTURE MODELS

We summarize the results of the ICL approach and the different VI methods regarding the GMM scenarios in Table 17. First, one can note that on the synthetic data, the ICL approach has a much lower C2ST score for scenario 1 and scenario 2 than the other methods. However, for scenarios 3 and 4, C2ST saturates, or at least almost saturates for all approaches. The MMD metric, however, shows that ICL not only has a high agreement with HMC in scenarios 1 and 2, but that it attains the significantly best result in scenarios 3 and 4 as well. This is supported by the $\mathcal{W}_2$ metric, which has the significantly lowest values for ICL in scenarios 2,3 and 4.

Analogously, on the real-world data, MMD shows that ICL is the best approach in all four scenarios without any other model coming into the two standard-deviation range. While the C2ST score is the lowest in scenario 1 and scenario 2 for ICL, it saturates for cases 3 and 4.

Table 10: Gaussian Mixture Models: Evaluation on 50 synthetic and 17 real-world datasets for six different scenarios. All results within two standard errors of the best average result for each scenario are marked in **bold**.

| Scenario | Model | Synthetic Evaluation | | | Real-World Evaluation | | |
|---|---|---|---|---|---|---|---|
| | | C2ST ($\downarrow$) | MMD ($\downarrow$) | $\mathcal{W}_2$ ($\downarrow$) | C2ST ($\downarrow$) | MMD ($\downarrow$) | $\mathcal{W}_2$ ($\downarrow$) |
| Scenario 1 | Laplace Approximation | 1.000 ($\pm$ 0.000) | 3.367 ($\pm$ 1.030) | **4.341** ($\pm$ 2.018) | 1.000 ($\pm$ 0.000) | 3.374 ($\pm$ 0.941) | **6.440** ($\pm$ 1.994) |
| | VI: DiagonalNormal | 0.988 ($\pm$ 0.013) | 1.175 ($\pm$ 1.189) | **2.961** ($\pm$ 1.669) | 0.995 ($\pm$ 0.006) | 1.919 ($\pm$ 1.217) | **5.145** ($\pm$ 2.489) |
| | VI: MultivariateNormal | 0.988 ($\pm$ 0.013) | 1.135 ($\pm$ 1.149) | **2.926** ($\pm$ 1.651) | 0.994 ($\pm$ 0.007) | 2.007 ($\pm$ 1.367) | **5.379** ($\pm$ 2.845) |
| | VI: Structured Normal | 0.987 ($\pm$ 0.015) | 1.126 ($\pm$ 1.145) | **2.944** ($\pm$ 1.663) | 0.993 ($\pm$ 0.009) | 1.943 ($\pm$ 1.359) | **5.313** ($\pm$ 2.737) |
| | VI: IAF | 0.989 ($\pm$ 0.013) | 1.017 ($\pm$ 1.036) | **3.104** ($\pm$ 1.523) | 0.995 ($\pm$ 0.010) | 1.888 ($\pm$ 1.051) | **5.402** ($\pm$ 2.310) |
| | **ICL** | **0.760** ($\pm$ **0.092**) | **0.303** ($\pm$ 0.548) | **2.095** ($\pm$ 1.692) | **0.847** ($\pm$ 0.082) | **0.486** ($\pm$ 0.623) | **4.054** ($\pm$ 2.782) |
| Scenario 2 | Laplace Approximation | 1.000 ($\pm$ 0.000) | 2.864 ($\pm$ 0.607) | 5.407 ($\pm$ 2.320) | 1.000 ($\pm$ 0.000) | 2.928 ($\pm$ 0.438) | 7.228 ($\pm$ 1.323) |
| | VI: DiagonalNormal | 0.989 ($\pm$ 0.024) | 1.425 ($\pm$ 0.829) | 4.933 ($\pm$ 2.379) | 0.998 ($\pm$ 0.003) | 1.525 ($\pm$ 0.356) | 6.091 ($\pm$ 0.931) |
| | VI: MultivariateNormal | 0.991 ($\pm$ 0.021) | 1.532 ($\pm$ 0.940) | 5.119 ($\pm$ 2.521) | 0.999 ($\pm$ 0.002) | 1.619 ($\pm$ 0.269) | 6.258 ($\pm$ 0.872) |
| | VI: Structured Normal | 0.992 ($\pm$ 0.017) | 1.487 ($\pm$ 0.899) | 5.085 ($\pm$ 2.530) | 0.999 ($\pm$ 0.002) | 1.580 ($\pm$ 0.337) | 6.241 ($\pm$ 0.960) |
| | VI: IAF | 0.992 ($\pm$ 0.021) | 1.319 ($\pm$ 0.854) | 5.265 ($\pm$ 2.534) | 0.998 ($\pm$ 0.004) | 1.256 ($\pm$ 0.320) | 6.201 ($\pm$ 0.892) |
| | **ICL** | **0.812** ($\pm$ **0.061**) | **0.159** ($\pm$ 0.154) | **2.314** ($\pm$ 0.926) | **0.937** ($\pm$ 0.041) | **0.282** ($\pm$ 0.131) | **3.947** ($\pm$ 1.055) |
| Scenario 3 | Laplace Approximation | 1.000 ($\pm$ 0.000) | 3.631 ($\pm$ 1.362) | 16.387 ($\pm$ 19.604) | 1.000 ($\pm$ 0.000) | 3.009 ($\pm$ 0.768) | 37.034 ($\pm$ 7.178) |
| | VI: DiagonalNormal | **0.996** ($\pm$ 0.011) | 2.127 ($\pm$ 1.479) | 16.864 ($\pm$ 19.301) | **0.992** ($\pm$ 0.018) | 2.429 ($\pm$ 0.516) | 35.355 ($\pm$ 6.608) |
| | VI: MultivariateNormal | 0.997 ($\pm$ 0.009) | 2.076 ($\pm$ 1.388) | 16.938 ($\pm$ 19.636) | **0.993** ($\pm$ 0.016) | 2.427 ($\pm$ 0.510) | 35.312 ($\pm$ 6.655) |
| | VI: Structured Normal | **0.995** ($\pm$ 0.017) | 2.049 ($\pm$ 1.462) | 16.723 ($\pm$ 19.093) | **0.993** ($\pm$ 0.016) | 2.301 ($\pm$ 0.549) | 34.217 ($\pm$ 5.461) |
| | VI: IAF | **0.994** ($\pm$ 0.018) | 1.675 ($\pm$ 1.049) | 14.311 ($\pm$ 9.266) | **0.993** ($\pm$ 0.017) | 2.148 ($\pm$ 0.528) | 34.336 ($\pm$ 5.398) |
| | **ICL** | 1.000 ($\pm$ 0.000) | **0.582** ($\pm$ 0.280) | **8.708** ($\pm$ 4.945) | 1.000 ($\pm$ 0.000) | **1.869** ($\pm$ 0.342) | **33.230** ($\pm$ 8.095) |
| Scenario 4 | Laplace Approximation | 1.000 ($\pm$ 0.000) | 6.260 ($\pm$ 1.427) | 13.497 ($\pm$ 29.702) | 1.000 ($\pm$ 0.000) | 5.924 ($\pm$ 1.145) | **12.400** ($\pm$ 4.313) |
| | VI: DiagonalNormal | **1.000** ($\pm$ 0.002) | 3.958 ($\pm$ 1.641) | 12.068 ($\pm$ 21.301) | 1.000 ($\pm$ 0.000) | 3.879 ($\pm$ 1.061) | **11.080** ($\pm$ 3.341) |
| | VI: MultivariateNormal | **1.000** ($\pm$ 0.002) | 3.875 ($\pm$ 1.691) | 12.150 ($\pm$ 22.198) | 1.000 ($\pm$ 0.000) | 3.896 ($\pm$ 1.057) | **11.112** ($\pm$ 3.321) |
| | VI: Structured Normal | **1.000** ($\pm$ 0.001) | 3.661 ($\pm$ 1.717) | 12.195 ($\pm$ 22.874) | **0.996** ($\pm$ 0.016) | 3.822 ($\pm$ 1.302) | **11.368** ($\pm$ 4.216) |
| | VI: IAF | **1.000** ($\pm$ 0.002) | 3.536 ($\pm$ 1.597) | 12.015 ($\pm$ 20.884) | 1.000 ($\pm$ 0.000) | 3.471 ($\pm$ 1.036) | **11.421** ($\pm$ 3.233) |
| | **ICL** | 1.000 ($\pm$ 0.000) | **2.451** ($\pm$ 0.868) | **8.333** ($\pm$ 4.202) | 1.000 ($\pm$ 0.000) | **2.518** ($\pm$ **0.694**) | **11.938** ($\pm$ 2.956) |

# C ABLATION: USING A DIFFUSION OBJECTIVE

To validate choosing the flow matching objective with optimal transport (OT) paths resulting in the objective in equation Eq. (6), we also conduct experiments using a diffusion-objective with variance preserving paths introduced by Song et al. (2020). We choose three selected GLM, FA and GMM scenarios with the same 50 synthetic and 17 real-world datasets for each scenario as in the other benchmarks.

Table 11: GLMs: Comparison of the OT flow matching and the VP diffusion objective on 50 synthetic and 17 real-world datasets for three different scenarios. All results within two standard errors of the best average result for each scenario are marked in **bold**.

| Scenario | Model | Synthetic Evaluation | | | Real-World Evaluation | | |
|---|---|---|---|---|---|---|---|
| | | C2ST ($\downarrow$) | MMD ($\downarrow$) | $\mathcal{W}_2$ ($\downarrow$) | C2ST ($\downarrow$) | MMD ($\downarrow$) | $\mathcal{W}_2$ ($\downarrow$) |
| Scenario 2 | Diffusion paths | 0.961 ($\pm$ 0.040) | **1.525** ($\pm$ 0.777) | 3.354 ($\pm$ 1.333) | 0.961 ($\pm$ 0.016) | 1.347 ($\pm$ 0.365) | 2.025 ($\pm$ 0.270) |
| | **OT paths** | **0.839** ($\pm$ 0.072) | **0.707** ($\pm$ 0.658) | **1.111** ($\pm$ 0.300) | **0.768** ($\pm$ 0.033) | **0.143** ($\pm$ 0.089) | **0.411** ($\pm$ 0.094) |
| Scenario 3 | Diffusion paths | 0.903 ($\pm$ 0.111) | 1.080 ($\pm$ 0.564) | 1.733 ($\pm$ 0.408) | 0.936 ($\pm$ 0.013) | 1.002 ($\pm$ 0.203) | 1.442 ($\pm$ 0.103) |
| | **OT paths** | **0.611** ($\pm$ 0.070) | **0.089** ($\pm$ 0.114) | **0.423** ($\pm$ 0.348) | **0.576** ($\pm$ 0.027) | **0.037** ($\pm$ 0.026) | **0.257** ($\pm$ 0.044) |
| Scenario 5 | Diffusion paths | **0.691** ($\pm$ 0.074) | 0.211 ($\pm$ 0.143) | 0.708 ($\pm$ 0.233) | **0.681** ($\pm$ 0.038) | 0.182 ($\pm$ 0.093) | **0.554** ($\pm$ **0.090**) |
| | **OT paths** | **0.621** ($\pm$ 0.063) | **0.067** ($\pm$ 0.080) | **0.299** ($\pm$ 0.195) | **0.610** ($\pm$ 0.045) | **0.046** ($\pm$ 0.020) | **0.242** ($\pm$ 0.038) |

In summary, the empirical results demonstrate that using the OT paths consistently outperforms the VP diffusion objective across all scenarios for both GLMs and FAs. For GLMs, OT paths achieve significantly lower C2ST values in all scenarios. In Scenario 2, OT paths reduce C2ST from 0.961

to 0.839 on synthetic data and from 0.961 to 0.768 on real-world data. Similarly, in Scenario 3, OT paths achieve substantial improvements, with C2ST dropping from 0.903 to 0.611 on synthetic data and from 0.936 to 0.576 on real-world data. This trend is complemented by consistent improvements in other metrics such as $\mathcal{W}_2$, where OT paths often achieve reductions by over 50%.

Table 12: FA: Comparison of the OT flow matching and the VP diffusion objective on 50 synthetic and 17 real-world datasets for three different scenarios. All results within two standard errors of the best average result for each scenario are marked in **bold**.

| Scenario | Model | Synthetic Evaluation | | | Real-World Evaluation | | |
|---|---|---|---|---|---|---|---|
| | | C2ST ($\downarrow$) | MMD ($\downarrow$) | $\mathcal{W}_2$ ($\downarrow$) | C2ST ($\downarrow$) | MMD ($\downarrow$) | $\mathcal{W}_2$ ($\downarrow$) |
| Scenario 1 | Diffusion paths | 0.622 ($\pm$ 0.043) | 0.207 ($\pm$ 0.121) | 0.692 ($\pm$ 0.192) | **0.595** ($\pm$ 0.012) | 0.089 ($\pm$ 0.011) | 0.475 ($\pm$ 0.019) |
| | OT paths | **0.552** ($\pm$ 0.028) | **0.034** ($\pm$ 0.034) | **0.289** ($\pm$ 0.083) | **0.606** ($\pm$ 0.038) | **0.068** ($\pm$ 0.069) | **0.265** ($\pm$ 0.078) |
| Scenario 2 | Diffusion paths | 0.826 ($\pm$ 0.036) | 0.768 ($\pm$ 0.238) | 1.219 ($\pm$ 0.276) | 0.878 ($\pm$ 0.028) | 0.793 ($\pm$ 0.154) | 1.056 ($\pm$ 0.084) |
| | OT paths | **0.542** ($\pm$ 0.006) | **0.017** ($\pm$ 0.006) | **0.244** ($\pm$ 0.033) | **0.622** ($\pm$ 0.032) | **0.098** ($\pm$ 0.039) | **0.287** ($\pm$ 0.046) |
| Scenario 3 | Diffusion paths | 0.751 ($\pm$ 0.048) | 0.387 ($\pm$ 0.216) | 0.834 ($\pm$ 0.163) | 0.944 ($\pm$ 0.008) | 1.514 ($\pm$ 0.056) | 1.332 ($\pm$ 0.028) |
| | OT paths | **0.537** ($\pm$ 0.023) | **0.024** ($\pm$ 0.021) | **0.259** ($\pm$ 0.088) | **0.609** ($\pm$ 0.019) | **0.124** ($\pm$ 0.037) | **0.179** ($\pm$ 0.018) |

For FA, the performance gap in C2ST remains notable. In Scenario 1, OT paths achieve the best results on synthetic data, reducing C2ST from 0.622 to 0.552, while also delivering improvements in $\mathcal{W}_2$ (0.289 compared to 0.692). On real-world datasets, OT paths maintain competitive results, matching or exceeding the performance of diffusion paths. The advantage is even more pronounced in Scenario 2, where OT paths consistently lead across all metrics, with a particularly striking reduction in MMD on synthetic data (0.017 compared to 0.768) and strong results for C2ST on real-world data (0.622 vs. 0.878). Similarly, in Scenario 3, OT paths achieve the lowest C2ST values, with synthetic results improving from 0.751 to 0.537 and real-world results from 0.944 to 0.609.

Table 13: GMMs: Comparison of the OT flow matching and the VP diffusion objective on 50 synthetic and 17 real-world datasets for three different scenarios. All results within two standard errors of the best average result for each scenario are marked in **bold**.

| Scenario | Model | Synthetic Evaluation | | | Real-World Evaluation | | |
|---|---|---|---|---|---|---|---|
| | | C2ST ($\downarrow$) | MMD ($\downarrow$) | $\mathcal{W}_2$ ($\downarrow$) | C2ST ($\downarrow$) | MMD ($\downarrow$) | $\mathcal{W}_2$ ($\downarrow$) |
| Scenario 1 | Diffusion paths | 0.924 ($\pm$ 0.024) | **0.241** ($\pm$ 0.381) | **2.195** ($\pm$ 1.431) | 0.958 ($\pm$ 0.030) | 0.890 ($\pm$ 0.912) | **5.328** ($\pm$ 2.544) |
| | OT paths | **0.760** ($\pm$ 0.092) | **0.303** ($\pm$ 0.548) | **2.095** ($\pm$ 1.692) | **0.847** ($\pm$ 0.082) | **0.486** ($\pm$ 0.623) | **4.054** ($\pm$ 2.782) |
| Scenario 2 | Diffusion paths | 0.942 ($\pm$ 0.020) | **0.213** ($\pm$ 0.187) | **2.748** ($\pm$ 0.659) | **0.984** ($\pm$ 0.012) | **0.411** ($\pm$ 0.162) | **5.397** ($\pm$ 1.458) |
| | OT paths | **0.812** ($\pm$ 0.061) | **0.159** ($\pm$ 0.154) | **2.314** ($\pm$ 0.926) | **0.937** ($\pm$ 0.041) | **0.282** ($\pm$ 0.131) | **3.947** ($\pm$ 1.055) |
| Scenario 3 | Diffusion paths | **1.000** ($\pm$ 0.000) | 0.582 ($\pm$ 0.280) | **8.708** ($\pm$ 4.945) | **1.000** ($\pm$ 0.000) | 1.869 ($\pm$ 0.342) | **33.230** ($\pm$ 8.095) |
| | OT paths | **0.999** ($\pm$ 0.001) | **0.267** ($\pm$ 0.154) | **7.234** ($\pm$ 2.974) | **1.000** ($\pm$ 0.000) | **1.155** ($\pm$ 0.258) | **26.956** ($\pm$ 3.114) |

In the case of Gaussian Mixture Models (GMMs), the empirical results indicate that the OT paths generally outperform the VP diffusion objective across most scenarios and metrics, though the differences are not always statistically significant in pair-wise comparisons. For example, in Scenario 1, OT paths achieve notably better results for C2ST on both synthetic and real-world datasets, with reductions from 0.924 to 0.760 and from 0.958 to 0.847, respectively. Similarly, for $\mathcal{W}_2$, OT paths exhibit better performance on real-world data (4.054 vs. 5.328). In Scenario 2, OT paths maintain a consistent advantage in metrics such as C2ST and $\mathcal{W}_2$. For instance, synthetic data shows a C2ST improvement from 0.942 to 0.812, while real-world data improves from 0.984 to 0.937. The OT paths also achieve lower MMD on synthetic data (0.159 vs. 0.213), supporting their effectiveness in this scenario. For Scenario 3, the differences in performance between OT paths and diffusion paths are more nuanced. OT paths achieve better results for $\mathcal{W}_2$ on both synthetic and real-world data, reducing it from 8.708 to 7.234 and from 33.230 to 26.956, respectively.

## D HYPERPARAMETERS, SOFTWARE AND COMPUTATIONAL SETUP

### D.1 ICL

To ensure maximum comparability across different experiments, we fix the hyperparameters for all ICL experiments: For the architecture of the model introduced in Section 3.4, we use the following configuration: The dimensionality of encoder representations is set to 512 and is expanded to 1024 in the feed-forward blocks. We use 8 heads and 8 encoder layers with a dropout rate of 0.1. For the decoder part we also use 512 as the dimensionality of the representations and 1024 as the intermediate representation in the feed-forward layers and a dropout rate of 0.1. Furthermore, 3 simple fully connected layers with adaLN conditioning are used for final processing in the decoder. For the time conditioning, we use 3 simple fully connected layers to map the scalar-valued time $t$ onto a 512 dimensional conditioning vector that is used for the adaLN blocks in the decoder. This yields a model of around 43.1 million parameters. We use no tokenization for either the encoder or the decoder and simple embedding layers to map the encoder- and decoder-input onto the feed-forward dimensions.

We use an Adam optimizer (Kingma, 2014) with a cosine learning rate schedule (Loshchilov & Hutter, 2016), where the maximum learning rate is $5 \cdot 10^{-4}$, the final division factor is $10^4$ and 10 percent of the epochs are used for warm-up. We use a weight decay parameter of $10^{-5}$ and a batch size of 1024 and gradient clipping with a maximum gradient norm of one. We use in total 75 million synthetic samples for all scenarios. Of the total number, half, i.e. 37.5 million, are used for training and 10 percent for validation and the remaining 40 percent for testing. Note that we observe convergence of the loss usually much earlier than after this training duration, but fix the number of samples for consistency across experiments. A single L4 GPU is used for the GLM scenarios and a single A100 GPU for the FA and GMM cases.

To solve the ODE for the sample generation, dopri5 (Dormand & Prince, 1980) as implemented in Torchdiffeq (Chen, 2018) is used in the adjoint version. We set the relative and absolute tolerance to $10^{-7}$. The $\sigma_{min}$ parameter in the CNF-loss is set to $10^{-4}$.

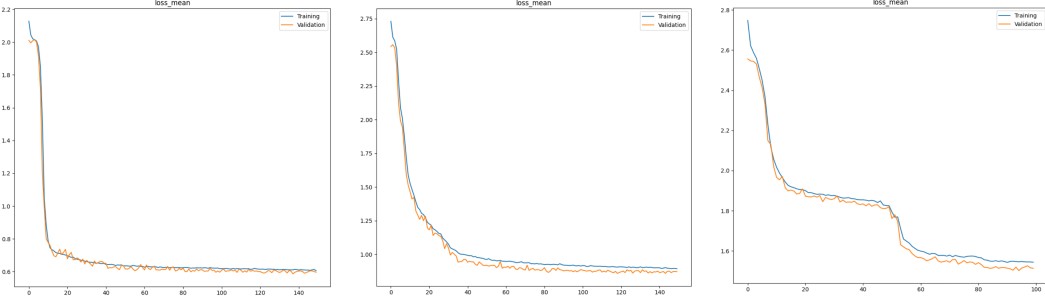

Figure 4: Learning curves for GLM scenario 1 with a Normal Prior on the coefficients $\beta$ and an Inverse Gamma prior on $\sigma_2$.

Figure 5: Learning curves for GMM scenario 1 with $M = 5$ components, $K = 50$ datapoints and $L = 1$ dimensions.

Figure 6: Learning curves for GMM scenario 3 with $M = 3$ components, $K = 50$ datapoints and $L = 5$ dimensions.

### D.2 HMC

We use HMC with a NUTS kernel (Hoffman et al., 2014) as a reference for all experiments where no analytical solution is available. We set the number of burn-in samples to 500 and use one chain for all uni-modal problems and three times the number of potential modes in all other cases. More specifically, we use $M \times 3$ chains for all GMM scenarios. The Pyro implementation of NUTS is used for the GLM scenarios (Bingham et al., 2019) and the conceptually identical, albeit computationally faster implementation in Numpyro for the FA and GMM cases (Phan et al., 2019).

### D.3 VI

For the variational inference methods, we utilize automatic guide generation based on the ground-truth data-generating processes (Kucukelbir et al., 2017). Pyro is used for the implementation of the probabilistic programs, which we also use to sample the synthetic training data, for the automatic guide generation, and for the implementation of the actual VI methods (Bingham et al., 2019). Default hyperparameters, as well as an Adam optimizer (Kingma, 2014) with a learning rate of $10^{-2}$ is used for all methods except for AutoIAF where a learning rate of $10^{-3}$ is used. We perform 2000 full-batch gradient update steps for each method.

## E RUNTIMES

We use a single L4 GPU for generating samples based on our ICL approach and HMC in the GLM scenarios, a single A100 for our ICL approach and HMC in the FA and GMM scenarios, and an Intel(R) Xeon(R) CPU @ 2.20GHz CPU with two virtual cores and 40 gigabytes of RAM for the VI methods. Across all considered GLM scenarios, pre-training takes on average 14.89 hours with a standard error of 18.01 minutes. For the FA scenarios, on average 3.95 hours with a standard error of 11.38 minutes is used for pretraining and for the GMM scenarios 10.63 with a standard error of 72.88 minutes.

When applied in order to generate samples for a new dataset, the benchmarked VI methods have, as expected the lowest runtime. The Laplace approximation is the fastest of all methods, while our ICL appraoch has consistently a lower runtime compared to HMC. Overall, the ICL method takes around 2 minutes on the GLM tasks, around 30 seconds in the FA scenarios and less than 2 minutes for the inference regarding the GMM tasks.

This difference is especially pronounced in the FA and GMM scenarios. Please note that the runtime of the ICL method also fundamentally depends on the used precision for solving the underlying differential equation where we use a relatively high relative and absolute precision of $10^{-7}$. Decreasing this value might lead to significantly faster inference time while maintaining sample quality.

Table 14: Runtime Metrics for all GLM, FA, and GMM Scenarios

| Scenario | Method | Mean Runtime (s) |
|---|---|---|
| GLM | Laplace Approximation | $10.48\,(\pm0.25)$ |
| | VI: DiagonalNormal | $12.02\,(\pm0.26)$ |
| | VI: MultivariateNormal | $13.70\,(\pm0.29)$ |
| | VI: Structured Normal | $19.81\,(\pm0.98)$ |
| | VI:IAF | $15.44\,(\pm0.30)$ |
| | HMC | $120.24\,(\pm13.94)$ |
| | **ICL** | $107.79\,(\pm17.36)$ |
| FA | Laplace Approximation | $17.85\,(\pm0.21)$ |
| | VI: DiagonalNormal | $20.94\,(\pm0.66)$ |
| | VI: MultivariateNormal | $20.84\,(\pm0.28)$ |
| | VI: Structured Normal | $36.17\,(\pm0.61)$ |
| | VI:IAF | $23.75\,(\pm0.38)$ |
| | HMC | $248.26\,(\pm57.88)$ |
| | **ICL** | $31.49\,(\pm4.97)$ |
| GMM | Laplace Approximation | $27.52\,(\pm0.40)$ |
| | VI: DiagonalNormal | $29.74\,(\pm0.57)$ |
| | VI: MultivariateNormal | $30.50\,(\pm0.41)$ |
| | VI: Structured Normal | $42.44\,(\pm0.44)$ |
| | VI:IAF | $33.39\,(\pm0.49)$ |
| | HMC | $239.67\,(\pm32.71)$ |
| | **ICL** | $93.88\,(\pm10.47)$ |

# F  COMPARISON TO SGLD

Besides comparing the samples from our ICL approach to samples from various VI methods, we additionally compare it against samples generated via stochastic gradient Langevin dynamics (SGLD) (Welling & Teh, 2011). We run SGLD with a learning rate of $10^{-3}$ for the GLM and GMM cases and a learning rate of $10^{-4}$ for FA and use 1000 gradient steps for warmup and partition the data into ten minibatches. We implement the preconditioning method introduced by Li et al. (2016) for more stable sampling behavior. Despite the preconditioning, SGLD consistently fails for GLMs scenario 7 because the sampler diverges causing singular covariance matrices. To facilitate running SGLD for the GMMs, which also include discrete variables, we marginalize over the discrete variables.

In summary, we find that ICL yields samples with much higher quality than SGLD compared to the gold standard HMC samples across almost all scenarios on both synthetic and real-world data. The poor sample quality with SGLD is expected given that numerous theoretical and empirical findings confirm that, while SGLD is computationally very cheap, it is substantially outperformed by, for instance, HMC, in terms of sample quality, which is especially pronounced when the posterior distributions are complex and parameters are correlated (Chen et al., 2014; Mangoubi & Vishnoi, 2019; Izmailov et al., 2021; Brosse et al., 2018) .

Table 15: SGLD vs. ICL: Evaluation on 50 synthetic and 17 real-world datasets for six different GLM scenarios. All results within two standard errors of the best average result for each scenario are marked in **bold**.

| Scenario | Model | Synthetic Evaluation | | | Real-World Evaluation | | |
|---|---|---|---|---|---|---|---|
| | | C2ST ($\downarrow$) | MMD ($\downarrow$) | $\mathcal{W}_2$ ($\downarrow$) | C2ST ($\downarrow$) | MMD ($\downarrow$) | $\mathcal{W}_2$ ($\downarrow$) |
| Scenario 1 | SGLD | 0.992 ($\pm$ 0.015) | 2.846 ($\pm$ 1.411) | 1.951 ($\pm$ 0.917) | 0.980 ($\pm$ 0.013) | 2.191 ($\pm$ 1.183) | 0.865 ($\pm$ 0.438) |
| | **ICL** | **0.765** ($\pm$ 0.123) | **0.767** ($\pm$ 0.727) | **0.585** ($\pm$ 0.301) | **0.614** ($\pm$ 0.074) | **0.175** ($\pm$ 0.219) | **0.310** ($\pm$ 0.138) |
| Scenario 2 | SGLD | 0.999 ($\pm$ 0.004) | 5.650 ($\pm$ 1.762) | 8.295 ($\pm$ 5.629) | 0.994 ($\pm$ 0.006) | 2.699 ($\pm$ 1.093) | 1.289 ($\pm$ 0.454) |
| | **ICL** | **0.839** ($\pm$ 0.072) | **0.707** ($\pm$ 0.658) | **1.111** ($\pm$ 0.300) | **0.768** ($\pm$ 0.033) | **0.143** ($\pm$ 0.089) | **0.411** ($\pm$ 0.094) |
| Scenario 3 | SGLD | 0.997 ($\pm$ 0.008) | 3.320 ($\pm$ 1.595) | 3.011 ($\pm$ 1.036) | 0.983 ($\pm$ 0.011) | 2.152 ($\pm$ 1.194) | 0.935 ($\pm$ 0.523) |
| | **ICL** | **0.611** ($\pm$ 0.070) | **0.089** ($\pm$ 0.114) | **0.423** ($\pm$ 0.348) | **0.576** ($\pm$ 0.027) | **0.037** ($\pm$ 0.026) | **0.257** ($\pm$ 0.044) |
| Scenario 4 | SGLD | 1.000 ($\pm$ 0.000) | 6.626 ($\pm$ 1.215) | 15.674 ($\pm$ 8.100) | 0.994 ($\pm$ 0.006) | 2.927 ($\pm$ 1.564) | 1.606 ($\pm$ 1.022) |
| | **ICL** | **0.753** ($\pm$ 0.049) | **0.171** ($\pm$ 0.153) | **0.631** ($\pm$ 0.294) | **0.762** ($\pm$ 0.015) | **0.105** ($\pm$ 0.046) | **0.597** ($\pm$ 0.104) |
| Scenario 5 | SGLD | 0.999 ($\pm$ 0.003) | 3.308 ($\pm$ 1.728) | 2.216 ($\pm$ 1.247) | 1.000 ($\pm$ 0.000) | 4.012 ($\pm$ 1.413) | 0.996 ($\pm$ 0.406) |
| | **ICL** | **0.621** ($\pm$ 0.063) | **0.067** ($\pm$ 0.080) | **0.299** ($\pm$ 0.195) | **0.610** ($\pm$ 0.045) | **0.046** ($\pm$ 0.020) | **0.242** ($\pm$ 0.038) |
| Scenario 6 | SGLD | 0.998 ($\pm$ 0.001) | 2.681 ($\pm$ 0.565) | 2.419 ($\pm$ 0.510) | 0.998 ($\pm$ 0.002) | 2.845 ($\pm$ 0.590) | 1.851 ($\pm$ 0.319) |
| | **ICL** | **0.532** ($\pm$ 0.019) | **0.016** ($\pm$ 0.008) | **0.590** ($\pm$ 0.066) | **0.556** ($\pm$ 0.017) | **0.035** ($\pm$ 0.015) | **0.504** ($\pm$ 0.038) |

For GLMs (Table 15), ICL achieves significantly better results, with notable improvements in C2ST. In Scenario 1, synthetic C2ST drops from 0.992 to 0.765 and real-world C2ST from 0.980 to 0.614. Similarly, Scenario 3 shows substantial gains, with synthetic C2ST improving from 0.997 to 0.611 and real-world C2ST from 0.983 to 0.576. These trends extend to metrics like $\mathcal{W}_2$, where ICL yields consistent reductions, such as in Scenario 2, reducing $\mathcal{W}_2$ from 8.295 to 1.111 on synthetic data.

Table 16: SGLD vs. ICL: Evaluation on 50 synthetic and 17 real-world datasets for six different FA scenarios. All results within two standard errors of the best average result for each scenario are marked in **bold**.

| Scenario | Model | Synthetic Evaluation | | | Real-World Evaluation | | |
|---|---|---|---|---|---|---|---|
| | | C2ST ($\downarrow$) | MMD ($\downarrow$) | $\mathcal{W}_2$ ($\downarrow$) | C2ST ($\downarrow$) | MMD ($\downarrow$) | $\mathcal{W}_2$ ($\downarrow$) |
| Scenario 1 | SGLD | 0.996 ($\pm$ 0.006) | 2.883 ($\pm$ 1.552) | 1.776 ($\pm$ 0.694) | 0.995 ($\pm$ 0.003) | 2.676 ($\pm$ 0.710) | 1.608 ($\pm$ 0.381) |
| | **ICL** | **0.552** ($\pm$ 0.028) | **0.034** ($\pm$ 0.034) | **0.289** ($\pm$ 0.083) | **0.606** ($\pm$ 0.038) | **0.068** ($\pm$ 0.069) | **0.265** ($\pm$ 0.078) |
| Scenario 2 | SGLD | 0.997 ($\pm$ 0.003) | 2.950 ($\pm$ 0.786) | 1.892 ($\pm$ 0.533) | 0.995 ($\pm$ 0.003) | 2.517 ($\pm$ 0.583) | 1.500 ($\pm$ 0.268) |
| | **ICL** | **0.542** ($\pm$ 0.006) | **0.017** ($\pm$ 0.006) | **0.244** ($\pm$ 0.033) | **0.622** ($\pm$ 0.032) | **0.098** ($\pm$ 0.039) | **0.287** ($\pm$ 0.046) |
| Scenario 3 | SGLD | 0.998 ($\pm$ 0.005) | 3.662 ($\pm$ 1.099) | 2.086 ($\pm$ 0.919) | 0.956 ($\pm$ 0.025) | 1.580 ($\pm$ 0.819) | 0.311 ($\pm$ 0.108) |
| | **ICL** | **0.537** ($\pm$ 0.023) | **0.024** ($\pm$ 0.021) | **0.259** ($\pm$ 0.088) | **0.609** ($\pm$ 0.019) | **0.124** ($\pm$ 0.037) | **0.179** ($\pm$ 0.018) |
| Scenario 4 | SGLD | 1.000 ($\pm$ 0.000) | 4.127 ($\pm$ 0.635) | 3.047 ($\pm$ 0.972) | **0.950** ($\pm$ 0.021) | **1.520** ($\pm$ 0.512) | **0.141** ($\pm$ 0.031) |
| | **ICL** | **0.684** ($\pm$ 0.060) | **0.198** ($\pm$ 0.141) | **0.918** ($\pm$ 0.246) | 0.988 ($\pm$ 0.003) | 1.764 ($\pm$ 0.026) | 1.248 ($\pm$ 0.008) |
| Scenario 5 | SGLD | 0.999 ($\pm$ 0.001) | 3.465 ($\pm$ 0.939) | 1.981 ($\pm$ 0.938) | 0.962 ($\pm$ 0.024) | 1.945 ($\pm$ 1.383) | **0.393** ($\pm$ 0.243) |
| | **ICL** | **0.535** ($\pm$ 0.016) | **0.021** ($\pm$ 0.011) | **0.279** ($\pm$ 0.060) | **0.886** ($\pm$ 0.017) | 1.207 ($\pm$ 0.101) | 1.002 ($\pm$ 0.042) |
| Scenario 6 | SGLD | 0.997 ($\pm$ 0.004) | 3.395 ($\pm$ 1.199) | 2.358 ($\pm$ 1.458) | 0.950 ($\pm$ 0.040) | 2.177 ($\pm$ 1.643) | 0.342 ($\pm$ 0.224) |
| | **ICL** | **0.543** ($\pm$ 0.021) | **0.023** ($\pm$ 0.015) | **0.345** ($\pm$ 0.173) | **0.666** ($\pm$ 0.020) | **0.200** ($\pm$ 0.034) | **0.224** ($\pm$ 0.014) |

For FA (Table 16), ICL also achieves superior performance, particularly in Scenarios 1 and 2. For example, in Scenario 1, synthetic C2ST decreases from 0.996 to 0.552, accompanied by improvements in $\mathcal{W}_2$ from 1.776 to 0.289. Scenario 2 sees further enhancements, with synthetic MMD dropping from 2.950 to 0.017 and real-world C2ST improving from 0.995 to 0.622.

Table 17: SGLD vs. ICL: Evaluation on 50 synthetic and 17 real-world datasets for four different GMM scenarios. All results within two standard errors of the best average result for each scenario are marked in **bold**.

| Scenario | Model | Synthetic Evaluation | | | Real-World Evaluation | | |
|---|---|---|---|---|---|---|---|
| | | C2ST ($\downarrow$) | MMD ($\downarrow$) | $\mathcal{W}_2$ ($\downarrow$) | C2ST ($\downarrow$) | MMD ($\downarrow$) | $\mathcal{W}_2$ ($\downarrow$) |
| Scenario 1 | SGLD | 1.000 ($\pm$ 0.001) | 2.629 ($\pm$ 0.868) | 3.279 ($\pm$ 1.330) | 1.000 ($\pm$ 0.000) | 3.421 ($\pm$ 0.877) | 6.510 ($\pm$ 1.763) |
| | ICL | **0.760** ($\pm$ 0.092) | **0.303** ($\pm$ 0.548) | **2.095** ($\pm$ 1.692) | **0.847** ($\pm$ 0.082) | **0.486** ($\pm$ 0.623) | **4.054** ($\pm$ 2.782) |
| Scenario 2 | SGLD | 1.000 ($\pm$ 0.000) | 3.046 ($\pm$ 1.041) | 6.015 ($\pm$ 4.265) | 1.000 ($\pm$ 0.000) | 2.487 ($\pm$ 0.521) | 6.858 ($\pm$ 1.618) |
| | ICL | **0.812** ($\pm$ 0.061) | **0.159** ($\pm$ 0.154) | **2.314** ($\pm$ 0.926) | **0.937** ($\pm$ 0.041) | **0.282** ($\pm$ 0.131) | **3.947** ($\pm$ 1.055) |
| Scenario 3 | SGLD | 1.000 ($\pm$ 0.000) | 4.631 ($\pm$ 1.169) | 23.247 ($\pm$ 30.646) | 1.000 ($\pm$ 0.000) | 2.655 ($\pm$ 0.437) | 26.356 ($\pm$ 2.699) |
| | ICL | **1.000** ($\pm$ 0.000) | **0.582** ($\pm$ 0.280) | **8.708** ($\pm$ 4.945) | **1.000** ($\pm$ 0.000) | **1.869** ($\pm$ 0.342) | **33.230** ($\pm$ 8.095) |
| Scenario 4 | SGLD | **1.000** ($\pm$ 0.000) | 3.464 ($\pm$ 1.098) | **6.995** ($\pm$ 5.554) | **1.000** ($\pm$ 0.000) | **2.555** ($\pm$ 0.494) | **9.477** ($\pm$ 3.432) |
| | ICL | **1.000** ($\pm$ 0.000) | 2.451 ($\pm$ 0.868) | **8.333** ($\pm$ 4.202) | 1.000 ($\pm$ 0.000) | **2.518** ($\pm$ 0.694) | **11.938** ($\pm$ 2.956) |

For GMMs (Table 17), ICL demonstrates a clear advantage in most scenarios. In Scenario 1, ICL reduces synthetic C2ST from 1.000 to 0.760 and real-world $\mathcal{W}_2$ from 6.510 to 4.054. Scenario 2 shows synthetic C2ST improving from 1.000 to 0.812, and MMD from 3.046 to 0.159. While in scenarios 3, ICL has a singificantly lower MMD score on the synthetic data, the other differences are not signigicant.

## G ROBUSTNESS TO OUT-OF-DISTRIBUTION DATA

To investigate how our ICL approach behaves under mismatches between the distribution of synthetic training data and the data used to infer the posterior, we conduct an ablation study by changing aspects of the distribution of training and testing data.

In summary, the results in Tables 19, 21 and 23 show that our ICL approach is, in most cases, capable of robustly generalizing beyond its specific pre-training distribution when various aspects of this distribution are changed. While the performance sometimes decreases when a mismatch between training and testing data occurs, the drops in performance are almost always modest and, in many cases, almost negligible.

### G.1 GLM SCENARIOS

For scenario 2, we change the variance of the prior on the covariates from a value of $\mathbb{V}(\boldsymbol{\beta}_{i,j}) = 1$ to $\mathbb{V}(\boldsymbol{\beta}_{i,j}) = 2$ for scenario 2.B and $\mathbb{V}(\boldsymbol{\beta}_{i,j}) = 4$ for scenario 2.C. In scenarios 2.D and 2.E we change the scale parameter of the prior on the variance $\sigma^2$ of the noise—thereby changing its mean from $\mathbb{E}[\sigma^2] = 0.5$ to a value of $\mathbb{E}[\sigma^2] \approx 0.7071$ for 2.D and $\mathbb{E}[\sigma^2] = 1$ for 2.E. The variance is changed from $\mathbb{V}[\sigma^2] \approx 0.0833$ to $\mathbb{V}[\sigma^2] \approx 0.1667$ and $\mathbb{V}[\sigma^2] \approx 0.333$.

For scenarios 3.B and 3.C, the variance of the coefficients is doubled from scenario 3 to scenario 3.B and from 3.B to 3.C again, analogously to scenarios 2.B and 2.C.0

For scenario 5, the rate parameter of the gamma distribution is changed. This leads to a decrease in the variance from $\mathbb{V}(\boldsymbol{\beta}_{i,j}) = 1$ to $\mathbb{V}(\boldsymbol{\beta}_{i,j}) = 0.5$ for scenario 5.B and $\mathbb{V}(\boldsymbol{\beta}_{i,j}) = 0.25$ for scenario 5.C. Notably, we also change the mean in the distribution of the covariates from mean from $\mathbb{E}[\boldsymbol{\beta}_{i,j}] = 1$ to a value of $\mathbb{E}[\boldsymbol{\beta}_{i,j}] \approx 0.7071$ for 2.D and $\mathbb{E}[\boldsymbol{\beta}_{i,j}] = 0.5$ for 2.E.

Table 18 shows that our ICL approach only exhibits modest degradation in performance when the variance of the coefficients is doubled or quadruple while the mean stays the same (Scenarios 2.B, 2.C and 3.B, 3.C). Increasing the variance of the noise term by a factor of two only has a small effect while multiplying it by four causes a drop in C2ST by 9.3%. However, decreasing the variance of the gamma prior in scenario 5, combined with decreasing the mean, leads to a notable drop in performance across all metrics.

Table 18: Distribution of variables for the OOD analysis on GLM scenarios.

| Scenario | $\beta_{i,j}$ | $\beta_{i,0}$ | $\sigma_i^2$ | $y_{i,j}\|(\boldsymbol{u}_{i,j},\boldsymbol{\beta}_i,\beta_{0,i},\sigma_i^2)$ |
|---|---|---|---|---|
| Scenario 2 | $\mathcal{N}(0,1)$ | $\mathcal{N}(0,9)$ | IG(5,2) | $\mathcal{N}(\boldsymbol{u}_{i,j}^\top\boldsymbol{\beta}_i,\sigma_i^2)$ |
| Scenario 2.B | $\mathcal{N}(0,2)$ | $\mathcal{N}(0,9)$ | IG(5,2) | $\mathcal{N}(\boldsymbol{u}_{i,j}^\top\boldsymbol{\beta}_i,\sigma_i^2)$ |
| Scenario 2.C | $\mathcal{N}(0,4)$ | $\mathcal{N}(0,9)$ | IG(5,2) | $\mathcal{N}(\boldsymbol{u}_{i,j}^\top\boldsymbol{\beta}_i,\sigma_i^2)$ |
| Scenario 2.D | $\mathcal{N}(0,1)$ | $\mathcal{N}(0,9)$ | IG(5,2$\sqrt{2}$) | $\mathcal{N}(\boldsymbol{u}_{i,j}^\top\boldsymbol{\beta}_i,\sigma_i^2)$ |
| Scenario 2.E | $\mathcal{N}(0,1)$ | $\mathcal{N}(0,9)$ | IG(5,4) | $\mathcal{N}(\boldsymbol{u}_{i,j}^\top\boldsymbol{\beta}_i,\sigma_i^2)$ |
| Scenario 3 | Laplace(0,1) | - | IG(5,2) | $\mathcal{N}(\boldsymbol{u}_{i,j}^\top\boldsymbol{\beta}_i,\sigma_i^2)$ |
| Scenario 3.B | Laplace(0,$\sqrt{2}$) | - | IG(5,2) | $\mathcal{N}(\boldsymbol{u}_{i,j}^\top\boldsymbol{\beta}_i,\sigma_i^2)$ |
| Scenario 3.C | Laplace(0,2) | - | IG(5,2) | $\mathcal{N}(\boldsymbol{u}_{i,j}^\top\boldsymbol{\beta}_i,\sigma_i^2)$ |
| Scenario 5 | Ga(1,1) | - | IG(5,2) | $\mathcal{N}(\boldsymbol{u}_{i,j}^\top\boldsymbol{\beta}_i,\sigma_i^2)$ |
| Scenario 5.B | Ga(1,$\sqrt{2}$) | - | IG(5,2) | $\mathcal{N}(\boldsymbol{u}_{i,j}^\top\boldsymbol{\beta}_i,\sigma_i^2)$ |
| Scenario 5.C | Ga(1,2) | - | IG(5,2) | $\mathcal{N}(\boldsymbol{u}_{i,j}^\top\boldsymbol{\beta}_i,\sigma_i^2)$ |

Table 19: OOD Performance: Evaluation on 50 synthetic datasets for 8 different GLM scenarios. All results within two standard errors of the non-OOD result for each scenario are marked in **bold**.

| Scenario | C2ST ($\downarrow$) | MMD ($\downarrow$) | $\mathcal{W}_2$ ($\downarrow$) |
|---|---|---|---|
| Scenario 2 | **0.839** ($\pm$ 0.072) | **0.707** ($\pm$ 0.658) | **1.111** ($\pm$ 0.300) |
| Scenario 2.B | **0.809** ($\pm$ 0.055) | **0.410** ($\pm$ 0.095) | **2.250** ($\pm$ 0.916) |
| Scenario 2.C | **0.857** ($\pm$ 0.105) | **0.634** ($\pm$ 0.318) | **3.067** ($\pm$ 1.759) |
| Scenario 2 | **0.839** ($\pm$ 0.072) | **0.707** ($\pm$ 0.658) | **1.111** ($\pm$ 0.300) |
| Scenario 2.D | **0.840** ($\pm$ 0.109) | **0.916** ($\pm$ 1.123) | **4.007** ($\pm$ 3.261) |
| Scenario 2.E | **0.932** ($\pm$ 0.120) | **1.556** ($\pm$ 1.127) | **4.850** ($\pm$ 2.261) |
| Scenario 3 | **0.611** ($\pm$ 0.070) | **0.089** ($\pm$ 0.114) | **0.423** ($\pm$ 0.348) |
| Scenario 3.B | **0.667** ($\pm$ 0.080) | **0.210** ($\pm$ 0.117) | 1.172 ($\pm$ 0.258) |
| Scenario 3.C | **0.720** ($\pm$ 0.108) | **0.362** ($\pm$ 0.248) | 1.891 ($\pm$ 0.678) |
| Scenario 5 | **0.621** ($\pm$ 0.063) | **0.067** ($\pm$ 0.080) | **0.299** ($\pm$ 0.195) |
| Scenario 5.B | 0.831 ($\pm$ 0.121) | 0.479 ($\pm$ 0.200) | 1.762 ($\pm$ 0.541) |
| Scenario 5.C | 0.920 ($\pm$ 0.064) | 0.753 ($\pm$ 0.424) | 3.159 ($\pm$ 1.254) |

## G.2 FA SCENARIOS

To construct the mismatch between training and test distribution, we vary the variance of the factor loading $W_{i,j,k}$ for scenarios 1, 2 and 3. Concretely, the variance is doubled and quadrupled.

Table 20: Distribution of variables for the OOD analysis on the FA scenarios.

| Scenario | $K$ | $P$ | $\mu_{i,j}$ | $\Psi_{i,j,j}$ | $W_{i,j,k}$ | $z_{i,j}$ | $\boldsymbol{z}_{dim}$ |
|---|---|---|---|---|---|---|---|
| Scenario 1 | 50 | 3 | $\mathcal{N}(0,1)$ | IG(5,1) | $\mathcal{N}(0,1)$ | $\mathcal{N}(0,1)$ | 3 |
| Scenario 1.B | 50 | 3 | $\mathcal{N}(0,1)$ | IG(5,1) | $\mathcal{N}(0,2)$ | $\mathcal{N}(0,1)$ | 3 |
| Scenario 1.C | 50 | 3 | $\mathcal{N}(0,1)$ | IG(5,1) | $\mathcal{N}(0,4)$ | $\mathcal{N}(0,1)$ | 3 |
| Scenario 2 | 50 | 3 | $\mathcal{N}(0,0.1)$ | IG(5,1) | Laplace(0,10) | $\mathcal{N}(0,1)$ | 3 |
| Scenario 2.B | 50 | 3 | $\mathcal{N}(0,0.1)$ | IG(5,1) | Laplace(0,$10\cdot\sqrt{2}$) | $\mathcal{N}(0,1)$ | 3 |
| Scenario 2.C | 50 | 3 | $\mathcal{N}(0,0.1)$ | IG(5,1) | Laplace(0,20) | $\mathcal{N}(0,1)$ | 3 |
| Scenario 3 | 25 | 5 | $\mathcal{N}(0,0.1)$ | IG(5,2) | $\mathcal{N}(0,3)$ | $\mathcal{N}(0,1)$ | 3 |
| Scenario 3 | 25 | 5 | $\mathcal{N}(0,0.1)$ | IG(5,2) | $\mathcal{N}(0,3\cdot\sqrt{2})$ | $\mathcal{N}(0,1)$ | 3 |
| Scenario 3 | 25 | 5 | $\mathcal{N}(0,0.1)$ | IG(5,2) | $\mathcal{N}(0,6)$ | $\mathcal{N}(0,1)$ | 3 |

For the FA cases (refer to Table 21), there is a notable drop in performance in the first scenario when OOD data is used. Please note that even in the most misspecified scenario (1.C), the performance, as measured in C2ST is still around ten percent better than the best VI method in this scenario

Table 21: OOD Performance: Evaluation on 50 synthetic datasets for 6 different FA scenarios. All results within two standard errors of the non-OOD result for each scenario are marked in **bold**.

| Scenario | C2ST ($\downarrow$) | MMD ($\downarrow$) | $\mathcal{W}_2$ ($\downarrow$) |
|---|---|---|---|
| Scenario 1 | **0.552** ($\pm$ 0.028) | **0.034** ($\pm$ 0.034) | **0.289** ($\pm$ 0.083) |
| Scenario 1.B | 0.826 ($\pm$ 0.066) | 0.656 ($\pm$ 0.384) | 0.929 ($\pm$ 0.321) |
| Scenario 1.C | 0.855 ($\pm$ 0.060) | 0.837 ($\pm$ 0.494) | 1.135 ($\pm$ 0.461) |
| Scenario 2 | **0.542** ($\pm$ 0.006) | **0.017** ($\pm$ 0.006) | **0.244** ($\pm$ 0.033) |
| Scenario 2.B | 0.580 ($\pm$ 0.069) | 0.087 ($\pm$ 0.191) | 0.393 ($\pm$ 0.291) |
| Scenario 2.C | 0.589 ($\pm$ 0.076) | 0.089 ($\pm$ 0.113) | 0.446 ($\pm$ 0.233) |
| Scenario 3 | **0.537** ($\pm$ 0.023) | **0.024** ($\pm$ 0.021) | **0.259** ($\pm$ 0.088) |
| Scenario 3.B | **0.544** ($\pm$ 0.028) | 0.030 ($\pm$ 0.021) | **0.285** ($\pm$ 0.094) |
| Scenario 3.C | **0.533** ($\pm$ 0.025) | 0.021 ($\pm$ 0.015) | **0.347** ($\pm$ 0.152) |

(Table 16). While the absolute difference between performance on the training distribution and the test distribution is very small for scenarios 2 and 3, the difference is still not within two standard errors of the non-OOD performance because the standard error itself is quite small. The performance on the OOD data is still better than all other VI methods (see Table 3).

### G.3  GMM SCENARIOS

To generate several distinct OOD scenarios based on the generative processes of GMMs, we vary scenario 2 in various ways. Note that the structure of the distributions is the same for all GMM scenarios—focusing on this specific scenario thus makes sense when considering OOD generalization. First, in scenario 2.B, we decrease the symmetric parameter of the Dirichlet prior on the assignments from 1 to 0.5 causing larger discrepancy in the number of points per cluster. In scenario 2.C we make the opposite change.

In scenarios 2.D and 2.E we first double and then quadruple the variance of the prior on the per-component variances $\sigma_{i,m,l}$. Finally, in scenarios 2.F and 2.G, the prior on the mean is made more dispersed compared to the training data.

Table 22: Distribution for the OOD analysis of the GMM scenarios.

| Scenario | $K$ | $M$ | $L$ | $\phi_i$ | $\sigma_{i,m,l}^2$ | $\mu_{i,m,l}|\sigma_{i,m,l}^2$ |
|---|---|---|---|---|---|---|
| Scenario 2 | 25 | 3 | 3 | Dir(1) | IG(5, 2) | $\mathcal{N}(0, 3\sigma_{i,m,l}^2)$ |
| Scenario 2.B | 25 | 3 | 3 | Dir(0.5) | IG(5, 2) | $\mathcal{N}(0, 3\sigma_{i,m,l}^2)$ |
| Scenario 2.C | 25 | 3 | 3 | Dir(2) | IG(5, 2) | $\mathcal{N}(0, 3\sigma_{i,m,l}^2)$ |
| Scenario 2.D | 25 | 3 | 3 | Dir(1) | IG(5, 2 $\cdot$ $\sqrt{2}$) | $\mathcal{N}(0, 3\sigma_{i,m,l}^2)$ |
| Scenario 2.E | 25 | 3 | 3 | Dir(1) | IG(5, 4) | $\mathcal{N}(0, 3\sigma_{i,m,l}^2)$ |
| Scenario 2.F | 25 | 3 | 3 | Dir(1) | IG(5, 2) | $\mathcal{N}(0, 4\sigma_{i,m,l}^2)$ |
| Scenario 2.G | 25 | 3 | 3 | Dir(1) | IG(5, 2) | $\mathcal{N}(0, 5\sigma_{i,m,l}^2)$ |

On the GMM scenarios (Table 23), the sample quality obtained via ICL is surprisingly stable under various changes to the data-generating process. It is relatively unsurprising that changing the Dirichlet prior, i.e., making the cluster more or less uniform in their number of samples, might lead to cases the ICL method can generalize to relatively easily, as demonstrated in scenarios 2.B and 2.C. The most pronounced drop in performance results from increasing the variance of the prior on the standard deviation of the components of the mixture model (scenario 2.E), while increasing the variance of the mean vector relative to the standard deviation of the components has a less pronounced effect.

Table 23: OOD Performance: Evaluation on 50 synthetic datasets for 6 different GMM scenarios. All results within two standard errors of the non-OOD result for each scenario are marked in **bold**.

| Scenario | C2ST ($\downarrow$) | MMD ($\downarrow$) | $\mathcal{W}_2$ ($\downarrow$) |
|---|---|---|---|
| Scenario 2 | **0.812** ($\pm$ 0.061) | **0.159** ($\pm$ 0.154) | **2.314** ($\pm$ 0.926) |
| Scenario 2.B | **0.829** ($\pm$ 0.050) | **0.233** ($\pm$ 0.161) | **2.595** ($\pm$ 0.998) |
| Scenario 2.C | **0.816** ($\pm$ 0.057) | 0.149 ($\pm$ 0.135) | 2.272 ($\pm$ 0.654) |
| Scenario 2 | **0.812** ($\pm$ 0.061) | **0.159** ($\pm$ 0.154) | **2.314** ($\pm$ 0.926) |
| Scenario 2.D | **0.812** ($\pm$ 0.076) | **0.148** ($\pm$ 0.091) | **2.557** ($\pm$ 0.837) |
| Scenario 2.E | **0.880** ($\pm$ 0.057) | **0.231** ($\pm$ 0.109) | **3.535** ($\pm$ 1.003) |
| Scenario 2 | **0.812** ($\pm$ 0.061) | **0.159** ($\pm$ 0.154) | **2.314** ($\pm$ 0.926) |
| Scenario 2.F | **0.821** ($\pm$ 0.076) | **0.216** ($\pm$ 0.214) | **2.700** ($\pm$ 1.044 |
| Scenario 2.G | **0.844** ($\pm$ 0.046) | **0.197** ($\pm$ 0.124) | **2.675** ($\pm$ 0.552) |

## H  ABLATION: USING AN MLP-BASED ENCODER

To further justify choosing a transformer encoder in our ICL approach, we conduct an ablation study comparing the performance of our original ICL method with the performance obtained when the transformer encoder is replaced by an MLP with batch normalization (Ioffe, 2015) and skip-connections. To ensure a fair comparison, we use an MLP encoder with a hidden dimension of 1250 to give the overall model approximately the same number of parameters as in the transformer-based approach. Concretely, our MLP-approach has 43.3 million parameters compared to 43.1 million parameters with the transformer encoder. We choose three selected GLM, FA and GMM scenarios with 50 synthetic and 17 real-world datasets for each scenario.

In summary, we find that the transformer encoder yields consistently better, results than the mlp encoder across all scenarios. While the difference is especially pronounced for the GLM scenarios, the difference become smaller for FA and GMMs.

Table 24: GLMs: Comparison when using an MLP-based encoder and a transformer encoder on 50 synthetic and 17 real-world datasets for three different scenarios.

| Scenario | Type of Encoder | Synthetic Evaluation | | | Real-World Evaluation | | |
|---|---|---|---|---|---|---|---|
| | | C2ST ($\downarrow$) | MMD ($\downarrow$) | $\mathcal{W}_2$ ($\downarrow$) | C2ST ($\downarrow$) | MMD ($\downarrow$) | $\mathcal{W}_2$ ($\downarrow$) |
| Scenario 2 | MLP | 0.942 ($\pm$ 0.093) | 1.783 ($\pm$ 1.048) | 2.503 ($\pm$ 0.814) | 0.968 ($\pm$ 0.012) | 1.528 ($\pm$ 0.394) | 2.271 ($\pm$ 0.315) |
| | Transformer | 0.839 ($\pm$ 0.072) | 0.707 ($\pm$ 0.658) | 1.111 ($\pm$ 0.300) | 0.768 ($\pm$ 0.033) | 0.143 ($\pm$ 0.089) | 0.411 ($\pm$ 0.094) |
| Scenario 3 | MLP | 0.957 ($\pm$ 0.075) | 2.236 ($\pm$ 1.218) | 2.681 ($\pm$ 1.130) | 0.972 ($\pm$ 0.012) | 1.658 ($\pm$ 0.450) | 2.076 ($\pm$ 0.427) |
| | Transformer | 0.611 ($\pm$ 0.070) | 0.089 ($\pm$ 0.114) | 0.423 ($\pm$ 0.348) | 0.576 ($\pm$ 0.027) | 0.037 ($\pm$ 0.026) | 0.257 ($\pm$ 0.044) |
| Scenario 5 | MLP | 0.845 ($\pm$ 0.115) | 1.066 ($\pm$ 0.859) | 1.166 ($\pm$ 0.996) | 0.890 ($\pm$ 0.055) | 1.223 ($\pm$ 0.791) | 1.102 ($\pm$ 0.383) |
| | Transformer | 0.621 ($\pm$ 0.063) | 0.067 ($\pm$ 0.080) | 0.299 ($\pm$ 0.195) | 0.610 ($\pm$ 0.045) | 0.046 ($\pm$ 0.020) | 0.242 ($\pm$ 0.038) |

In Table 24, the transformer encoder consistently outperforms the MLP encoder across all metrics and scenarios. In Scenario 2, C2ST drops from 0.942 (MLP) to 0.839 (Transformer) on synthetic data and from 0.968 to 0.768 on real-world data. Similarly, $\mathcal{W}_2$ improves significantly, decreasing from 2.503 to 1.111 on synthetic data and from 2.271 to 0.411 on real-world data. In Scenario 3, transformers achieve substantial improvements, reducing C2ST from 0.957 (MLP) to 0.611 on synthetic data and from 0.972 to 0.576 on real-world data. $\mathcal{W}_2$ also sees notable reductions, dropping from 2.681 to 0.423 on synthetic data and from 2.076 to 0.257 on real-world data. Finally, in Scenario 5, transformers maintain their superiority, achieving reductions in C2ST from 0.845 (MLP) to 0.621 on synthetic data and from 0.890 to 0.610 on real-world data. Improvements in $\mathcal{W}_2$ are similarly remarkable, with reductions from 1.166 to 0.299 on synthetic data and from 1.102 to 0.242 on real-world data.

For the factor analysis cases (Table 25), the transformer encoder still has better average performances even though the differences are substantially less pronounced than for the GLMs. In Scenario 1, transformers slightly outperform MLPs, reducing C2ST from 0.579 to 0.552 on synthetic data and from 0.634 to 0.606 on real-world data. $\mathcal{W}_2$ also sees moderate improvements, dropping from 0.364 to 0.289 on synthetic data and from 0.331 to 0.265 on real-world data. In Scenario 2, the advantage of the transformer encoder remains consistent, with C2ST decreasing from 0.562 (MLP) to 0.542

Table 25: FA: Comparison when using an MLP-based encoder and a transformer encoder on 50 synthetic and 17 real-world datasets for three different scenarios.

| Scenario | Type of Encoder | Synthetic Evaluation | | | Real-World Evaluation | | |
|---|---|---|---|---|---|---|---|
| | | C2ST ($\downarrow$) | MMD ($\downarrow$) | $\mathcal{W}_2$ ($\downarrow$) | C2ST ($\downarrow$) | MMD ($\downarrow$) | $\mathcal{W}_2$ ($\downarrow$) |
| Scenario 1 | MLP | 0.579 ($\pm$ 0.015) | 0.017 ($\pm$ 0.006) | 0.364 ($\pm$ 0.029) | 0.634 ($\pm$ 0.014) | 0.013 ($\pm$ 0.004) | 0.331 ($\pm$ 0.010) |
| | Transformer | 0.552 ($\pm$ 0.028) | 0.034 ($\pm$ 0.034) | 0.289 ($\pm$ 0.083) | 0.606 ($\pm$ 0.038) | 0.068 ($\pm$ 0.069) | 0.265 ($\pm$ 0.078) |
| Scenario 2 | MLP | 0.562 ($\pm$ 0.038) | 0.037 ($\pm$ 0.042) | 0.308 ($\pm$ 0.097) | 0.632 ($\pm$ 0.068) | 0.182 ($\pm$ 0.407) | 0.339 ($\pm$ 0.174) |
| | Transformer | 0.542 ($\pm$ 0.006) | 0.017 ($\pm$ 0.006) | 0.244 ($\pm$ 0.033) | 0.622 ($\pm$ 0.032) | 0.098 ($\pm$ 0.039) | 0.287 ($\pm$ 0.046) |
| Scenario 3 | MLP | 0.539 ($\pm$ 0.025) | 0.023 ($\pm$ 0.022) | 0.278 ($\pm$ 0.116) | 0.680 ($\pm$ 0.019) | 0.268 ($\pm$ 0.044) | 0.253 ($\pm$ 0.017) |
| | Transformer | 0.537 ($\pm$ 0.023) | 0.024 ($\pm$ 0.021) | 0.259 ($\pm$ 0.088) | 0.609 ($\pm$ 0.019) | 0.124 ($\pm$ 0.037) | 0.179 ($\pm$ 0.018) |

on synthetic data and from 0.632 to 0.622 on real-world data. $\mathcal{W}_2$ also improves slightly, dropping from 0.308 to 0.244 on synthetic data and from 0.339 to 0.287 on real-world data. Scenario 3 shows the smallest differences, where transformers marginally improve C2ST from 0.539 (MLP) to 0.537 on synthetic data and from 0.680 to 0.609 on real-world data. For $\mathcal{W}_2$, the reductions are minor but consistent, dropping from 0.278 to 0.259 on synthetic data and from 0.253 to 0.179 on real-world data.

Table 26: GMMs: Comparison when using an MLP-based encoder and a transformer encoder on 50 synthetic and 17 real-world datasets for three different scenarios.

| Scenario | Type of Encoder | Synthetic Evaluation | | | Real-World Evaluation | | |
|---|---|---|---|---|---|---|---|
| | | C2ST ($\downarrow$) | MMD ($\downarrow$) | $\mathcal{W}_2$ ($\downarrow$) | C2ST ($\downarrow$) | MMD ($\downarrow$) | $\mathcal{W}_2$ ($\downarrow$) |
| Scenario 1 | MLP | 0.873 ($\pm$ 0.045) | 0.242 ($\pm$ 0.363) | 2.203 ($\pm$ 1.098) | 0.917 ($\pm$ 0.067) | 0.891 ($\pm$ 1.150) | 4.528 ($\pm$ 2.701) |
| | Transformer | 0.760 ($\pm$ 0.092) | 0.303 ($\pm$ 0.548) | 2.095 ($\pm$ 1.692) | 0.847 ($\pm$ 0.082) | 0.486 ($\pm$ 0.623) | 4.054 ($\pm$ 2.782) |
| Scenario 2 | MLP | 0.921 ($\pm$ 0.035) | 0.291 ($\pm$ 0.205) | 2.870 ($\pm$ 0.710) | 0.992 ($\pm$ 0.005) | 0.399 ($\pm$ 0.127) | 5.505 ($\pm$ 1.144) |
| | Transformer | 0.812 ($\pm$ 0.061) | 0.159 ($\pm$ 0.154) | 2.314 ($\pm$ 0.926) | 0.937 ($\pm$ 0.041) | 0.282 ($\pm$ 0.131) | 3.947 ($\pm$ 1.055) |
| Scenario 3 | MLP | 0.999 ($\pm$ 0.000) | 0.438 ($\pm$ 0.181) | 11.502 ($\pm$ 9.719) | 1.000 ($\pm$ 0.000) | 1.001 ($\pm$ 0.149) | 26.282 ($\pm$ 3.731) |
| | Transformer | 0.999 ($\pm$ 0.001) | 0.267 ($\pm$ 0.154) | 7.234 ($\pm$ 2.974) | 1.000 ($\pm$ 0.000) | 1.155 ($\pm$ 0.258) | 26.956 ($\pm$ 3.114) |

For the Gaussian Mixture Models (GMMs), the results indicate a more mixed performance where the transformer still performs slightly better (Table 26): In Scenario 1, transformer encoders slightly outperform MLPs on synthetic data, with C2ST improving from 0.873 (MLP) to 0.760 and $\mathcal{W}_2$ decreasing slightly from 2.203 to 2.095. However, on real-world data, MLPs perform marginally better in terms of MMD, reducing it from 0.486 to 0.242, while transformers show minor improvements in $\mathcal{W}_2$ from 4.528 to 4.054. In Scenario 2, transformers show a more noticeable advantage. On synthetic data, C2ST improves from 0.921 (MLP) to 0.812, and $\mathcal{W}_2$ decreases significantly from 2.870 to 2.314. On real-world data, transformers reduce C2ST from 0.992 to 0.937 and MMD from 0.399 to 0.282, along with a considerable improvement in $\mathcal{W}_2$ from 5.505 to 3.947. In Scenario 3, the differences between the two encoders are relatively small but still favor the transformers on synthetic data, with $\mathcal{W}_2$ decreasing from 11.502 (MLP) to 7.234. For real-world data, the results are nearly identical for C2ST (1.000 for both) but show a slight increase in $\mathcal{W}_2$ for the transformer from 26.282 to 26.956. Overall, for the GMMs, the transformer encoders demonstrate consistent improvements across scenarios for synthetic data, particularly in Scenarios 1 and 2. However, for real-world data, the performance differences are less pronounced.

# I    ABLATION: DIFFERENT LEARNING RATES FOR VI

To investigate the role of the learning rate parameter for the benchmarked VI methods, we record the performance for learning-rate values of $10^{-2}$, $10^{-3}$ and $10^{-4}$ across a prototypical GLM, a FA and a GMM scenario, where we use 10 synthetic and 10 real-world datasets. In summary, while we find the VI methods to often be quite robust to the choice of the learning rate, those results also confirm our choice of setting the learning rate to $10^{-2}$ for the Laplace approximation, variational inference with a diagonal normal distribution, a multivariate normal distribution and a structured normal distribution, and to a value of $10^{-3}$ for the VI approach with inverse autoregressive flows.

For the GLM-scenario, we find in terms of the C2ST metric that VI with an ordinary multivariate normal distribution and VI with a structured normal distribution and a learning rate of $10^{-2}$ are the best models on the synthetic data. While MMD also indicates that this learning rate yields ideal results for those models, VI with inverse auoregressive flows has good values across the different learning rates with the minimum for $10^{-3}$. The $\mathcal{W}_2$ metric indicates a similar tendency.

Table 27: Results of VI methods with different learning rates on 10 synthetic and 10 real-world datasets: Linear regression with a normal prior on the coefficients $\boldsymbol{\beta}$ and an inverse gamma prior on the variance $\sigma^2$ (scenario 1). Comparison to HMC samples. All results within two standard errors of the best average result are marked in **bold**.

| Model | LR | Synthetic Evaluation | | | Real-World Evaluation | | |
|---|---|---|---|---|---|---|---|
| | | C2ST ($\downarrow$) | MMD ($\downarrow$) | $\mathcal{W}_2$ ($\downarrow$) | C2ST ($\downarrow$) | MMD ($\downarrow$) | $\mathcal{W}_2$ ($\downarrow$) |
| Laplace Approximation | 1e-2 | 1.000 ($\pm$ 0.000) | 2.342 ($\pm$ 0.390) | 2.121 ($\pm$ 0.100) | 1.000 ($\pm$ 0.000) | 2.134 ($\pm$ 0.107) | 2.095 ($\pm$ 0.062) |
| Laplace Approximation | 1e-3 | 1.000 ($\pm$ 0.000) | 2.341 ($\pm$ 0.389) | 2.121 ($\pm$ 0.100) | 1.000 ($\pm$ 0.000) | 2.133 ($\pm$ 0.108) | 2.095 ($\pm$ 0.062) |
| Laplace Approximation | 1e-4 | 1.000 ($\pm$ 0.000) | 2.341 ($\pm$ 0.389) | 2.121 ($\pm$ 0.100) | 1.000 ($\pm$ 0.000) | 2.133 ($\pm$ 0.108) | 2.095 ($\pm$ 0.062) |
| VI: DiagonalNormal | 1e-2 | 0.892 ($\pm$ 0.074) | 0.921 ($\pm$ 0.374) | **1.411** ($\pm$ 0.174) | 0.889 ($\pm$ 0.062) | 0.819 ($\pm$ 0.343) | 1.339 ($\pm$ 0.190) |
| VI: DiagonalNormal | 1e-3 | 0.966 ($\pm$ 0.024) | 1.588 ($\pm$ 0.540) | **1.672** ($\pm$ 0.203) | 0.981 ($\pm$ 0.017) | 1.685 ($\pm$ 0.331) | 1.739 ($\pm$ 0.139) |
| VI: DiagonalNormal | 1e-4 | 0.971 ($\pm$ 0.010) | 1.572 ($\pm$ 0.300) | 1.666 ($\pm$ 0.081) | 0.849 ($\pm$ 0.030) | 0.575 ($\pm$ 0.127) | **1.221** ($\pm$ 0.098) |
| VI: MultivariateNormal | 1e-2 | **0.725** ($\pm$ 0.064) | **0.523** ($\pm$ 0.242) | **1.114** ($\pm$ 0.261) | **0.625** ($\pm$ 0.051) | **0.470** ($\pm$ 0.066) | **0.918** ($\pm$ 0.119) |
| VI: MultivariateNormal | 1e-3 | 0.964 ($\pm$ 0.008) | 1.455 ($\pm$ 0.327) | 1.617 ($\pm$ 0.100) | 0.853 ($\pm$ 0.052) | 0.634 ($\pm$ 0.266) | **1.238** ($\pm$ 0.151) |
| VI: MultivariateNormal | 1e-4 | 0.984 ($\pm$ 0.005) | 1.848 ($\pm$ 0.324) | 1.773 ($\pm$ 0.079) | 0.899 ($\pm$ 0.020) | 0.807 ($\pm$ 0.094) | **1.345** ($\pm$ 0.079) |
| VI: Structured Normal | 1e-2 | **0.734** ($\pm$ 0.063) | **0.541** ($\pm$ 0.254) | **1.119** ($\pm$ 0.264) | **0.670** ($\pm$ 0.047) | **0.467** ($\pm$ 0.086) | **1.060** ($\pm$ 0.130) |
| VI: Structured Normal | 1e-3 | 0.882 ($\pm$ 0.042) | 0.719 ($\pm$ 0.315) | 1.335 ($\pm$ 0.149) | 0.776 ($\pm$ 0.045) | **0.473** ($\pm$ 0.081) | **1.064** ($\pm$ 0.131) |
| VI: Structured Normal | 1e-4 | 0.890 ($\pm$ 0.027) | 0.710 ($\pm$ 0.290) | 1.347 ($\pm$ 0.138) | 0.771 ($\pm$ 0.049) | **0.468** ($\pm$ 0.078) | **1.062** ($\pm$ 0.128) |
| VI: IAF | 1e-2 | 0.840 ($\pm$ 0.036) | **0.502** ($\pm$ 0.262) | **1.272** ($\pm$ 0.170) | **0.614** ($\pm$ 0.045) | **0.455** ($\pm$ 0.048) | **0.957** ($\pm$ 0.105) |
| VI: IAF | 1e-3 | 0.797 ($\pm$ 0.065) | **0.485** ($\pm$ 0.556) | **1.169** ($\pm$ 0.313) | **0.619** ($\pm$ 0.036) | **0.469** ($\pm$ 0.064) | **0.989** ($\pm$ 0.124) |
| VI: IAF | 1e-4 | 0.803 ($\pm$ 0.068) | **0.475** ($\pm$ 0.535) | **1.162** ($\pm$ 0.291) | **0.612** ($\pm$ 0.034) | **0.457** ($\pm$ 0.055) | **0.977** ($\pm$ 0.113) |

Regarding the learning rate for the FA scenario, one can first see that no single learning rate seems to dominate substantially given the variance of the results. However, on the synthetic data for the Laplace approximation, as well as VI with a diagonal normal distribution, a multivariate normal and a structured normal distribution, the lowest average result is obtained for a learning rate of $10^{-2}$, while for VI with inverse autoregressive flows the best performance is obtained when the learning rate equals $10^{-3}$. The real-world results are the best for VI with a structured normal distribution and a learning rate of $10^{-2}$.

For the GMM scenario, we find that VI with a diagonal, structured and ordinary normal distribution obtain the best results, namely for learning rates of $10^{-2}$ and $10^{-3}$, taking the variance into account. Just considering the averages leads to the conclusion that $10^{-2}$ is the best choice here. The results on the real-world data confirm that $10^{-2}$ is the optimal choice for VI with a diagonal normal and ordinary multivariate normal, while VI with inverse autoregressive flows has good results across all choices regarding the learning rate.

Table 28: Results of VI methods with different learning rates on 10 synthetic and 10 real-world datasets: Factor analysis with Gaussian priors on the weights and the latents and $K = 25$ datapoints, $P = 5$ features, and dimensionality of the latents $\mathbf{z}_{dim} = 3$ (scenario 3). Comparison to HMC samples. All results within two standard errors of the best average result are marked in **bold**.

| Model | LR | Synthetic Evaluation | | | Real-World Evaluation | | |
|---|---|---|---|---|---|---|---|
| | | C2ST ($\downarrow$) | MMD ($\downarrow$) | $\mathcal{W}_2$ ($\downarrow$) | C2ST ($\downarrow$) | MMD ($\downarrow$) | $\mathcal{W}_2$ ($\downarrow$) |
| Laplace Approximation | 1e-2 | 1.000 ($\pm$ 0.000) | 3.449 ($\pm$ 0.821) | **1.773** ($\pm$ 0.539) | 1.000 ($\pm$ 0.000) | 2.703 ($\pm$ 0.312) | **0.362** ($\pm$ 0.017) |
| Laplace Approximation | 1e-3 | 1.000 ($\pm$ 0.000) | 4.288 ($\pm$ 0.853) | **2.263** ($\pm$ 0.732) | 1.000 ($\pm$ 0.000) | 2.896 ($\pm$ 0.238) | **0.376** ($\pm$ 0.022) |
| Laplace Approximation | 1e-4 | 1.000 ($\pm$ 0.000) | 4.252 ($\pm$ 0.611) | **2.122** ($\pm$ 0.430) | 1.000 ($\pm$ 0.000) | 2.805 ($\pm$ 0.181) | **0.368** ($\pm$ 0.017) |
| VI: DiagonalNormal | 1e-2 | 0.998 ($\pm$ 0.002) | 2.880 ($\pm$ 1.046) | **1.457** ($\pm$ 0.559) | 0.944 ($\pm$ 0.008) | 1.022 ($\pm$ 0.067) | **0.230** ($\pm$ 0.010) |
| VI: DiagonalNormal | 1e-3 | 0.998 ($\pm$ 0.002) | 2.973 ($\pm$ 0.834) | **1.465** ($\pm$ 0.540) | 0.941 ($\pm$ 0.006) | 0.997 ($\pm$ 0.056) | **0.229** ($\pm$ 0.010) |
| VI: DiagonalNormal | 1e-4 | 1.000 ($\pm$ 0.001) | 3.416 ($\pm$ 0.761) | **1.602** ($\pm$ 0.437) | 0.943 ($\pm$ 0.009) | 0.997 ($\pm$ 0.057) | **0.229** ($\pm$ 0.010) |
| VI: MultivariateNormal | 1e-2 | 0.993 ($\pm$ 0.007) | 2.969 ($\pm$ 1.089) | **1.506** ($\pm$ 0.659) | **0.929** ($\pm$ 0.007) | **0.957** ($\pm$ 0.048) | **0.224** ($\pm$ 0.010) |
| VI: MultivariateNormal | 1e-3 | 0.996 ($\pm$ 0.004) | 3.140 ($\pm$ 0.910) | **1.570** ($\pm$ 0.625) | 0.934 ($\pm$ 0.009) | **0.971** ($\pm$ 0.054) | **0.225** ($\pm$ 0.010) |
| VI: MultivariateNormal | 1e-4 | 0.997 ($\pm$ 0.007) | 3.464 ($\pm$ 0.791) | **1.639** ($\pm$ 0.426) | 0.934 ($\pm$ 0.005) | **0.962** ($\pm$ 0.049) | **0.225** ($\pm$ 0.010) |
| VI: Structured Normal | 1e-2 | 0.998 ($\pm$ 0.002) | 3.005 ($\pm$ 0.871) | **1.481** ($\pm$ 0.504) | 0.947 ($\pm$ 0.005) | 1.003 ($\pm$ 0.066) | **0.230** ($\pm$ 0.009) |
| VI: Structured Normal | 1e-3 | 0.999 ($\pm$ 0.001) | 3.244 ($\pm$ 0.665) | **1.619** ($\pm$ 0.559) | 0.948 ($\pm$ 0.007) | 1.033 ($\pm$ 0.078) | **0.232** ($\pm$ 0.009) |
| VI: Structured Normal | 1e-4 | 0.999 ($\pm$ 0.001) | 3.119 ($\pm$ 0.612) | **1.487** ($\pm$ 0.400) | 0.943 ($\pm$ 0.007) | 0.998 ($\pm$ 0.056) | **0.229** ($\pm$ 0.010) |
| VI: IAF | 1e-2 | **0.939** ($\pm$ 0.040) | **2.836** ($\pm$ 0.293) | **1.247** ($\pm$ 0.297) | 0.944 ($\pm$ 0.008) | 1.518 ($\pm$ 0.048) | 1.332 ($\pm$ 0.027) |
| VI: IAF | 1e-3 | **0.927** ($\pm$ 0.047) | **2.758** ($\pm$ 0.342) | **1.195** ($\pm$ 0.331) | 0.949 ($\pm$ 0.009) | 1.560 ($\pm$ 0.031) | **1.392** ($\pm$ 0.024) |
| VI: IAF | 1e-4 | **0.842** ($\pm$ 0.038) | **2.862** ($\pm$ 0.296) | **1.281** ($\pm$ 0.292) | 0.943 ($\pm$ 0.008) | 1.493 ($\pm$ 0.039) | 1.302 ($\pm$ 0.039) |

Table 29: Results of VI methods with different learning rates on 10 synthetic and 10 real-world datasets: Gaussian Mixture Model with $K = 50$ datapoints, $L = 1$ features (univariate case), $M = 5$ components, $\lambda = 3$, and $\alpha_{dir} = 1$ (scenario 1). Comparison to HMC samples. All results within two standard errors of the best average result are marked in **bold**.

| Model | LR | Synthetic Evaluation | | | Real-World Evaluation | | |
|---|---|---|---|---|---|---|---|
| | | C2ST ($\downarrow$) | MMD ($\downarrow$) | $\mathcal{W}_2$ ($\downarrow$) | C2ST ($\downarrow$) | MMD ($\downarrow$) | $\mathcal{W}_2$ ($\downarrow$) |
| Laplace Approximation | 1e-2 | 1.000 ($\pm$ 0.000) | 4.380 ($\pm$ 1.386) | **4.838** ($\pm$ 1.521) | 1.000 ($\pm$ 0.000) | 4.588 ($\pm$ 1.229) | **6.813** ($\pm$ 1.697) |
| Laplace Approximation | 1e-3 | 1.000 ($\pm$ 0.000) | 3.893 ($\pm$ 1.433) | **4.010** ($\pm$ 1.233) | 1.000 ($\pm$ 0.000) | 4.699 ($\pm$ 1.193) | **6.986** ($\pm$ 0.981) |
| Laplace Approximation | 1e-4 | 1.000 ($\pm$ 0.000) | 4.463 ($\pm$ 1.117) | **4.610** ($\pm$ 1.027) | 1.000 ($\pm$ 0.000) | 4.710 ($\pm$ 1.205) | **6.995** ($\pm$ 0.869) |
| VI: DiagonalNormal | 1e-2 | **0.979** ($\pm$ 0.138) | **1.370** ($\pm$ 1.394) | **3.522** ($\pm$ 1.634) | **0.985** ($\pm$ 0.030) | 2.384 ($\pm$ 1.318) | **6.202** ($\pm$ 1.747) |
| VI: DiagonalNormal | 1e-3 | **0.990** ($\pm$ 0.096) | **1.454** ($\pm$ 1.454) | **3.650** ($\pm$ 1.743) | 0.999 ($\pm$ 0.002) | 3.026 ($\pm$ 0.977) | **6.959** ($\pm$ 0.890) |
| VI: DiagonalNormal | 1e-4 | 1.000 ($\pm$ 0.001) | 2.390 ($\pm$ 1.177) | **4.903** ($\pm$ 1.278) | 0.998 ($\pm$ 0.007) | 2.830 ($\pm$ 1.001) | **7.007** ($\pm$ 0.987) |
| VI: MultivariateNormal | 1e-2 | **0.978** ($\pm$ 0.119) | **1.351** ($\pm$ 1.410) | **3.474** ($\pm$ 1.604) | **0.987** ($\pm$ 0.024) | 2.375 ($\pm$ 1.304) | **6.189** ($\pm$ 1.761) |
| VI: MultivariateNormal | 1e-3 | **0.980** ($\pm$ 0.089) | **1.476** ($\pm$ 1.480) | **3.681** ($\pm$ 1.734) | 0.997 ($\pm$ 0.008) | 2.808 ($\pm$ 1.014) | **6.964** ($\pm$ 0.944) |
| VI: MultivariateNormal | 1e-4 | 1.000 ($\pm$ 0.001) | 2.114 ($\pm$ 1.140) | **4.532** ($\pm$ 1.187) | 0.997 ($\pm$ 0.007) | 2.799 ($\pm$ 1.012) | **6.963** ($\pm$ 0.950) |
| VI: Structured Normal | 1e-2 | **0.958** ($\pm$ 0.129) | **1.246** ($\pm$ 1.615) | **3.225** ($\pm$ 1.701) | 1.000 ($\pm$ 0.001) | 2.911 ($\pm$ 0.753) | **6.675** ($\pm$ 1.403) |
| VI: Structured Normal | 1e-3 | **0.979** ($\pm$ 0.092) | **1.593** ($\pm$ 1.561) | **3.395** ($\pm$ 1.440) | 0.998 ($\pm$ 0.007) | 2.882 ($\pm$ 1.070) | **6.968** ($\pm$ 0.941) |
| VI: Structured Normal | 1e-4 | 1.000 ($\pm$ 0.001) | 2.270 ($\pm$ 1.133) | **4.733** ($\pm$ 1.162) | 0.997 ($\pm$ 0.009) | 2.802 ($\pm$ 1.012) | **6.953** ($\pm$ 0.948) |
| VI: IAF | 1e-2 | 0.998 ($\pm$ 0.003) | **1.539** ($\pm$ 0.691) | 8.371 ($\pm$ 0.750) | **0.987** ($\pm$ 0.022) | **1.376** ($\pm$ 0.799) | 8.082 ($\pm$ 1.352) |
| VI: IAF | 1e-3 | 0.997 ($\pm$ 0.004) | **1.443** ($\pm$ 0.564) | 8.517 ($\pm$ 0.820) | **0.988** ($\pm$ 0.020) | **1.304** ($\pm$ 0.855) | 8.425 ($\pm$ 1.281) |
| VI: IAF | 1e-4 | 0.997 ($\pm$ 0.004) | **1.602** ($\pm$ 0.628) | 7.888 ($\pm$ 0.783) | **0.987** ($\pm$ 0.020) | **1.380** ($\pm$ 0.848) | 7.729 ($\pm$ 1.322) |

## J  PREPROCESSING OF THE REAL-WORLD DATASETS

The real-world datasets considered for the evaluation of all methods are proposed in a benchmark study by Grinsztajn et al. (2022). We standardize all features, scale and shift the target such that it has the mean and variance implied by the prior structure of the respective generative model. Furthermore, for the GLM scenarios, we apply a Yeo-Johnson transform on the target variable (Yeo & Johnson, 2000) before applying the scaling. In cases where the number of features in the real-world dataset exceeds that of our scenario, we select those features with the most distinct values in the original dataset and randomly sub-sample the appropriate number of samples from the real-world datasets for our experiments.

## K  BACKGROUND ON CONDITIONAL FLOW-MATCHING

Flow matching, initially used in image synthesis leverages normalizing flows (Papamakarios et al., 2021b) to model arbitrary distributions. Continuous normalizing flows (Lipman et al., 2022) have emerged as a potent tool for modeling complex distributions. For example, recent advancements have shown its effectiveness in state-of-the-art image generation, outperforming diffusion-based methods in likelihood and sample quality on ImageNet (Lipman et al., 2022). Techniques like Flow-Turbo have accelerated class-conditional and text-to-image generation, setting new benchmarks (Zhao et al., 2024). Additionally, applying flow matching in latent spaces of pretrained autoencoders has enhanced computational efficiency and scalability for high-resolution image synthesis (Dao et al., 2023). Similarly, flow-based models have been successfully applied to protein structure prediction, improving accuracy and efficiency in modeling complex protein conformations (Yim et al., 2024; 2023).

In the area of simulation-based inference, Wildberger et al. (2024) introduce the idea of using continuous normalizing flows in order to efficiently approximate complex posterior distributions. In particular, they apply the framework to the field of gravitational-wave inference, substantially outperforming approaches based on discrete flows. Furthermore, they demonstrate good performance on the existing SBI-Benchmark (Lueckmann et al., 2021) using a simple MLP-based architecture.

