# OpenReview forum: "In-Context Learning for Full Bayesian Inference"
_ICLR.cc/2025/Conference — Submitted to ICLR 2025_

### Official Review · Reviewer_prH2 · 2024-11-03

**Soundness:** 3
**Presentation:** 2
**Contribution:** 3
**Rating:** 6
**Confidence:** 2

**Summary:**

This work proposes an in-context learning (ICL) approach for full Bayesian inference.
Specifically, PFN is used as a prior, and flow matching is adopted for posterior inference.
Experiments show that ICL can perform similarly to the Hamiltonian Monte Carlo sampler, and its sample quality is better than variational inference.

**Strengths:**

- The experimental results demonstrate that the proposed method approximates the posterior more accurately than VI-based approaches.
In particular, Figure 2 clearly illustrates that the ICL's posterior aligns well with the ground-truth HMC's, while the posteriors by VIs fail to approximate multi-modal distributions. Interestingly, the autoregressive flow-based VI also fails in that way, contrary to its expected property.

**Weaknesses:**

- How to infer the posterior using the proposed method is not explicitly explained.
- The writing could be improved. For example, the current manuscript carefully explains continuous normalizing flows, which is appreciated, but it is unclear how they are used in the proposed method.
- To my understanding, in the ICL framework, the (foundation) model, typically, Transformer, learns algorithms implicitly through the pre-training (e.g., Garg's work).
In this work, however, the model seems designed for inference and pre-training on many related tasks.
I would call such a framework meta-learning, rather than ICL.

**Questions:**

- It would be interesting if the runtime of both pre-training and inference of the proposed mthod was shown.

---

> ### Author Response · Authors · 2024-11-20
> **Response to weaknesses**
>
> Thank you for taking the time to read our manuscript and providing detailed feedback.
>
> We address your questions and comments below. We use double quotation marks (") to mark parts directly quoted from the revised manuscript. In the manuscript itself, all added parts have the text color blue.
> > How to infer the posterior using the proposed method is not explicitly explained [...]
> how [continuous normalizing flows] are used in the proposed method
>
> To address these points, we have added a new section "Implementing Flow Matching" (Section 3.5) to the manuscript that details how exactly the flow matching framework is used during training and how posterior samples are generated based on a pre-trained in-context learner.
>
> Below is the added text quoted from the revised manuscript for your convenience:
>
> “**Implementing Flow Matching**
>
> During the training phase, a tuple $(\mathbf{z}\_1,\mathbf{x})$ is drawn from the distribution $P^{\mathbf{z},\mathbf{x}}$. Additionally, a time step $t \sim \mathcal{U}\[0,1\]$ and a sample $\mathbf{z}\_0$ is drawn from the base distribution $P\_{\mathcal{B}}$, which is a standard Gaussian for all our applications. Subsequently, the ground-truth conditional flow $\psi(\mathbf{z}\_0|\mathbf{x}) = 1 - ( 1- \sigma_{min})t)\mathbf{z}\_0 + t \mathbf{z}\_1$ is computed, pushing forward $P_{\mathcal{B}}$ into $P^{\mathbf{z}|\mathbf{x}}$ up to time-point $t$. The transformer encoder processes $\mathbf{x}$ and the decoder takes the representation of the encoder into account in order to output $v\_{t,\mathbf{x}}^{\theta}(\psi(\mathbf{z}\_0|\mathbf{x}))$. This output should match the vector field that describes how the ground-truth flow $\psi(\mathbf{z}\_0|\mathbf{x})$ continues at time $t$. The discrepancy to the ground-truth vector field is measured with the MSE-loss in Equation 6.
> In the sampling phase, we are given $\mathbf{x}$ and the goal is to sample from $P^{\mathbf{z}|\mathbf{x}}$. To do so, first a vector $\mathbf{z}\_0 \sim P\_{\mathcal{B}}$ is drawn. The data $\mathbf{x}$ is passed through the encoder. The decoder defines a function that maps a time-point $t$ and a vector $\mathbf{\nu}$ onto a vector field: $(t, \mathbf{\nu}) \mapsto v_{t,\mathbf{x}}^{\theta}(\mathbf{\nu})$ taking $\mathbf{x}$ into account. This function is given to an ODE-solver in order to forward-solve the corresponding ODE with boundary conditions $0 \leq t \leq 1$.”
>
> The new section provides a computational perspective on flow matching as it is used in conjunction with the introduced architecture.
>
> > To my understanding, in the ICL framework, the (foundation) model, typically, Transformer, learns algorithms implicitly through the pre-training (e.g., Garg's work). In this work, however, the model seems designed for inference and pre-training on many related tasks. I would call such a framework meta-learning, rather than ICL.
>
> We fully agree that ICL is a special case of meta-learning and included an additional explanation in the revised manuscript in the first paragraph of Section 2:
>
> “ICL is a special case of meta-learning [1] characterized by using a large pre-trained model to learn from a context dataset without explicitly updating task-specific parameters. [...]”
>
> We use the more specific term in-context learning instead of meta-learning to emphasize that our approach yields posterior samples directly based on a context set processed by a large transformer network without requiring additional parameter updates. Taking seminal work on meta-learning into account, e.g. [2,3,4], we believe that the term in-context learning is useful to contrast the underlying process of our approach.
>
> Thank you again for your constructive feedback. Please let us know if our response addresses your questions and concerns, and if there is anything else we need to clarify.
>
> ### References
>
> [1] Vilalta, Ricardo, and Youssef Drissi. "A perspective view and survey of meta-learning." Artificial intelligence review 18 (2002): 77-95.
>
> [2] Finn, Chelsea, et al. "Model-agnostic meta-learning for fast adaptation of deep networks." International conference on machine learning. PMLR, 2017.
>
> [3] Sung, Flood, et al. "Learning to compare: Relation network for few-shot learning." Proceedings of the IEEE conference on computer vision and pattern recognition. 2018.
>
> [4] Vinyals, Oriol, et al. "Matching networks for one shot learning." Advances in neural information processing systems 29 (2016).

---

> > ### Author Response · Authors · 2024-11-20
> > **Response to the question**
> >
> > > It would be interesting if the runtime of both pre-training and inference of the proposed mthod was shown.
> >
> > Thank you for this suggestion. We now include the runtime of all methods during pre-training and inference in a new section in the manuscript ("Appendix E Runtimes"). This demonstrates that while in-context learning requires a relatively large amount of computational resources during the pre-training phase, generating samples based on a new dataset can be done much more efficiently with ICL compared to HMC (for FA even achieving a factor 8 of speed-up).
> > For your convenience, we also include the results (in Appendix E in the revised manuscript) below.
> >
> > | **Scenario** | **Method**              | **Mean Runtime (s)**       |
> > |--------------|--------------------------|----------------------------|
> > | **GLM**      | Laplace Approximation    | 10.48 ± 0.25          |
> > |              | VI: DiagonalNormal       | 12.02 ± 0.26           |
> > |              | VI: MultivariateNormal   | 13.70 ± 0.29          |
> > |              | VI: Structured Normal    | 19.81 ± 0.98               |
> > |              | VI: IAF                  | 15.44 ± 0.30               |
> > |              | HMC                      | 120.24 ± 13.94             |
> > |              | **ICL**                  | 107.79 ± 17.36             |
> > | **FA**       | Laplace Approximation    | 17.85 ± 0.21           |
> > |              | VI: DiagonalNormal       | 20.94 ± 0.66           |
> > |              | VI: MultivariateNormal   | 20.84 ± 0.28          |
> > |              | VI: Structured Normal    | 36.17 ± 0.61               |
> > |              | VI: IAF                  | 23.75 ± 0.38               |
> > |              | HMC                      | 248.26 ± 57.88             |
> > |              | **ICL**                  | 31.49 ± 4.97               |
> > | **GMM**      | Laplace Approximation    | 27.52 ± 0.40           |
> > |              | VI: DiagonalNormal       | 29.74 ± 0.57           |
> > |              | VI: MultivariateNormal   | 30.50 ± 0.41           |
> > |              | VI: Structured Normal    | 42.44 ± 0.44               |
> > |              | VI: IAF                  | 33.39 ± 0.49               |
> > |              | HMC                      | 239.67 ± 32.71             |
> > |              | **ICL**                  | 93.88 ± 10.47              |
> >
> > Thank you again for your constructive feedback. Please let us know if our response addresses your questions and concerns, and if there is anything else we need to clarify.

---

> > > ### Author Response · Authors · 2024-11-24
> > > **The end of the rebuttal phase is approaching**
> > >
> > > Dear Reviewer prH2,
> > >
> > > Thank you again for providing highly valuable feedback on our manuscript.
> > >
> > > As the discussion period will end in approximately 72 hours, we would like to ask if our answers so far have successfully addressed your concerns and answered your questions.
> > >
> > > In summary, we addressed the points raised by you as follows: We added a detailed explanation on how flow matching is used in our approach to the main body of the manuscript, and explain why we use the term in-context learning instead of meta-learning. We also found your proposal to have the runtimes of all methods in the paper very valuable and thus now include them in the revised version.
> > >
> > > If there are any remaining concerns, please let us know. In case we sufficiently addressed your questions, we would greatly appreciate it if you consider raising the score.

---

> > > > ### Comment · Reviewer_prH2 · 2024-11-26
> > > >
> > > > Thank you for the detailed clarification, which helped me understand your work more deeply.
> > > > This raises another question: why is a Transformer necessary in your work?

---

> > > > > ### Author Response · Authors · 2024-11-26
> > > > > **Response to your new question: results - part 1**
> > > > >
> > > > > For your convenience, we also include the results (in Appendix H in the revised manuscript) below.
> > > > >
> > > > > #### GLMs: Comparison when using an MLP-based encoder and a transformer encoder on 50 synthetic and 17 real-world datasets for three different scenarios.
> > > > >
> > > > > | **Scenario**  | **Type of Encoder** | **Synthetic Evaluation**       	|              	|              	| **Real-World Evaluation**   	|              	|              	|
> > > > > |---------------|----------------------|-------------------------------------|------------------|------------------|----------------------------------|------------------|------------------|
> > > > > |           	|                  	| **C2ST (↓)**                   	| **MMD (↓)** 	| **W₂ (↓)**  	| **C2ST (↓)**                	| **MMD (↓)** 	| **W₂ (↓)**  	|
> > > > > | **Scenario 2** | MLP              	| 0.942 (± 0.093)                	| 1.783 (± 1.048) | 2.503 (± 0.814) | 0.968 (± 0.012)             	| 1.528 (± 0.394) | 2.271 (± 0.315) |
> > > > > |           	| Transformer       	| 0.839 (± 0.072)                	| 0.707 (± 0.658) | 1.111 (± 0.300) | 0.768 (± 0.033)             	| 0.143 (± 0.089) | 0.411 (± 0.094) |
> > > > > | **Scenario 3** | MLP              	| 0.957 (± 0.075)                	| 2.236 (± 1.218) | 2.681 (± 1.130) | 0.972 (± 0.012)             	| 1.658 (± 0.450) | 2.076 (± 0.427) |
> > > > > |           	| Transformer       	| 0.611 (± 0.070)                	| 0.089 (± 0.114) | 0.423 (± 0.348) | 0.576 (± 0.027)             	| 0.037 (± 0.026) | 0.257 (± 0.044) |
> > > > > | **Scenario 5** | MLP              	| 0.845 (± 0.115)                	| 1.066 (± 0.859) | 1.166 (± 0.996) | 0.890 (± 0.055)             	| 1.223 (± 0.791) | 1.102 (± 0.383) |
> > > > > |           	| Transformer       	| 0.621 (± 0.063)                	| 0.067 (± 0.080) | 0.299 (± 0.195) | 0.610 (± 0.045)             	| 0.046 (± 0.020) | 0.242 (± 0.038) |

---

> > > > > ### Author Response · Authors · 2024-11-26
> > > > > **Response to your new question: results - part 2**
> > > > >
> > > > > #### FA: Comparison when using an MLP-based encoder and a transformer encoder on 50 synthetic and 17 real-world datasets for three different scenarios.
> > > > >
> > > > > | **Scenario**  | **Type of Encoder** | **Synthetic Evaluation**       	|              	|              	| **Real-World Evaluation**   	|              	|              	|
> > > > > |---------------|----------------------|-------------------------------------|------------------|------------------|----------------------------------|------------------|------------------|
> > > > > |           	|                  	| **C2ST (↓)**                   	| **MMD (↓)** 	| **W₂ (↓)**  	| **C2ST (↓)**                	| **MMD (↓)** 	| **W₂ (↓)**  	|
> > > > > | **Scenario 1** | MLP              	| 0.579 (± 0.015)                	| 0.017 (± 0.006) | 0.364 (± 0.029) | 0.634 (± 0.014)             	| 0.013 (± 0.004) | 0.331 (± 0.010) |
> > > > > |           	| Transformer       	| 0.552 (± 0.028)                	| 0.034 (± 0.034) | 0.289 (± 0.083) | 0.606 (± 0.038)             	| 0.068 (± 0.069) | 0.265 (± 0.078) |
> > > > > | **Scenario 2** | MLP              	| 0.562 (± 0.038)                	| 0.037 (± 0.042) | 0.308 (± 0.097) | 0.632 (± 0.068)             	| 0.182 (± 0.407) | 0.339 (± 0.174) |
> > > > > |           	| Transformer       	| 0.542 (± 0.006)                	| 0.017 (± 0.006) | 0.244 (± 0.033) | 0.622 (± 0.032)             	| 0.098 (± 0.039) | 0.287 (± 0.046) |
> > > > > | **Scenario 3** | MLP              	| 0.539 (± 0.025)                	| 0.023 (± 0.022) | 0.278 (± 0.116) | 0.680 (± 0.019)             	| 0.268 (± 0.044) | 0.253 (± 0.017) |
> > > > > |           	| Transformer       	| 0.537 (± 0.023)                	| 0.024 (± 0.021) | 0.259 (± 0.088) | 0.609 (± 0.019)             	| 0.124 (± 0.037) | 0.179 (± 0.018) |
> > > > >
> > > > > #### GMMs: Comparison when using an MLP-based encoder and a transformer encoder on 50 synthetic and 17 real-world datasets for three different scenarios.
> > > > >
> > > > > | **Scenario**  | **Type of Encoder** | **Synthetic Evaluation**       	|              	|              	| **Real-World Evaluation**   	|              	|              	|
> > > > > |---------------|----------------------|-------------------------------------|------------------|------------------|----------------------------------|------------------|------------------|
> > > > > |           	|                  	| **C2ST (↓)**                   	| **MMD (↓)** 	| **W₂ (↓)**  	| **C2ST (↓)**                	| **MMD (↓)** 	| **W₂ (↓)**  	|
> > > > > | **Scenario 1** | MLP              	| 0.873 (± 0.045)                	| 0.242 (± 0.363) | 2.203 (± 1.098) | 0.917 (± 0.067)             	| 0.891 (± 1.150) | 4.528 (± 2.701) |
> > > > > |           	| Transformer       	| 0.760 (± 0.092)                	| 0.303 (± 0.548) | 2.095 (± 1.692) | 0.847 (± 0.082)             	| 0.486 (± 0.623) | 4.054 (± 2.782) |
> > > > > | **Scenario 2** | MLP              	| 0.921 (± 0.035)                	| 0.291 (± 0.205) | 2.870 (± 0.710) | 0.992 (± 0.005)             	| 0.399 (± 0.127) | 5.505 (± 1.144) |
> > > > > |           	| Transformer       	| 0.812 (± 0.061)                	| 0.159 (± 0.154) | 2.314 (± 0.926) | 0.937 (± 0.041)             	| 0.282 (± 0.131) | 3.947 (± 1.055) |
> > > > > | **Scenario 3** | MLP              	| 0.999 (± 0.000)                	| 0.438 (± 0.181) | 11.502 (± 9.719) | 1.000 (± 0.000)             	| 1.001 (± 0.149) | 26.282 (± 3.731) |
> > > > > |           	| Transformer       	| 0.999 (± 0.001)                	| 0.267 (± 0.154) | 7.234 (± 2.974) | 1.000 (± 0.000)             	| 1.155 (± 0.258) | 26.956 (± 3.114) |

---

> ### Author Response · Authors · 2024-11-26
> **Response to your new question**
>
> Dear Reviewer prH2,
>
> We are pleased to hear that our clarifications helped you understand our work. We would also like to thank you for inquiring about the role of the transformer in our paper, which is indeed very insightful.
>
> > This raises another question: why is a Transformer necessary in your work?
>
> There are several conceptual reasons why transformers are central to our approach:
>
> First, transformers are a commonly used baseline architecture working extremely well across a broad variety of fields such as computer vision, NLP, or even tabular data [1,2,3]. In addition, the papers introducing the PFN methodology [4,5], a central pillar of our approach, heavily rely on the transformer. Therefore, using a transformer is a natural and reasonable choice for our proposal, whose central point is to extend [4,5] and demonstrate the feasibility of in-context learning for full Bayesian inference.
> Furthermore, the notion of in-context learning itself, which we want to extend to full Bayesian inference, is closely connected to the transformer architecture [6,7,8]. This is confirmed by findings suggesting the attention mechanism to play a crucial role for in-context learning [9,10].
>
> **Ablation Study**
>
> Intrigued by your question, we conducted another ablation experiment to study your question from an empirical point-of-view and investigate in which cases the transformer architecture is indeed a crucial component.
> For the ablation, we exchange the transformer-encoder in our approach with an MLP using skip connections and batch normalization and ensure that the transformer encoder and the MLP have approximately the same number of parameters. We then ran the previous experiments using the different models (GLMs, FA, GMMs) in three different scenarios each; every scenario has 50 synthetic and 17 real-world datasets.
>
> In summary, the results confirm our initial hypothesis that a transformer encoder is important for complex conditioning on the input data.
>
> For the GLM scenarios, this is especially pronounced with substantial performance drops across all scenarios when using the MLP baseline instead of the transformer. In the case of FA and GMMs, the improvement of the transformer-encoder is smaller, albeit consistent. In particular, there is no case where the MLP performance is above one standard error of the transformer’s performance. For FA and GMMs, the gap between the MLP and the transformer encoder is, for example, especially noticeable in scenarios 1 and 3 of the FA cases on the real-world data and in scenarios 1 and 2 for the GMM cases.
>
> If you have any remaining concerns or questions, please let us know.
>
> ### References
>
> [1] Dosovitskiy, Alexey, et al. "An Image is Worth 16x16 Words: Transformers for Image Recognition at Scale." International Conference on Learning Representations. 2020.
>
> [2] Dubey, Abhimanyu, et al. "The llama 3 herd of models." arXiv preprint arXiv:2407.21783 (2024).
>
> [3] Gorishniy, Yury, et al. "Revisiting deep learning models for tabular data." Advances in Neural Information Processing Systems 34 (2021): 18932-18943.
>
> [4] Müller, Samuel, et al. "Transformers Can Do Bayesian Inference." International Conference on Learning Representations.
>
> [5] Hollmann, Noah, et al. "TabPFN: A Transformer That Solves Small Tabular Classification Problems in a Second." The Eleventh International Conference on Learning Representations.
>
> [6] Panwar, Madhur, Kabir Ahuja, and Navin Goyal. "In-Context Learning through the Bayesian Prism." The Twelfth International Conference on Learning Representations.
>
> [7] Dong, Qingxiu, et al. "A survey on in-context learning." arXiv preprint arXiv:2301.00234 (2022).
>
> [8] Garg, Shivam, et al. "What can transformers learn in-context? a case study of simple function classes." Advances in Neural Information Processing Systems 35 (2022): 30583-30598.
>
> [9] Akyürek, Ekin, et al. "In-Context Language Learning: Architectures and Algorithms." Forty-first International Conference on Machine Learning.
>
> [10] Lee, Ivan, Nan Jiang, and Taylor Berg-Kirkpatrick. "Is attention required for ICL? Exploring the Relationship Between Model Architecture and In-Context Learning Ability." The Twelfth International Conference on Learning Representations.

---

> > ### Comment · Reviewer_prH2 · 2024-11-27
> >
> > Thank you for answering my question.

---

> > > ### Author Response · Authors · 2024-11-27
> > > **Do you have further concerns or questions?**
> > >
> > > Dear Reviewer prH2,
> > >
> > > We are happy to hear that we could answer your question. Do you have any further concerns or questions?
> > >
> > > If not, we would be grateful if you could consider raising the score.
> > >
> > > Thank you again for your insightful comments and active participation in the rebuttal.

---

### Official Review · Reviewer_EBeR · 2024-11-03

**Soundness:** 3
**Presentation:** 3
**Contribution:** 3
**Rating:** 8
**Confidence:** 4

**Summary:**

This paper introduces a novel approach to in-context learning that leverages transformer architectures and continuous normalized flow for a fully Bayesian inference framework. The main objective of this method is to simplify the conditions of fully bayesian inference by employing the in-context learning with transformer architecture, which is widely used in large language models and is recognized for its generalized capabilities across various tasks. Unlike existing algorithms that rely on pre-defined tractable distribution for latent variables, this approach enables the model to implicitly learn the relationship between latent and output variables. This implicit learning led by continuous normalizing flow aligns with both the data and a tractable distribution, thus removing the need for precise proxy distributions.

To validate the effectiveness of their method, the authors performed experiments on latent variable models, specifically GMM, Factor Analysis, and GLM. Results indicate that the in-context learning approach provides a more generalizable and efficient means of utilizing learned information across synthetic datasets, consistently achieving robust generalization on latent variable models. Further analysis of the marginal distributions demonstrates that the proposed method more closely aligns with the true latent distributions compared to existing approximate techniques.

**Strengths:**

- The paper offers comprehensive explanations of each concept, articulating the roles and functions of all elements with clarity. Additionally, it effectively contextualizes the research within the scope of existing studies, highlighting connections to prior work and establishing a solid foundation in the literature.

- The paper provides a clear account of the experimental design and the significance of the findings. By testing the model on diverse latent variable models, it illustrates the model’s applicability and potential impact, offering valuable insights into its relevance and effectiveness across multiple contexts.

**Weaknesses:**

- The paper does not clearly present the inherent limitations of its approach. Specifically, except for the GMM case, there is insufficient discussion regarding concerns and constraints related to model specification. I believe that this limitations are not fully addressed in the most cases. To strengthen the paper, it is essential to explicitly outline potential failure cases for this model and specify how it may underperform compared to existing models in certain scenarios by varying model specification. Without such clarification, there is a potential risk that the contributions of this paper may be overrated.

- Although the paper briefly mentions synthetic data, a more thorough exploration of design of synthetic datasets is essential. I observed simple describing synthetic data in appendix, however, it is important to consider how variations in the datasets or hyper-parameters impact model performance. Additionally, I believe that analysis of the model’s behavior in out-of-distribution scenarios would be valuable insights. Including these aspects would offer a more comprehensive understanding of the model’s robustness and applicability across diverse conditions.

**Questions:**

I would like to inquire whether a comparison with gradient descent-based VI approaches, such as Langevin dynamics[1] or stochastic variational inference[2], would be feasible in terms of evolving latent samples. Since this paper uses flow-matching to model changes, comparing the latent variable sampling steps with these gradient-based VI methods could provide further insights. Such a comparison may help demonstrate that the proposed approach more effectively captures variations in the latent variables, potentially highlighting its advantages in representing the implicit distribution of latent spaces.


**Reference**

[1] Welling, Max, and Yee W. Teh. "Bayesian learning via stochastic gradient Langevin dynamics." Proceedings of the 28th international conference on machine learning (ICML-11). 2011.

[2] Hoffman, Matthew D., et al. "Stochastic variational inference." Journal of Machine Learning Research (2013).

**Details Of Ethics Concerns:**

No concern

---

> ### Author Response · Authors · 2024-11-20
> **Response to weaknesses**
>
> Thank you for taking the time to read our manuscript and providing detailed feedback.
>
> We address your questions and comments below.  We use double quotation marks (") to mark parts directly quoted from the revised manuscript. In the manuscript itself, all added parts have the text color blue.
>
> > there is insufficient discussion regarding concerns and constraints related to model specification. I believe that this limitations are not fully addressed in the most cases.
>
> Regarding issues with model specification, we would like to clarify that the structure of the Bayesian model we want to learn in-context directly determines the distribution of synthetic data we have to use to fit the in-context learner. Our goal is to demonstrate that established GLM, GMM, and FA approaches with standard modeling assumptions and commonly used priors can be fitted in-context. Therefore, the generative processes are fixed for our analysis. Changing this process would result in models not estimating the correct posterior distribution and not allowing the access to a ground truth or gold standard model (as given by the analytic solution or the HMC in our experiments). In particular, this would mean that we cannot judge whether one approach works better or worse than another approach, as no reference distribution is available.
>
> We thank the reviewer for the comment and fully acknowledge that discussing the issue of model (mis-)specification is indeed very valuable. We, therefore, extended Section 3.3 to include a discussion regarding model misspecification for FA and GMMs.
>
> Below is the added text for your convenience:
>
> “To be able to access a reference or ground truth distribution, the data generating processes in our experiments need to match the structure of the GLM, FA and GMM approaches. While the generative processes of FA and GMMs directly prescribe how all parts of the data are generated, this can potentially cause a discrepancy between synthetically generated and real-world datasets. However, our empirical results (Section 4.1) demonstrate that the in-context learner can generalize to real-world data despite the discrepancy to the simulated datasets.”
>
> We further added a discussion of model misspecification and the associated limitations in the discussion part (Section 5) of our manuscript. Below is the added text quoted from the revised manuscript for your convenience:
>
> “Even though our experiments show that ICL works well despite being trained on data that is potentially very different from real-world data, the approach will only be as flexible as the data and model structures it was trained on. As a result, ICL might fail if the model, which implies the synthetic data generation, is severely misspecified. However, this is the same limitation as when misspecifying the hypothesis space of, e.g., a deep neural network or other machine learning approaches, effectively providing the model with the wrong inductive bias.”
>
> > Although the paper briefly mentions synthetic data, a more thorough exploration of the design of synthetic datasets is essential.
>
> Key aspects of this point have been answered in our response above. The structure of the standard GLM, GMM and FA approaches directly determines how the synthetic data has to be generated, where GLMs are somewhat of an exception. As noted above, we will make this more clear in a revised manuscript version and thank the reviewer for bringing up this important point.
>
> Thank you again for your constructive feedback. Please let us know if our response addresses your questions and concerns, and if there is anything else we need to clarify.

---

> > ### Author Response · Authors · 2024-11-20
> > **Response to the question**
> >
> > > I would like to inquire whether a comparison with gradient descent-based VI approaches, such as Langevin dynamics[1] or stochastic variational inference[2], would be feasible in terms of evolving latent samples.
> >
> > We would like to point out that our benchmarks already include state-of-the-art gradient-based VI approaches using variational inference as introduced in [2]. However, because the considered datasets comfortably fit into the RAM of our hardware, we can afford full-batch gradient steps.
> >
> > Furthermore, we understand the stochastic gradient Langevin dynamics (SGLD) approach proposed in [1] to be intended for sample-based inference and not for stochastic VI. In order to compare our method with this approach we are currently working on implementing and benchmarking an SGLD approach and we will post the results and add them to the paper as soon as we have them.
> >
> > Thank you again for your constructive feedback. Please let us know if our response addresses your questions and concerns, and if there is anything else we need to clarify.
> >
> > ### References
> > [1] Welling, Max, and Yee W. Teh. "Bayesian learning via stochastic gradient Langevin dynamics." Proceedings of the 28th international conference on machine learning (ICML-11). 2011.
> >
> > [2] Hoffman, Matthew D., et al. "Stochastic variational inference." Journal of Machine Learning Research (2013).

---

> > > ### Author Response · Authors · 2024-11-21
> > > **Additional Results for SGLD**
> > >
> > > Thank you again for proposing to additionally compare the samples from our ICL approach against SGLD. We conducted an extensive benchmark study comparing samples generated via ICL and SGLD. We use the same setup used to compare ICL against the VI methods.
> > >
> > > In summary, we find that across six different GLM scenarios, six different FA scenarios, and four different GMM scenarios with 50 synthetic and 17 real-world datasets each, ICL yields consistently better samples than SGLD. This is somewhat expected since SGLD is primarily designed as a scalable method that is also applicable to large datasets [4]. Numerous theoretical and empirical findings confirm that, even though SGLD is computationally inexpensive and very scalable, it is substantially outperformed by, for instance, HMC, in terms of sample quality, which is especially pronounced when the posterior distributions are complex and parameters are correlated [3, 4, 5, 6, 7] .
> > >
> > > Thank you again for your constructive feedback. Please let us know if our response addresses your questions and concerns, and if there is anything else we need to clarify.
> > >
> > > ### References
> > > [3] Chen, Tianqi, et al. "Stochastic gradient hamiltonian monte carlo." International conference on machine learning. PMLR, 2014.
> > >
> > > [4] Izmailov, Pavel, et al. "What are Bayesian neural network posteriors really like?." International conference on machine learning. PMLR, 2021.
> > >
> > > [5] Li, Chunyuan, et al. "Preconditioned stochastic gradient Langevin dynamics for deep neural networks." Proceedings of the AAAI conference on artificial intelligence. Vol. 30. No. 1. 2016.
> > >
> > > [6] Brosse, Nicolas, et al.. "The promises and pitfalls of stochastic gradient Langevin dynamics." Advances in Neural Information Processing Systems 31 (2018).
> > >
> > > [7] Mangoubi, Oren, et al.. "Nonconvex sampling with the Metropolis-adjusted Langevin algorithm." Conference on learning theory. PMLR, 2019.

---

> ### Author Response · Authors · 2024-11-21
> **Additional Results for SGLD: Table 1**
>
> For your convenience, we also include the results of benchmarking SGLD below. (Please also refer to Appendix F in the revised manuscript) .
>
>
> #### SGLD vs. ICL: Evaluation on 50 synthetic and 17 real-world datasets for six different GLM scenarios. All results within two standard errors of the best average result for each scenario are marked in **bold**.
>
> | **Scenario** | **Model** | **Synthetic Evaluation** | **Synthetic Evaluation** | **Synthetic Evaluation** | **Real-World Evaluation** | **Real-World Evaluation** | **Real-World Evaluation** |
> |--------------|-----------|--------------------------|--------------------------|--------------------------|---------------------------|---------------------------|---------------------------|
> |          	|       	| C2ST (↓)            	| MMD (↓)             	| W2 (↓)             	| C2ST (↓)             	| MMD (↓)              	| W2             	|
> | **Scenario 1** | SGLD  	| 0.992 (± 0.015)     	| 2.846 (± 1.411)     	| 1.951 (± 0.917)     	| 0.980 (± 0.013)      	| 2.191 (± 1.183)      	| 0.865 (± 0.438)      	|
> |          	| **ICL**   | **0.765** (± 0.123) 	| **0.767** (± 0.727) 	| **0.585** (± 0.301) 	| **0.614** (± 0.074)  	| **0.175** (± 0.219)  	| **0.310** (± 0.138)  	|
> | **Scenario 2** | SGLD  	| 0.999 (± 0.004)     	| 5.650 (± 1.762)     	| 8.295 (± 5.629)     	| 0.994 (± 0.006)      	| 2.699 (± 1.093)      	| 1.289 (± 0.454)      	|
> |          	| **ICL**   | **0.839** (± 0.072) 	| **0.707** (± 0.658) 	| **1.111** (± 0.300) 	| **0.768** (± 0.033)  	| **0.143** (± 0.089)  	| **0.411** (± 0.094)  	|
> | **Scenario 3** | SGLD  	| 0.997 (± 0.008)     	| 3.320 (± 1.595)     	| 3.011 (± 1.036)     	| 0.983 (± 0.013)      	| 2.152 (± 1.194)      	| 0.935 (± 0.523)      	|
> |          	| **ICL**   | **0.611** (± 0.070) 	| **0.089** (± 0.114) 	| **0.423** (± 0.348) 	| **0.576** (± 0.027)  	| **0.037** (± 0.026)  	| **0.257** (± 0.044)  	|
> | **Scenario 4** | SGLD  	| 1.000 (± 0.000)     	| 6.626 (± 1.215)     	| 15.674 (± 8.100)    	| 0.994 (± 0.006)      	| 2.927 (± 1.564)      	| 1.606 (± 1.022)      	|
> |          	| **ICL**   | **0.753** (± 0.049) 	| **0.171** (± 0.153) 	| **0.631** (± 0.294) 	| **0.762** (± 0.015)  	| **0.105** (± 0.046)  	| **0.597** (± 0.104)  	|
> | **Scenario 5** | SGLD  	| 0.999 (± 0.003)     	| 3.308 (± 1.728)     	| 2.216 (± 1.247)     	| 1.000 (± 0.000)      	| 4.012 (± 1.413)      	| 0.996 (± 0.406)      	|
> |          	| **ICL**   | **0.621** (± 0.063) 	| **0.067** (± 0.080) 	| **0.299** (± 0.195) 	| **0.610** (± 0.045)  	| **0.046** (± 0.020)  	| **0.242** (± 0.038)  	|
> | **Scenario 6** | SGLD  	| 0.998 (± 0.001)     	| 2.681 (± 0.565)     	| 2.419 (± 0.510)     	| 0.998 (± 0.002)      	| 2.845 (± 0.590)      	| 1.851 (± 0.319)      	|
> |          	| **ICL**   | **0.532** (± 0.019) 	| **0.016** (± 0.008) 	| **0.590** (± 0.066) 	| **0.556** (± 0.017)  	| **0.035** (± 0.015)  	| **0.504** (± 0.038)  	|

---

> > ### Author Response · Authors · 2024-11-21
> > **Additional Results for SGLD: Table 2**
> >
> > #### SGLD vs. ICL: Evaluation on 50 synthetic and 17 real-world datasets for six different FA scenarios.  All results within two standard errors of the best average result for each scenario are marked in **bold**.
> > | **Scenario** | **Model** | **Synthetic Evaluation** | **Synthetic Evaluation** | **Synthetic Evaluation** | **Real-World Evaluation** | **Real-World Evaluation** | **Real-World Evaluation** |
> > |--------------|-----------|--------------------------|--------------------------|--------------------------|---------------------------|---------------------------|---------------------------|
> > |          	|       	| C2ST (↓)            	| MMD (↓)             	| W2 (↓)             	| C2ST (↓)             	| MMD (↓)              	| W2 (↓)               	|
> > | **Scenario 1** | SGLD  	| 0.996 (± 0.006)     	| 2.883 (± 1.552)     	| 1.776 (± 0.694)     	| 0.995 (± 0.003)      	| 2.676 (± 0.710)      	| 1.608 (± 0.381)      	|
> > |          	| **ICL**   | **0.552** (± 0.028) 	| **0.034** (± 0.034) 	| **0.289** (± 0.083) 	| **0.606** (± 0.038)  	| **0.068** (± 0.069)  	| **0.265** (± 0.078)  	|
> > | **Scenario 2** | SGLD  	| 0.997 (± 0.003)     	| 2.950 (± 0.786)     	| 1.892 (± 0.533)     	| 0.995 (± 0.003)      	| 2.517 (± 0.583)      	| 1.500 (± 0.268)      	|
> > |          	| **ICL**   | **0.542** (± 0.006) 	| **0.017** (± 0.006) 	| **0.244** (± 0.033) 	| **0.622** (± 0.032)  	| **0.098** (± 0.039)  	| **0.287** (± 0.046)  	|
> > | **Scenario 3** | SGLD  	| 0.998 (± 0.005)     	| 3.662 (± 1.099)     	| 2.086 (± 0.919)     	| 0.956 (± 0.025)      	| 1.580 (± 0.819)      	| 0.311 (± 0.108)      	|
> > |          	| **ICL**   | **0.537** (± 0.023) 	| **0.024** (± 0.021) 	| **0.259** (± 0.088) 	| **0.609** (± 0.019)  	| **0.124** (± 0.037)  	| **0.179** (± 0.018)  	|
> > | **Scenario 4** | SGLD  	| 1.000 (± 0.000)     	| 4.127 (± 0.635)     	| 3.047 (± 0.972)     	| **0.950** (± 0.021)  	| **1.520** (± 0.512)  	| **0.141** (± 0.031)  	|
> > |          	| **ICL**   | **0.684** (± 0.060) 	| **0.198** (± 0.141) 	| **0.918** (± 0.246) 	| 0.988 (± 0.003)      	| 1.764 (± 0.026)      	| 1.248 (± 0.008)      	|
> > | **Scenario 5** | SGLD  	| 0.999 (± 0.001)     	| 3.465 (± 0.939)     	| 1.981 (± 0.938)     	| 0.962 (± 0.024)      	| 1.945 (± 1.383)      	| **0.393** (± 0.243)  	|
> > |          	| **ICL**   | **0.535** (± 0.016) 	| **0.021** (± 0.011) 	| **0.279** (± 0.060) 	| **0.886** (± 0.017)  	| **1.207** (± 0.101)  	| 1.002 (± 0.042)      	|
> > | **Scenario 6** | SGLD  	| 0.997 (± 0.004)     	| 3.395 (± 1.199)     	| 2.358 (± 1.458)     	| 0.950 (± 0.040)      	| 2.177 (± 1.643)      	| 0.342 (± 0.224)      	|
> > |          	| **ICL**   | **0.543** (± 0.021) 	| **0.023** (± 0.015) 	| **0.345** (± 0.173) 	| **0.666** (± 0.020)  	| **0.200** (± 0.034)  	| **0.224** (± 0.014)  	|

---

> ### Author Response · Authors · 2024-11-21
> **Additional Results for SGLD: Table 3**
>
> #### SGLD vs. ICL: Evaluation on 50 synthetic and 17 real-world datasets for four different GMM scenarios. All results within two standard errors of the best average result for each scenario are marked in **bold**.
>
> | **Scenario** | **Model** | **Synthetic Evaluation** | **Synthetic Evaluation** | **Synthetic Evaluation** | **Real-World Evaluation** | **Real-World Evaluation** | **Real-World Evaluation** |
> |--------------|-----------|--------------------------|--------------------------|--------------------------|---------------------------|---------------------------|---------------------------|
> |          	|       	| C2ST (↓)            	| MMD (↓)             	| W2 (↓)             	| C2ST (↓)             	| MMD (↓)              	| W2 (↓)              	|
> | **Scenario 1** | SGLD  	| 1.000 (± 0.001)     	| 2.629 (± 0.868)     	| 3.279 (± 1.330)     	| 1.000 (± 0.000)      	| 3.421 (± 0.877)      	| 6.510 (± 1.763)      	|
> |          	| **ICL**   | **0.760** (± 0.092) 	| **0.303** (± 0.548) 	| **2.095** (± 1.692) 	| **0.847** (± 0.082)  	| **0.486** (± 0.623)  	| **4.054** (± 2.782)  	|
> | **Scenario 2** | SGLD  	| 1.000 (± 0.000)     	| 3.046 (± 1.041)     	| 6.015 (± 4.265)     	| 1.000 (± 0.000)      	| 2.487 (± 0.521)      	| 6.858 (± 1.618)      	|
> |          	| **ICL**   | **0.812** (± 0.061) 	| **0.159** (± 0.154) 	| **2.314** (± 0.926) 	| **0.937** (± 0.041)  	| **0.282** (± 0.131)  	| **3.947** (± 1.055)  	|
> | **Scenario 3** | SGLD  	| 1.000 (± 0.000)     	| 4.631 (± 1.169)     	| 23.247 (± 30.646)   	| 1.000 (± 0.000)      	| 2.655 (± 0.437)      	| 26.356 (± 2.699)     	|
> |          	| **ICL**   | **1.000** (± 0.000) 	| **0.582** (± 0.280) 	| **8.708** (± 4.945) 	| **1.000** (± 0.000)  	| **1.869** (± 0.342)  	| **33.230** (± 8.095) 	|
> | **Scenario 4** | SGLD  	| **1.000** (± 0.000) 	| 3.464 (± 1.098)     	| **6.995** (± 5.554) 	| **1.000** (± 0.000)  	| **2.555** (± 0.494)  	| **9.477** (± 3.432)  	|
> |          	| **ICL**   | **1.000** (± 0.000) 	| 2.451 (± 0.868)     	| **8.333** (± 4.202) 	| 1.000 (± 0.000)      	| **2.518** (± 0.694)  	| **11.938** (± 2.956) 	|

---

> ### Comment · Reviewer_EBeR · 2024-11-21
> **Response to authors rebuttal**
>
> I appreciate the updates in the revised manuscript, particularly the inclusion of discussions the possibility of misspecification and the more detailed comparisons between gradient methods and the proposed approach. The presented results appear promising and provide meaningful statements considering the standard of ICLR 2025.
>
> However, there remains a significant gap in addressing the generalizability of the proposed method to novel data. Specifically, the authors should demonstrate how the method performs when there is a slight increase in the discrepancy between training and test data, and whether such shifts result in a notable drop in performance. This would provide more clarity on the robustness of the method.
>
> Additionally, I am concerned about the reproducibility. While the updated experimental sections seem valuable for readers who explore the method, implementation does not still reflect the updates in the rebuttal phase.
>
> Depending on the updates, I am inclined to raise my score from 6 to 8. I continue to monitor their responses.

---

> > ### Author Response · Authors · 2024-11-22
> > **Results for train-test mismatch**
> >
> > Dear Reviewer EBeR,
> >
> > Thank you for your prompt response! We are pleased that our results “appear promising” and “provide meaningful statements considering the standard of ICLR 2025”.
> >
> > We are further grateful for the reminder to update the anonymized repository. We have now uploaded the code and configuration files to run all additional benchmarks and ablation studies from the rebuttal and provide a short explanation in the readme file on how to reproduce the results.
> >
> > ### Discrepancy between train and test data
> >
> > Following up on your concerns regarding the performance of our method when faced with a discrepancy in the distribution of training and test data, we conducted an additional new ablation study investigating this situation.
> > For this, we use our in-context learner trained on a specific generative process and then test the approach on data stemming from a different generative process. We consider in total 20 different setups for GLMS, FA and GMMs, and vary several aspects of the data-generating processes to create a mismatch between train- and test distributions. For GLMs, these aspects include a mismatch in the mean or the variance of the coefficients’ distribution and the variance of the additive noise. For FA, we investigate discrepancies in the train and test data by altering the variance of the factor loadings. For the GMMs, we vary the uniformity in the number of samples per cluster, the variance in the covariances of the mixture components, as well as the variance of the mean vectors.
> >
> > **Results**
> >
> > While we find that the performance of the in-context learner can decrease in specific setups, the overall sample quality remains stable across all cases. When the performance decreases, the drop is usually moderate and often not significant.
> >
> > We thank you again for your valuable feedback. We think that especially these new results underline the strength and flexibility of our method. Please let us know if our response addresses your concerns, and if there is anything else left to clarify.

---

> > > ### Author Response · Authors · 2024-11-22
> > > **Result tables**
> > >
> > > For your convenience, we also include the results (in Appendix G in the revised manuscript) below. In Appendix G, we also include a description of the different considered scenarios.
> > >
> > > #### OOD Performance: Evaluation on 50 synthetic datasets for 8 different GLM scenarios. All results within two standard errors of the non-OOD result for each scenario are marked in **bold**.
> > >
> > > | **Scenario**  | C2ST (↓)           	| MMD (↓)           	| W2 (↓)             	|
> > > |---------------|-------------------------|------------------------|------------------------|
> > > | Scenario 2	| **0.839** (± 0.072)	| **0.707** (± 0.658)   | **1.111** (± 0.300)   |
> > > | Scenario 2.B  | **0.809** (± 0.055)	| **0.410** (± 0.095)   | **2.250** (± 0.916)   |
> > > | Scenario 2.C  | **0.857** (± 0.105)	| **0.634** (± 0.318)   | **3.067** (± 1.759)   |
> > > |---------------|-------------------------|------------------------|------------------------|
> > > | Scenario 2	| **0.839** (± 0.072)	| **0.707** (± 0.658)   | **1.111** (± 0.300)   |
> > > | Scenario 2.D  | **0.840** (± 0.109)	| **0.916** (± 1.123)   | **4.007** (± 3.261)   |
> > > | Scenario 2.E  | **0.932** (± 0.120)	| **1.556** (± 1.127)   | **4.850** (± 2.261)   |
> > > |---------------|-------------------------|------------------------|------------------------|
> > > | Scenario 3	| **0.611** (± 0.070)	| **0.089** (± 0.114)   | **0.423** (± 0.348)   |
> > > | Scenario 3.B  | **0.667** (± 0.080)	| **0.210** (± 0.117)   | 1.172 (± 0.258)   	|
> > > | Scenario 3.C  | **0.720** (± 0.108)	| **0.362** (± 0.248)   | 1.891 (± 0.678)   	|
> > > |---------------|-------------------------|------------------------|------------------------|
> > > | Scenario 5	| **0.621** (± 0.063)	| **0.067** (± 0.080)   | **0.299** (± 0.195)   |
> > > | Scenario 5.B  | 0.831 (± 0.121)    	| 0.479 (± 0.200)   	| 1.762 (± 0.541)   	|
> > > | Scenario 5.C  | 0.920 (± 0.064)    	| 0.753 (± 0.424)   	| 3.159 (± 1.254)   	|
> > >
> > >
> > > #### OOD Performance: Evaluation on 50 synthetic datasets for 6 different FA scenarios.All results within two standard errors of the non-OOD result for each scenario are marked in **bold**.
> > >
> > > | **Scenario**  | C2ST (↓)           	| MMD (↓)           	| W2 (↓)             	|
> > > |---------------|-------------------------|------------------------|------------------------|
> > > | Scenario 1	| **0.552** (± 0.028)	| **0.034** (± 0.034)   | **0.289** (± 0.083)   |
> > > | Scenario 1.B  | 0.826 (± 0.066)    	| 0.656 (± 0.384)   	| 0.929 (± 0.321)   	|
> > > | Scenario 1.C  | 0.855 (± 0.060)    	| 0.837 (± 0.494)   	| 1.135 (± 0.461)   	|
> > > |---------------|-------------------------|------------------------|------------------------|
> > > | Scenario 2	| **0.542** (± 0.006)	| **0.017** (± 0.006)   | **0.244** (± 0.033)   |
> > > | Scenario 2.B  | 0.580 (± 0.069)    	| 0.087 (± 0.191)   	| 0.393 (± 0.291)   	|
> > > | Scenario 2.C  | 0.589 (± 0.076)    	| 0.089 (± 0.113)   	| 0.446 (± 0.233)   	|
> > > |---------------|-------------------------|------------------------|------------------------|
> > > | Scenario 3	| **0.537** (± 0.023)	| **0.024** (± 0.021)   | **0.259** (± 0.088)   |
> > > | Scenario 3.B  | **0.544** (± 0.028)	| 0.030 (± 0.021)   	| **0.285** (± 0.094)   |
> > > | Scenario 3.C  | **0.533** (± 0.025)	| 0.021 (± 0.015)   	| **0.347** (± 0.152)   |
> > >
> > >
> > > #### OOD Performance: Evaluation on 50 synthetic datasets for 6 different GMM scenarios.All results within two standard errors of the non-OOD result for each scenario are marked in **bold**.
> > >
> > > | **Scenario**  | C2ST (↓)           	| MMD (↓)           	| W2 (↓)             	|
> > > |---------------|-------------------------|------------------------|------------------------|
> > > | Scenario 2	| **0.812** (± 0.061)	| **0.159** (± 0.154)   | **2.314** (± 0.926)   |
> > > | Scenario 2.B  | **0.829** (± 0.050)	| **0.233** (± 0.161)   | **2.595** (± 0.998)   |
> > > | Scenario 2.C  | **0.816** (± 0.057)	| 0.149 (± 0.135)   	| 2.272 (± 0.654)   	|
> > > |---------------|-------------------------|------------------------|------------------------|
> > > | Scenario 2	| **0.812** (± 0.061)	| **0.159** (± 0.154)   | **2.314** (± 0.926)   |
> > > | Scenario 2.D  | **0.812** (± 0.076)	| **0.148** (± 0.091)   | **2.557** (± 0.837)   |
> > > | Scenario 2.E  | **0.880** (± 0.057)	| **0.231** (± 0.109)   | **3.535** (± 1.003)   |
> > > |---------------|-------------------------|------------------------|------------------------|
> > > | Scenario 2	| **0.812** (± 0.061)	| **0.159** (± 0.154)   | **2.314** (± 0.926)   |
> > > | Scenario 2.F  | **0.821** (± 0.076)	| **0.216** (± 0.214)   | **2.700** (± 1.044)   |
> > > | Scenario 2.G  | **0.844** (± 0.046)	| **0.197** (± 0.124)   | **2.675** (± 0.552)   |

---

> > > > ### Comment · Reviewer_EBeR · 2024-11-22
> > > > **Official comment to authors response**
> > > >
> > > > I appreciate the updates and I am pleased to see that your implementation includes new experimental results such as SGLD and OOD settings. All my concerns has been addressed, so I have raised my score from 6 to 8.

---

### Official Review · Reviewer_rWy9 · 2024-11-04

**Soundness:** 2
**Presentation:** 3
**Contribution:** 2
**Rating:** 5
**Confidence:** 3

**Summary:**

This paper explores using in-context learning within transformers to perform full Bayesian inference on complex statistical models. The authors propose a framework leveraging continuous normalizing flows and flow matching, enabling transformers to infer posterior distributions in-context without explicit parameter updates. The approach is applied to GLMs, latent factor models, and GMMs, which often require techniques like Hamiltonian Monte Carlo (HMC) or variational inference (VI). The reliance on synthetic data for training may introduce biases when tested on real-world distributions that differ from synthetic priors. The results highlight ICL's potential as an efficient, flexible Bayesian inference tool that can operate directly on a diverse range of models and posterior complexities. This paper opens up further exploration into using ICL for full probabilistic modeling, potentially extending its use beyond traditional Bayesian inference.

**Strengths:**

1. Innovative Framework for Bayesian Inference: The paper introduces a novel approach to using in-context learning (ICL) with transformers for Bayesian inference. By employing continuous normalizing flows (CNFs) and flow matching, the framework effectively handles high-dimensional posterior distributions, making it relevant for complex statistical models like generalized linear models (GLMs) and Gaussian mixture models (GMMs).

2. Strong Empirical Comparisons: The authors conduct extensive experiments, benchmarking ICL-based Bayesian inference against both Hamiltonian Monte Carlo (HMC) and various variational inference (VI) methods. Their approach shows promising results, especially in scenarios where VI models struggle with multimodality or non-normal distributions.

3. Real-World Applications: This paper evaluates performance on real-world datasets, enhancing the practical applicability of the proposed method. This broadens the scope of ICL by demonstrating its potential to rival or even outperform established inference techniques on complex, non-synthetic data.

**Weaknesses:**

1. Computational Expense and Scalability Issues. While ICL may yield accurate approximations of posterior distributions, it requires a high computational investment in training. This can limit the approach’s scalability, especially in real-world applications requiring quick adaptability across various data regimes. The paper's acknowledgment of this limitation highlights the need for optimization and scalability improvements, which were not sufficiently addressed.

2. Lack of Detailed Architectural Justification. Although the authors introduce a transformer-based architecture with flow matching for Bayesian inference, the rationale for specific architectural choices, such as the reliance on continuous normalizing flows over other methods, is underexplored. This omission leaves unanswered questions regarding the necessity or superiority of these choices compared to simpler or more established architectures for similar tasks.

**Questions:**

N/A. See weakness.

---

> ### Author Response · Authors · 2024-11-20
> **Response to weakness 1**
>
> Thank you for taking the time to read our manuscript and providing detailed feedback.
>
> We address your questions and comments below.  We use double quotation marks (") to mark parts directly quoted from the revised manuscript. In the manuscript itself, all added parts have the text color blue.
>
> > While ICL may yield accurate approximations of posterior distributions, it requires a high computational investment in training.
>
> In-context learning is inherently expensive during the pre-training phase and our method is no exception—the arguably most prominent in-context learners, Large Language Models, are currently even trained on more than ten thousand GPUs in order to achieve state-of-the-art performance [1].
> We have revised the limitations section in section 5 of the manuscript to emphasize this aspect even more clearly.
> It is noteworthy, however, that when practically applied to a new dataset, our approach can sample the posterior distribution very quickly. To substantiate this claim, we have conducted additional runtime comparisons of all methods. While some VI methods are faster than ICL, at the cost of inferior performance, our approach is consistently faster than HMC (for FA even achieving a factor 8 of speed-up).
>
> For your convenience, we also include the results (in Appendix E in the revised manuscript) below.
> | **Scenario** | **Method**              | **Mean Runtime (s)**       |
> |--------------|--------------------------|----------------------------|
> | **GLM**      | Laplace Approximation    | 10.48 ± 0.25          |
> |              | VI: DiagonalNormal       | 12.02 ± 0.26           |
> |              | VI: MultivariateNormal   | 13.70 ± 0.29          |
> |              | VI: Structured Normal    | 19.81 ± 0.98               |
> |              | VI: IAF                  | 15.44 ± 0.30               |
> |              | HMC                      | 120.24 ± 13.94             |
> |              | **ICL**                  | 107.79 ± 17.36             |
> | **FA**       | Laplace Approximation    | 17.85 ± 0.21           |
> |              | VI: DiagonalNormal       | 20.94 ± 0.66           |
> |              | VI: MultivariateNormal   | 20.84 ± 0.28          |
> |              | VI: Structured Normal    | 36.17 ± 0.61               |
> |              | VI: IAF                  | 23.75 ± 0.38               |
> |              | HMC                      | 248.26 ± 57.88             |
> |              | **ICL**                  | 31.49 ± 4.97               |
> | **GMM**      | Laplace Approximation    | 27.52 ± 0.40           |
> |              | VI: DiagonalNormal       | 29.74 ± 0.57           |
> |              | VI: MultivariateNormal   | 30.50 ± 0.41           |
> |              | VI: Structured Normal    | 42.44 ± 0.44               |
> |              | VI: IAF                  | 33.39 ± 0.49               |
> |              | HMC                      | 239.67 ± 32.71             |
> |              | **ICL**                  | 93.88 ± 10.47              |
>
>
> Thank you again for your constructive feedback. Please let us know if our response addresses your questions and concerns, and if there is anything else we need to clarify.

---

> ### Author Response · Authors · 2024-11-20
> **Response to weakness 2**
>
> > the rationale for specific architectural choices, such as the reliance on continuous normalizing flows over other methods, is underexplored.
>
> While exploring other architectural choices is an interesting avenue, we would like to note that [2] introduces theoretical results backing the choice of continuous normalizing flows over, for instance, various variants of Diffusion methods. They also empirically find flow matching to be a more robust and efficient objective. [3] confirms these results across several simulation-based inference tasks that are closely related to those in our paper.
>
> To again substantiate our claims, we added additional empirical results to the manuscript by replacing flow matching based on optimal transport (OT) paths in our approach with a diffusion objective based on variance preserving (VP) diffusion probability paths from [4]. When comparing samples based on the VP diffusion probability paths and our initial flow matching implementation with OT transport paths, we find that our original approach consistently yields superior samples.
>
> Thank you again for your constructive feedback. Please let us know if our response addresses your questions and concerns, and if there is anything else we need to clarify.
>
> ### References
>
> [1] Dubey, Abhimanyu, et al. "The llama 3 herd of models." arXiv preprint arXiv:2407.21783 (2024).
>
> [2] Lipman, Yaron, et al. "Flow matching for generative modeling." arXiv preprint arXiv:2210.02747 (2022).
>
> [3] Wildberger, Jonas, et al. "Flow matching for scalable simulation-based inference." Advances in Neural Information Processing Systems 36 (2024).
>
> [4] Song, Yang, et al. "Score-based generative modeling through stochastic differential equations." arXiv preprint arXiv:2011.13456 (2020).

---

> ### Author Response · Authors · 2024-11-20
> **Response to weakness 2: results**
>
> For your convenience, we also include the results when comparing samples based on the VP diffusion probability paths and our initial flow matching implementation with OT transport paths (in Appendix C in the revised manuscript) below.
>
> #### GLMs: Comparison of the OT flow matching and the VP diffusion objective on 50 synthetic and 17 real-world datasets for three different scenarios. All results within two standard errors of the best average result for each scenario are marked in **bold**.
>
> | Scenario   | Model         	| C2ST (↓)       	| MMD (↓)        	| W2 (↓)         	| C2ST (↓)       	| MMD (↓)        	| W2 (↓)         	|
> |------------|-------------------|--------------------|--------------------|--------------------|--------------------|--------------------|--------------------|
> | **Scenario 2** | Diffusion paths  | 0.961 (± 0.040)	| **1.525** (± 0.777) | 3.354 (± 1.333)	| 0.961 (± 0.016)	| 1.347 (± 0.365)	| 2.025 (± 0.270)	|
> |        	| **OT paths**   	| **0.839** (± 0.072) | **0.707** (± 0.658) | **1.111** (± 0.300) | **0.768** (± 0.033) | **0.143** (± 0.089) | **0.411** (± 0.094) |
> | **Scenario 3** | Diffusion paths  | 0.903 (± 0.111)	| 1.080 (± 0.564)	| 1.733 (± 0.408)	| 0.936 (± 0.013)	| 1.002 (± 0.203)	| 1.442 (± 0.103)	|
> |        	| **OT paths**   	| **0.611** (± 0.070) | **0.089** (± 0.114) | **0.423** (± 0.348) | **0.576** (± 0.027) | **0.037** (± 0.026) | **0.257** (± 0.044) |
> | **Scenario 5** | Diffusion paths  | **0.691** (± 0.074) | 0.211 (± 0.143)	| 0.708 (± 0.233)	| **0.681** (± 0.038) | 0.182 (± 0.093)	| **0.554** (± 0.090) |
> |        	| **OT paths**   	| **0.621** (± 0.063) | **0.067** (± 0.080) | **0.299** (± 0.195) | **0.610** (± 0.045) | **0.046** (± 0.020) | **0.242** (± 0.038) |
>
> #### FA: Comparison of the OT flow matching and the VP diffusion objective on 50 synthetic and 17 real-world datasets for three different scenarios. All results within two standard errors of the best average result for each scenario are marked in **bold**.
>
> | Scenario   | Model         	| C2ST (↓)       	| MMD (↓)        	| W2 (↓)         	| C2ST (↓)       	| MMD (↓)        	| W2 (↓)         	|
> |------------|-------------------|--------------------|--------------------|--------------------|--------------------|--------------------|--------------------|
> | **Scenario 1** | Diffusion paths  | 0.622 (± 0.043)	| 0.207 (± 0.121)	| 0.692 (± 0.192)	| **0.595** (± 0.012) | 0.089 (± 0.011)	| 0.475 (± 0.019)	|
> |        	| **OT paths**   	| **0.552** (± 0.028) | **0.034** (± 0.034) | **0.289** (± 0.083) | **0.606** (± 0.038) | **0.068** (± 0.069) | **0.265** (± 0.078) |
> | **Scenario 2** | Diffusion paths  | 0.826 (± 0.036)	| 0.768 (± 0.238)	| 1.219 (± 0.276)	| 0.878 (± 0.028)	| 0.793 (± 0.154)	| 1.056 (± 0.084)	|
> |        	| **OT paths**   	| **0.542** (± 0.006) | **0.017** (± 0.006) | **0.244** (± 0.033) | **0.622** (± 0.032) | **0.098** (± 0.039) | **0.287** (± 0.046) |
> | **Scenario 3** | Diffusion paths  | 0.751 (± 0.048)	| 0.387 (± 0.216)	| 0.834 (± 0.163)	| 0.944 (± 0.008)	| 1.514 (± 0.056)	| 1.332 (± 0.028)	|
> |        	| **OT paths**   	| **0.537** (± 0.023) | **0.024** (± 0.021) | **0.259** (± 0.088) | **0.609** (± 0.019) | **0.124** (± 0.037) | **0.179** (± 0.018) |
>
> #### GMMs: Comparison of the OT flow matching and the VP diffusion objective on 50 synthetic and 17 real-world datasets for three different scenarios. All results within two standard errors of the best average result for each scenario are marked in **bold**.
>
> | Scenario   | Model         	| C2ST (↓)       	| MMD (↓)        	| W2 (↓)         	| C2ST (↓)       	| MMD (↓)        	| W2 (↓)         	|
> |------------|-------------------|--------------------|--------------------|--------------------|--------------------|--------------------|--------------------|
> | **Scenario 1** | Diffusion paths  | 0.924 (± 0.024)	| **0.241** (± 0.381) | **2.195** (± 1.431) | 0.958 (± 0.030)	| 0.890 (± 0.912)	| **5.328** (± 2.544) |
> |        	| **OT paths**   	| **0.760** (± 0.092) | **0.303** (± 0.548) | **2.095** (± 1.692) | **0.847** (± 0.082) | **0.486** (± 0.623) | **4.054** (± 2.782) |
> | **Scenario 2** | Diffusion paths  | 0.942 (± 0.020)	| **0.213** (± 0.187) | **2.748** (± 0.659) | **0.984** (± 0.012) | **0.411** (± 0.162) | **5.397** (± 1.458) |
> |        	| **OT paths**   	| **0.812** (± 0.061) | **0.159** (± 0.154) | **2.314** (± 0.926) | **0.937** (± 0.041) | **0.282** (± 0.131) | **3.947** (± 1.055) |
> | **Scenario 3** | Diffusion paths  | **1.000** (± 0.000) | 0.582 (± 0.280)	| **8.708** (± 4.945) | **1.000** (± 0.000) | 1.869 (± 0.342)	| **33.230** (± 8.095) |
> |        	| **OT paths**   	| **0.999** (± 0.001) | **0.267** (± 0.154) | **7.234** (± 2.974) | **1.000** (± 0.000) | **1.155** (± 0.258) | **26.956** (± 3.114) |

---

> > ### Author Response · Authors · 2024-11-27
> > **Response to weakness 2 contd.: reasons for choosing a transformer**
> >
> > Dear Reviewer rWy9,
> >
> > Besides the ablation we provided for your initial review regarding the diffusion objective, we have now also conducted an extensive additional ablation study for the transformer encoder, a key component in our ICL approach.
> >
> > We think that providing further conceptual justification as well as empirical evidence on this choice is important for addressing your concern that
> >
> > > the rationale for specific architectural choices [...] is underexplored
> >
> > **Conceptual reasons**
> >
> > There are several conceptual reasons why transformers are central to our approach:
> >
> > Transformers are a widely used baseline architecture that perform exceptionally well across various domains, including computer vision, natural language processing (NLP), and tabular data [1,2,3]. Moreover, the PFN methodology introduced in [4,5], which forms a cornerstone of our approach, is built directly on the transformer architecture. Consequently, employing a transformer is both a logical and appropriate choice for our proposal, which focuses on extending [4,5] to showcase the feasibility of in-context learning for full Bayesian inference.
> > Additionally, the concept of in-context learning, which we aim to generalize to full Bayesian inference, is intrinsically linked to the transformer architecture [6,7,8]. This connection is further supported by research highlighting the critical role of the attention mechanism in enabling in-context learning [9,10].
> >
> > **Empirical Results**
> >
> > We conducted another ablation experiment to study your question from an empirical point-of-view and investigate in which cases the transformer architecture is indeed a crucial component.
> >
> > For the ablation, we exchange the transformer-encoder in our approach with an MLP using skip connections and batch normalization and ensure that the transformer encoder and the MLP have approximately the same number of parameters. We then ran the previous experiments using the different models (GLMs, FA, GMMs) in three different scenarios each; every scenario has 50 synthetic and 17 real-world datasets.
> >
> > In summary, the results confirm our initial hypothesis that a transformer encoder is important for complex conditioning on the input data.
> >
> > For the GLM scenarios, this is especially pronounced with substantial performance drops across all scenarios when using the MLP baseline instead of the transformer. In the case of FA and GMMs, the improvement of the transformer-encoder is smaller, albeit consistent. In particular, there is no case where the MLP performance is above one standard error of the transformer’s performance. For FA and GMMs, the gap between the MLP and the transformer encoder is, for example, especially noticeable in scenarios 1 and 3 of the FA cases on the real-world data and in scenarios 1 and 2 for the GMM cases.
> >
> > If there are any remaining concerns or questions, please let us know. If not, we would be grateful if you could consider raising the score.
> >
> > ### References
> >
> > [1] Dosovitskiy, Alexey, et al. "An Image is Worth 16x16 Words: Transformers for Image Recognition at Scale." International Conference on Learning Representations. 2020.
> >
> > [2] Dubey, Abhimanyu, et al. "The llama 3 herd of models." arXiv preprint arXiv:2407.21783 (2024).
> >
> > [3] Gorishniy, Yury, et al. "Revisiting deep learning models for tabular data." Advances in Neural Information Processing Systems 34 (2021): 18932-18943.
> >
> > [4] Müller, Samuel, et al. "Transformers Can Do Bayesian Inference." International Conference on Learning Representations.
> >
> > [5] Hollmann, Noah, et al. "TabPFN: A Transformer That Solves Small Tabular Classification Problems in a Second." The Eleventh International Conference on Learning Representations.
> >
> > [6] Panwar, Madhur, Kabir Ahuja, and Navin Goyal. "In-Context Learning through the Bayesian Prism." The Twelfth International Conference on Learning Representations.
> >
> > [7] Dong, Qingxiu, et al. "A survey on in-context learning." arXiv preprint arXiv:2301.00234 (2022).
> >
> > [8] Garg, Shivam, et al. "What can transformers learn in-context? a case study of simple function classes." Advances in Neural Information Processing Systems 35 (2022): 30583-30598.
> >
> > [9] Akyürek, Ekin, et al. "In-Context Language Learning: Architectures and Algorithms." Forty-first International Conference on Machine Learning.
> >
> > [10] Lee, Ivan, Nan Jiang, and Taylor Berg-Kirkpatrick. "Is attention required for ICL? Exploring the Relationship Between Model Architecture and In-Context Learning Ability." The Twelfth International Conference on Learning Representations.

---

> > > ### Author Response · Authors · 2024-11-27
> > > **Response to weakness 2 contd.: reasons for choosing a transformer- results part 1**
> > >
> > > For your convenience, we also include the results (in Appendix H in the revised manuscript) below.
> > >
> > > #### GLMs: Comparison when using an MLP-based encoder and a transformer encoder on 50 synthetic and 17 real-world datasets for three different scenarios.
> > >
> > > | **Scenario**  | **Type of Encoder** | **Synthetic Evaluation**       	|              	|              	| **Real-World Evaluation**   	|              	|              	|
> > > |---------------|----------------------|-------------------------------------|------------------|------------------|----------------------------------|------------------|------------------|
> > > |           	|                  	| **C2ST (↓)**                   	| **MMD (↓)** 	| **W₂ (↓)**  	| **C2ST (↓)**                	| **MMD (↓)** 	| **W₂ (↓)**  	|
> > > | **Scenario 2** | MLP              	| 0.942 (± 0.093)                	| 1.783 (± 1.048) | 2.503 (± 0.814) | 0.968 (± 0.012)             	| 1.528 (± 0.394) | 2.271 (± 0.315) |
> > > |           	| Transformer       	| 0.839 (± 0.072)                	| 0.707 (± 0.658) | 1.111 (± 0.300) | 0.768 (± 0.033)             	| 0.143 (± 0.089) | 0.411 (± 0.094) |
> > > | **Scenario 3** | MLP              	| 0.957 (± 0.075)                	| 2.236 (± 1.218) | 2.681 (± 1.130) | 0.972 (± 0.012)             	| 1.658 (± 0.450) | 2.076 (± 0.427) |
> > > |           	| Transformer       	| 0.611 (± 0.070)                	| 0.089 (± 0.114) | 0.423 (± 0.348) | 0.576 (± 0.027)             	| 0.037 (± 0.026) | 0.257 (± 0.044) |
> > > | **Scenario 5** | MLP              	| 0.845 (± 0.115)                	| 1.066 (± 0.859) | 1.166 (± 0.996) | 0.890 (± 0.055)             	| 1.223 (± 0.791) | 1.102 (± 0.383) |
> > > |           	| Transformer       	| 0.621 (± 0.063)                	| 0.067 (± 0.080) | 0.299 (± 0.195) | 0.610 (± 0.045)             	| 0.046 (± 0.020) | 0.242 (± 0.038) |
> > >
> > > #### FA: Comparison when using an MLP-based encoder and a transformer encoder on 50 synthetic and 17 real-world datasets for three different scenarios.
> > >
> > > | **Scenario**  | **Type of Encoder** | **Synthetic Evaluation**       	|              	|              	| **Real-World Evaluation**   	|              	|              	|
> > > |---------------|----------------------|-------------------------------------|------------------|------------------|----------------------------------|------------------|------------------|
> > > |           	|                  	| **C2ST (↓)**                   	| **MMD (↓)** 	| **W₂ (↓)**  	| **C2ST (↓)**                	| **MMD (↓)** 	| **W₂ (↓)**  	|
> > > | **Scenario 1** | MLP              	| 0.579 (± 0.015)                	| 0.017 (± 0.006) | 0.364 (± 0.029) | 0.634 (± 0.014)             	| 0.013 (± 0.004) | 0.331 (± 0.010) |
> > > |           	| Transformer       	| 0.552 (± 0.028)                	| 0.034 (± 0.034) | 0.289 (± 0.083) | 0.606 (± 0.038)             	| 0.068 (± 0.069) | 0.265 (± 0.078) |
> > > | **Scenario 2** | MLP              	| 0.562 (± 0.038)                	| 0.037 (± 0.042) | 0.308 (± 0.097) | 0.632 (± 0.068)             	| 0.182 (± 0.407) | 0.339 (± 0.174) |
> > > |           	| Transformer       	| 0.542 (± 0.006)                	| 0.017 (± 0.006) | 0.244 (± 0.033) | 0.622 (± 0.032)             	| 0.098 (± 0.039) | 0.287 (± 0.046) |
> > > | **Scenario 3** | MLP              	| 0.539 (± 0.025)                	| 0.023 (± 0.022) | 0.278 (± 0.116) | 0.680 (± 0.019)             	| 0.268 (± 0.044) | 0.253 (± 0.017) |
> > > |           	| Transformer       	| 0.537 (± 0.023)                	| 0.024 (± 0.021) | 0.259 (± 0.088) | 0.609 (± 0.019)             	| 0.124 (± 0.037) | 0.179 (± 0.018) |

---

> > > > ### Author Response · Authors · 2024-11-27
> > > > **Response to weakness 2 contd.: reasons for choosing a transformer- results part 2**
> > > >
> > > > #### GMMs: Comparison when using an MLP-based encoder and a transformer encoder on 50 synthetic and 17 real-world datasets for three different scenarios.
> > > >
> > > > | **Scenario**  | **Type of Encoder** | **Synthetic Evaluation**       	|              	|              	| **Real-World Evaluation**   	|              	|              	|
> > > > |---------------|----------------------|-------------------------------------|------------------|------------------|----------------------------------|------------------|------------------|
> > > > |           	|                  	| **C2ST (↓)**                   	| **MMD (↓)** 	| **W₂ (↓)**  	| **C2ST (↓)**                	| **MMD (↓)** 	| **W₂ (↓)**  	|
> > > > | **Scenario 1** | MLP              	| 0.873 (± 0.045)                	| 0.242 (± 0.363) | 2.203 (± 1.098) | 0.917 (± 0.067)             	| 0.891 (± 1.150) | 4.528 (± 2.701) |
> > > > |           	| Transformer       	| 0.760 (± 0.092)                	| 0.303 (± 0.548) | 2.095 (± 1.692) | 0.847 (± 0.082)             	| 0.486 (± 0.623) | 4.054 (± 2.782) |
> > > > | **Scenario 2** | MLP              	| 0.921 (± 0.035)                	| 0.291 (± 0.205) | 2.870 (± 0.710) | 0.992 (± 0.005)             	| 0.399 (± 0.127) | 5.505 (± 1.144) |
> > > > |           	| Transformer       	| 0.812 (± 0.061)                	| 0.159 (± 0.154) | 2.314 (± 0.926) | 0.937 (± 0.041)             	| 0.282 (± 0.131) | 3.947 (± 1.055) |
> > > > | **Scenario 3** | MLP              	| 0.999 (± 0.000)                	| 0.438 (± 0.181) | 11.502 (± 9.719) | 1.000 (± 0.000)             	| 1.001 (± 0.149) | 26.282 (± 3.731) |
> > > > |           	| Transformer       	| 0.999 (± 0.001)                	| 0.267 (± 0.154) | 7.234 (± 2.974) | 1.000 (± 0.000)             	| 1.155 (± 0.258) | 26.956 (± 3.114) |

---

> ### Author Response · Authors · 2024-11-24
> **The end of the rebuttal phase is approaching**
>
> Dear Reviewer rWy9,
>
> Thank you again for providing highly valuable feedback on our manuscript . We are pleased that you point out that we propose an “innovative framework for Bayesian inference”,  and include “strong empirical comparisons”.
>
> As the discussion period will end in approximately 72 hours, we would like to ask if our answers so far have successfully addressed your concerns and answered your questions.
>
> In summary, we addressed the points raised by you as follows: We added a table comparing the inference time of our ICL method with that of all other approaches, which shows that ICL is consistently faster than HMC during deployment. Additionally, we conducted an ablation study using a diffusion objective to address your concern regarding the reliance on continuous normalizing flows. Our new empirical results confirm the original choice of using flow matching with optimal transport paths.
>
> Thank you again for your time. If there are any remaining concerns, please let us know. In case we sufficiently addressed your questions, we would greatly appreciate it if you consider raising the score.

---

> ### Author Response · Authors · 2024-11-30
> **The end of the extended rebuttal phase is approaching**
>
> Dear Reviewer rWy9,
>
> We would like to kindly remind you that the extended rebuttal period will end in less than 72 hours.
>
> We would be happy to hear if we have addressed your concerns so far and if you have further questions. In case we have sufficiently addressed your questions, we would greatly appreciate it if you consider raising the score.
>
> Thank you again for your time.

---

### Official Review · Reviewer_nf5T · 2024-11-04

**Soundness:** 3
**Presentation:** 3
**Contribution:** 2
**Rating:** 5
**Confidence:** 2

**Summary:**

This work aims to investigate whether the In-context learning (ICL) using the transformer can yield similar samples with fully Bayesian inference. To this end, the authors first set the distribution of the dataset, induced by GLM models and factors models, as the target distribution. Then, they demonstrate that when the transformer-based model is trained using the variant of ICL loss $d_{CFM}$ and continuous normalization flow (CNF), then the trained model can yield the posterior distribution that more accurately approximates the targeted posterior than the existing laplace and variational inference methods.

**Strengths:**

* This work demonstrates that in-context learning can yield the posterior distribution of datasets on more complex real datasets as the posterior distribution obtained by fully Bayesian inference can sample.

* This work is well written to understand the proposed methodology.

**Weaknesses:**

* Based on [1], it has been already investigated that transformer-based in-context learning (ICL) can do Bayesian inference.
Without presenting the advantage of fully Bayesian inference of ICL,  demonstrating the extension of ICL capability, seems incremental contribution.




* In experiment, the baseline models for Laplace and VI except for (IAF) seem small compared to the size of the transformer used for ICL because the baseline of LA and VI use GLM or Factor models, whereas the ICL uses the transformer.


[1] Transformers Can Do Bayesian Inference - ICLR 22

**Questions:**

*   Why do the authors consider extending the ICL capability to fully Bayesian inference? What is the advantage of extending the ICL capability to full Bayesian inference beyond Bayesian inference? In this work, most experiments seem to focus on demonstrating the good approximation of the true posterior distribution for GLM and the factor models by the ICL without its upcoming benefits.


* To strengthen the novelty of this work, it would be better if the authors could demonstrate that the ICL capability of being good at posterior approximation, can be effective for improving the performance of Bayesian optimization or few-shot learning as conducted in [1].


* If ICL uses other distribution match methods such as VAE or Score matching instead of the continuous normalization flow (CNF), could the ICL still approximate the posterior distribution accurately? It seems less clear how important the type of loss in Eq. (5) is in the sense of accurate approximation for ICL.  It would be better to conduct the ablation study comparing the loss of Eq. (5) with the loss of other distribution matching methods, and thus explaining why the loss of Eq. (5) is necessary.


* In Tables 1, 2, 3, and 4, how complex models are used for the baseline of VI and Laplace approximation (LA)? Could you compare the mode size used for LA, VI, and ICL? If the model size of ICL is too large compared to that of LA and VI, I am worried about whether this comparison is fair.  If each model size is different, could compare the LA, VI, and ICL under the same model size ?


[1] Transformers Can Do Bayesian Inference - ICLR 22

---

> ### Author Response · Authors · 2024-11-20
> **Response to questions 1 & 2**
>
> Thank you for taking the time to read our manuscript and providing detailed feedback.
>
> We address your questions and comments below.  We use double quotation marks (") to mark parts directly quoted from the revised manuscript. In the manuscript itself, all added parts have the text color blue.
>
> > Why do the authors consider extending the ICL capability to fully Bayesian inference? What is the advantage of extending the ICL capability to full Bayesian inference beyond Bayesian inference?
>
> In contrast to previous literature such as [1], that focuses on the posterior predictive distributions, we tackle a much more complex and challenging problem, namely learning the posterior of latent variables. In contrast to [1], this posterior is multivariate and thus much more difficult to obtain. Except for cases where the model is linear in the parameters and a conjugate prior is chosen, obtaining such a posterior distribution of latent variables usually requires MCMC methods and a careful choice of the sampling procedure. While variational and local approximations exist, these do not provide nominal coverage guarantees. Hence neither [1] nor variational approaches would be able to, e.g., provide a 95%-credible interval for the effect of the covariates on medical charges in a hospital. Full Bayesian inference is a fundamental approach in probabilistic modeling which is basically always desirable, even though often infeasible.
>
> To emphasize why full Bayesian inference is a central paradigm, we added some extra content in Section 1.1 and marked it blue in the updated manuscript. We think that this addition will provide valuable context especially to readers less familiar with Bayesian inference.
>
> Below is the added text quoted from the revised manuscript for your convenience:
>
> “Full Bayesian inference is thus fundamental for a wide range of applications in Bayesian statistics and probabilistic machine learning, even though it cannot always be achieved.”
>
> > To strengthen the novelty of this work, it would be better if the authors could demonstrate that the ICL capability of being good at posterior approximation, can be effective for improving the performance of Bayesian optimization or few-shot learning as conducted in [1].
>
> Our scope is fundamentally different from that of Mueller et al. [1]. While [1] shows that transformers can learn univariate and discrete posterior **predictive** distributions, we target continuous multivariate distribution posteriors of latent variables. This implies that the tasks considered in [1], for instance on Bayesian Optimization, are not well-suited to benchmark our approach.
>
> Thank you again for your constructive feedback. Please let us know if our response addresses your questions and concerns, and if there is anything else we need to clarify.
>
> ### References
>
> [1] Müller, Samuel, et al. "Transformers Can Do Bayesian Inference." International Conference on Learning Representations.

---

> ### Author Response · Authors · 2024-11-20
> **Response to question 3**
>
> > If ICL uses other distribution match methods such as VAE or Score matching instead of the continuous normalization flow (CNF), could the ICL still approximate the posterior distribution accurately? It seems less clear how important the type of loss in Eq. (5) is in the sense of accurate approximation for ICL. It would be better to conduct the ablation study comparing the loss of Eq. (5) with the loss of other distribution matching methods, and thus explaining why the loss of Eq. (5) is necessary.
>
> We would like to highlight that [2] provides theoretical justification for the use of continuous normalizing flows and empirically demonstrates that flow matching is a more robust and efficient objective compared to various variants of Diffusion objectives. Similarly, [3] corroborates these findings across various simulation-based inference tasks closely related to those discussed in our paper.
>
> To further support our claims, we incorporated additional comprehensive empirical results across three selected scenarios for GLMs, GMMs, and FA with 50 synthetic and 17 real-world datasets each into the manuscript by replacing the flow matching objective with optimal transport (OT) paths from our original  approach with variance preserving (VP)  diffusion probability paths as proposed in [4]. A comparison of samples generated using VP diffusion probability paths and our original flow matching implementation with OT transport paths reveals that our initial approach consistently produces higher-quality samples.
>
> Thank you again for your constructive feedback. Please let us know if our response addresses your questions and concerns, and if there is anything else we need to clarify.
>
> ### References
>
> [2] Lipman, Yaron, et al. "Flow matching for generative modeling." arXiv preprint arXiv:2210.02747 (2022).
>
> [3] Wildberger, Jonas, et al. "Flow matching for scalable simulation-based inference." Advances in Neural Information Processing Systems 36 (2024).
>
> [4] Song, Yang, et al. "Score-based generative modeling through stochastic differential equations." arXiv preprint arXiv:2011.13456 (2020).

---

> > ### Author Response · Authors · 2024-11-20
> > **Response to question 3: results**
> >
> > For your convenience, we also include the results when comparing samples based on the VP diffusion probability paths and our initial flow matching implementation with OT transport paths (in Appendix C in the revised manuscript) below.
> >
> > #### GLMs: Comparison of the OT flow matching and the VP diffusion objective on 50 synthetic and 17 real-world datasets for three different scenarios. All results within two standard errors of the best average result for each scenario are marked in **bold**.
> >
> > | Scenario   | Model         	| C2ST (↓)       	| MMD (↓)        	| W2 (↓)         	| C2ST (↓)       	| MMD (↓)        	| W2 (↓)         	|
> > |------------|-------------------|--------------------|--------------------|--------------------|--------------------|--------------------|--------------------|
> > | **Scenario 2** | Diffusion paths  | 0.961 (± 0.040)	| **1.525** (± 0.777) | 3.354 (± 1.333)	| 0.961 (± 0.016)	| 1.347 (± 0.365)	| 2.025 (± 0.270)	|
> > |        	| **OT paths**   	| **0.839** (± 0.072) | **0.707** (± 0.658) | **1.111** (± 0.300) | **0.768** (± 0.033) | **0.143** (± 0.089) | **0.411** (± 0.094) |
> > | **Scenario 3** | Diffusion paths  | 0.903 (± 0.111)	| 1.080 (± 0.564)	| 1.733 (± 0.408)	| 0.936 (± 0.013)	| 1.002 (± 0.203)	| 1.442 (± 0.103)	|
> > |        	| **OT paths**   	| **0.611** (± 0.070) | **0.089** (± 0.114) | **0.423** (± 0.348) | **0.576** (± 0.027) | **0.037** (± 0.026) | **0.257** (± 0.044) |
> > | **Scenario 5** | Diffusion paths  | **0.691** (± 0.074) | 0.211 (± 0.143)	| 0.708 (± 0.233)	| **0.681** (± 0.038) | 0.182 (± 0.093)	| **0.554** (± 0.090) |
> > |        	| **OT paths**   	| **0.621** (± 0.063) | **0.067** (± 0.080) | **0.299** (± 0.195) | **0.610** (± 0.045) | **0.046** (± 0.020) | **0.242** (± 0.038) |
> >
> >
> >
> > #### FA: Comparison of the OT flow matching and the VP diffusion objective on 50 synthetic and 17 real-world datasets for three different scenarios. All results within two standard errors of the best average result for each scenario are marked in **bold**.
> >
> > | Scenario   | Model         	| C2ST (↓)       	| MMD (↓)        	| W2 (↓)         	| C2ST (↓)       	| MMD (↓)        	| W2 (↓)         	|
> > |------------|-------------------|--------------------|--------------------|--------------------|--------------------|--------------------|--------------------|
> > | **Scenario 1** | Diffusion paths  | 0.622 (± 0.043)	| 0.207 (± 0.121)	| 0.692 (± 0.192)	| **0.595** (± 0.012) | 0.089 (± 0.011)	| 0.475 (± 0.019)	|
> > |        	| **OT paths**   	| **0.552** (± 0.028) | **0.034** (± 0.034) | **0.289** (± 0.083) | **0.606** (± 0.038) | **0.068** (± 0.069) | **0.265** (± 0.078) |
> > | **Scenario 2** | Diffusion paths  | 0.826 (± 0.036)	| 0.768 (± 0.238)	| 1.219 (± 0.276)	| 0.878 (± 0.028)	| 0.793 (± 0.154)	| 1.056 (± 0.084)	|
> > |        	| **OT paths**   	| **0.542** (± 0.006) | **0.017** (± 0.006) | **0.244** (± 0.033) | **0.622** (± 0.032) | **0.098** (± 0.039) | **0.287** (± 0.046) |
> > | **Scenario 3** | Diffusion paths  | 0.751 (± 0.048)	| 0.387 (± 0.216)	| 0.834 (± 0.163)	| 0.944 (± 0.008)	| 1.514 (± 0.056)	| 1.332 (± 0.028)	|
> > |        	| **OT paths**   	| **0.537** (± 0.023) | **0.024** (± 0.021) | **0.259** (± 0.088) | **0.609** (± 0.019) | **0.124** (± 0.037) | **0.179** (± 0.018) |
> >
> >
> >
> > #### GMMs: Comparison of the OT flow matching and the VP diffusion objective on 50 synthetic and 17 real-world datasets for three different scenarios. All results within two standard errors of the best average result for each scenario are marked in **bold**.
> >
> > | Scenario   | Model         	| C2ST (↓)       	| MMD (↓)        	| W2 (↓)         	| C2ST (↓)       	| MMD (↓)        	| W2 (↓)         	|
> > |------------|-------------------|--------------------|--------------------|--------------------|--------------------|--------------------|--------------------|
> > | **Scenario 1** | Diffusion paths  | 0.924 (± 0.024)	| **0.241** (± 0.381) | **2.195** (± 1.431) | 0.958 (± 0.030)	| 0.890 (± 0.912)	| **5.328** (± 2.544) |
> > |        	| **OT paths**   	| **0.760** (± 0.092) | **0.303** (± 0.548) | **2.095** (± 1.692) | **0.847** (± 0.082) | **0.486** (± 0.623) | **4.054** (± 2.782) |
> > | **Scenario 2** | Diffusion paths  | 0.942 (± 0.020)	| **0.213** (± 0.187) | **2.748** (± 0.659) | **0.984** (± 0.012) | **0.411** (± 0.162) | **5.397** (± 1.458) |
> > |        	| **OT paths**   	| **0.812** (± 0.061) | **0.159** (± 0.154) | **2.314** (± 0.926) | **0.937** (± 0.041) | **0.282** (± 0.131) | **3.947** (± 1.055) |
> > | **Scenario 3** | Diffusion paths  | **1.000** (± 0.000) | 0.582 (± 0.280)	| **8.708** (± 4.945) | **1.000** (± 0.000) | 1.869 (± 0.342)	| **33.230** (± 8.095) |
> > |        	| **OT paths**   	| **0.999** (± 0.001) | **0.267** (± 0.154) | **7.234** (± 2.974) | **1.000** (± 0.000) | **1.155** (± 0.258) | **26.956** (± 3.114) |

---

> ### Author Response · Authors · 2024-11-20
> **Response to question 4**
>
> > In Tables 1, 2, 3, and 4, how complex models are used for the baseline of VI and Laplace approximation (LA)? Could you compare the mode size used for LA, VI, and ICL?
>
> Since our approach operates in-context and is thus a meta-learner (learning multiple tasks at the same time whereas LA, VI, etc. only learn on one specific dataset), it is impossible to directly compare the numbers of parameters. From a theoretical perspective, all benchmarked methods (variational inference, Hamiltonian Monte Carlo, and our in-context learning approach) implement inference for the same models (generalized linear models, factor analysis, Gaussian mixture models), which therefore always have the same number of parameters. While these other methods thus require solving a separate optimization problem for every single task, our method amortizes this computation via the meta-learned ICL model.
> In order to provide some comparison between the methods, we have however performed a runtime benchmark. The benchmark highlights the efficiency of our ICL approach during the deployment phase.  While some VI methods are faster than ICL, at the cost of inferior performance, our approach is consistently faster than HMC (for FA even achieving a factor 8 of speed-up).
>
> For your convenience, we also include the results (in Appendix E in the revised manuscript) below.
> | **Scenario** | **Method**              | **Mean Runtime (s)**       |
> |--------------|--------------------------|----------------------------|
> | **GLM**      | Laplace Approximation    | 10.48 ± 0.25          |
> |              | VI: DiagonalNormal       | 12.02 ± 0.26           |
> |              | VI: MultivariateNormal   | 13.70 ± 0.29          |
> |              | VI: Structured Normal    | 19.81 ± 0.98               |
> |              | VI: IAF                  | 15.44 ± 0.30               |
> |              | HMC                      | 120.24 ± 13.94             |
> |              | **ICL**                  | 107.79 ± 17.36             |
> | **FA**       | Laplace Approximation    | 17.85 ± 0.21           |
> |              | VI: DiagonalNormal       | 20.94 ± 0.66           |
> |              | VI: MultivariateNormal   | 20.84 ± 0.28          |
> |              | VI: Structured Normal    | 36.17 ± 0.61               |
> |              | VI: IAF                  | 23.75 ± 0.38               |
> |              | HMC                      | 248.26 ± 57.88             |
> |              | **ICL**                  | 31.49 ± 4.97               |
> | **GMM**      | Laplace Approximation    | 27.52 ± 0.40           |
> |              | VI: DiagonalNormal       | 29.74 ± 0.57           |
> |              | VI: MultivariateNormal   | 30.50 ± 0.41           |
> |              | VI: Structured Normal    | 42.44 ± 0.44               |
> |              | VI: IAF                  | 33.39 ± 0.49               |
> |              | HMC                      | 239.67 ± 32.71             |
> |              | **ICL**                  | 93.88 ± 10.47              |
>
>
> Thank you again for your constructive feedback. Please let us know if our response addresses your questions and concerns, and if there is anything else we need to clarify.

---

> ### Author Response · Authors · 2024-11-24
> **The end of the rebuttal phase is approaching**
>
> Dear Reviewer nf5T,
>
> Thank you again for providing highly valuable feedback on our manuscript. We are pleased that you find our manuscript “well written”, and point out that our ICl approach can yield the posterior distribution on complex real-world datasets.
>
> As the discussion period will end in approximately 72 hours, we would like to ask if our answers so far have successfully addressed your concerns and answered your questions.
>
> In summary, we addressed the points raised by you as follows: We explain why extending ICL to full Bayesian inference is a valuable contribution. We also clarify this point further in the main body of the manuscript. Additionally, we conducted an ablation study using a diffusion objective to address your concern regarding the reliance on continuous normalizing flows, which confirms our original choice of using flow matching with optimal transport paths.
>
> To compare the computational expense for running ICL, LA, VI and HMC in practical applications, we have included the runtimes of all methods in the paper and find that ICL is consistently faster than HMC during inference.
>
> Thank you again for your time. If there are any remaining concerns, please let us know. In case we sufficiently addressed your questions, we would greatly appreciate it if you consider raising the score.

---

> ### Author Response · Authors · 2024-11-30
> **The end of the extended rebuttal phase is approaching**
>
> Dear Reviewer nf5T,
>
> We would like to kindly remind you that the extended rebuttal period will end in less than 72 hours.
>
> We would be happy to hear if we have addressed your concerns so far and if you have further questions. In case we have sufficiently addressed your questions, we would greatly appreciate it if you consider raising the score.
>
> Thank you again for your time.

---

### Author Response · Authors · 2024-12-03
**Summary of updates from the authors**

Dear Area Chair and dear Reviewers,

We would like to thank the Reviewers again for their valuable feedback and comments. We are thankful for the active participation of reviewers **EBeR** and **prH2** during the rebuttal phase and are glad that all their concerns were resolved.

We are furthermore confident that our new ablation study thoroughly testing a diffusion objective as an alternative addresses the concern of reviewer **nf5T** regarding the choice to use flow matching. This ablation, combined with new results where we benchmark an MLP-baseline, also directly addresses the question of reviewer **rWy9** about architectural aspects of our method. Finally, in response to points related to runtime and complexity considerations raised by reviewers **nf5T**, **rWy9**, and **prH2**, we included the time needed for training and inference for all methods benchmarked in our paper.

We would like to summarize our revision and point out the key improvements to our manuscript:
- **Justifications for using flow matching**: We conducted an ablation study demonstrating that using a flow matching objective with optimal transport paths yields substantially better results than an objective with variance preserving diffusion paths.

- **Architectural choices**: Our additional experimental results show that using an alternative architecture based on an MLP with the same size leads to consistently worse results than relying on the originally proposed transformer architecture.

- **Performance under a mismatch of train and test data**: Our new ablation study investigates how our ICL approach performs under a distribution shift between train and test data and reveals that our method can handle those situations robustly across different scenarios and setups.

- **Runtime concerns**: We included runtime comparison of all benchmarked methods showing that our ICL method is consistently faster than HMC.

- **Comparison with SGLD**: We now also benchmark against SGLD and find that it leads to samples with inferior quality compared to our ICL method.

- **Clarifications for how flow matching is used in our approach**: To further clarify how flow matching is used in our approach, we added a new section to the revised version of the manuscript that details its role during training and inference.

Again, we greatly appreciate the reviewers’ feedback. We firmly believe that we have thoroughly addressed the concerns and questions raised by all reviewers and thank them for helping to substantially improve our manuscript.

---

### Meta-Review · Area_Chair_pmBo · 2024-12-23

**Metareview:**

This paper develops a meta-learning approach for Bayesian inference. The “in-context” learner takes as input, a datum x, and learns to predict the posterior on the latents p(z | x) using data collected from a continuous normalizing flow. Multiple synthetic training datasets are used to train an attention-based network. These synthetic data are from generalized linear models, Gaussian mixture models, and factor analysis.

This paper resulted in an extensive discussion. The updated manuscript is substantially improved from the initial version, and the authors have done a wonderful job of adding more ablation experiments, clarifying some of the details of their approach, and positioning the approach more carefully.

Let me summarize the main concern that has not been addressed. For an approach that is so general, one needs to develop a more rigorous understanding of when it will work and when it will not work. For example, the meta-learner will not work on “out of distribution” samples and while the authors have analyzed this experimentally, I would encourage them to modify the title and introduction to reflect this issue. It stands to reason that the only way to meta-learn Bayesian inference, is if the underlying set of problems share some regularities, e.g., all of them come from GLM models. And this is what the authors have shown. The evidence presented in this paper does not mean that the meta-learner will generalize to Bayesian inference for other distributions, e.g., using training data from a GLM but predicting for a GMM. This needs to be reflected in the narrative of the paper.

The current manuscript is very poorly written. I would recommend this not be accepted on the basis of the reviewer-author discussion. But this decision can be bumped up.

**Additional Comments On Reviewer Discussion:**

Reviewer EBeR had a broadly positive rating but they were concerned about potential limitations of such as general approach, and wanted a better analysis of the synthetic data experiments. The authors discussed answers to these questions. After this discourse, the reviewer has raised their score to 8. There was some confusion about one of the questions: can SGLD be used to create the training samples....the authors seem to have conducted experiments to use SGLD as a sampler for the latent variables, not as a replacement of continuous normalizing flows. I found this quite surprising.

Reviewer prH2 had questions on positioning the approach (meta-learning vs in-context learning), calculating the posterior, and some details of the method. The authors have responded to these questions satisfactorily and also modified the manuscript to take these comments into account.

Reviewer nf5T had concerns about connections to existing work, the fact the the approach of this paper uses conditional normalizing flows to train but compares to variational inference or score-matching during evaluations, size of the model etc. I share these concerns. The authors have addressed some of these concerns, except one I think. Instead of training a network to replication something akin to variational inference or score matching, the authors instead modified their approach to use optimal-transform based trajectories (instead of those from conditional normalizing flow) for training. This does not really address Reviewer nf5T (and my) concern.

Reviewer rWy9 had concerns about the computational cost of the in-context learning approach and similar concerns as those of Reviewer nf5T on why continuous normalizing flows are being used here. The authors had responded to the first concern satisfactorily, but not the second one. They also added a few more ablation experiments on using multi-layer perceptrons instead of self-attention-based networks to encode the test datum x.

---

### Decision · Program_Chairs · 2025-01-22

Reject